# Functional Equivalence in Attention:
# A Comprehensive Study with Applications to Linear Mode Connectivity

Viet-Hoang Tran[* 1]  Vinh Khanh Bui[2]  Van-Hoan Trinh[3]  Tan Lai Ngoc[4]  Tan M. Nguyen[1]

## Abstract

Neural network parameter spaces are inherently non-injective, as distinct parameter configurations can realize identical functions through functional equivalence. While this symmetry is well understood in classical fully connected and convolutional models, it becomes substantially more intricate in modern attention-based architectures. Existing analyses of multihead attention have largely focused on the vanilla formulation, overlooking positional encodings that fundamentally reshape architectural symmetries. In this work, we provide a formal study of functional equivalence in Transformers with positional encodings. Focusing on the two most widely used variants–sinusoidal and rotary positional encodings (RoPE)–we show that sinusoidal encodings preserve the equivalence structure of vanilla attention, whereas rotary encodings significantly reduce the symmetry group, thereby enhancing expressivity. This offers a principled explanation for the growing prominence of RoPE in practice. We further examine how positional encodings affect linear mode connectivity, and through an alignment algorithm, empirically demonstrate that the presence and variability of connectivity across Transformer settings crucially depend on the positional encoding.

## 1. Introduction

The training of deep neural networks reveals a seeming paradox: despite the high dimensionality and non-convexity of the loss landscape with numerous local minima, simple optimization methods such as stochastic gradient descent (SGD) consistently discover solutions that generalize well.

[1]National University of Singapore [2]Center for AI Research, VinUniversity [3]Technical University of Munich [4]Independent Researcher. Correspondence to: Viet-Hoang Tran <hoang.tranviet@u.nus.edu>.

*Proceedings of the 43rd International Conference on Machine Learning*, Seoul, South Korea. PMLR 306, 2026. Copyright 2026 by the author(s).

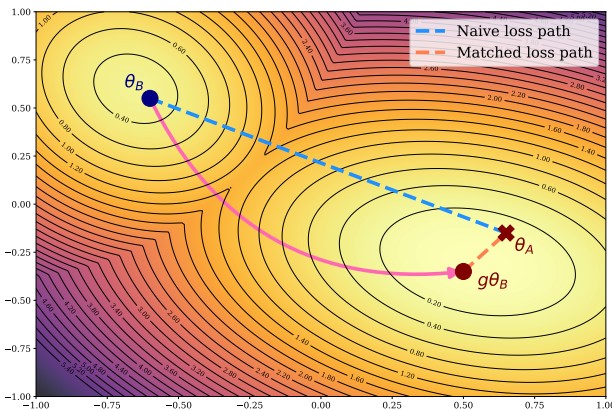

*Figure 1.* Illustration of Linear Mode Connectivity

**(Linear) Mode Connectivity.** One influential perspective on this phenomenon is offered by the concept of *mode connectivity* (MC) (Frankle, 2020; Keskar et al., 2017; Sagun et al., 2018; Venturi et al., 2019; Neyshabur et al., 2020; Tatro et al., 2020; Yunis et al., 2022; Zhou et al., 2023), which reveals that solutions discovered through independent optimization trajectories are rarely isolated; rather, they lie within extensive connected manifolds of parameters yielding comparably low loss. A particularly tractable instance of this principle is *linear mode connectivity* (LMC) (Frankle et al., 2020; Entezari et al., 2022), in which two trained models can be joined by a straight-line interpolation in parameter space that remains confined to a low-loss region. Formally, consider a model $f(\cdot; \theta)$ parameterized by $\theta$, with loss function $\mathcal{L}(\theta) \geq 0$. Optimization amounts to minimizing $\mathcal{L}(\theta)$ over $\Theta$. Two solutions $\theta_A, \theta_B \in \Theta$ are said to exhibit LMC when the associated *loss barrier* (Frankle et al., 2020; Entezari et al., 2022) vanishes (or is negligible):

$$B(\theta_A, \theta_B) := \sup_{t \in [0,1]} \big[ \mathcal{L}\big(t\theta_A + (1-t)\theta_B\big) - t\mathcal{L}(\theta_A) - (1-t)\mathcal{L}(\theta_B) \big] \approx 0.$$

Empirical investigations have revealed that independently trained networks on small datasets are often connected by low-loss paths (Freeman & Bruna, 2017; Garipov et al., 2018; Draxler et al., 2018), and even that nearly arbitrary pairs of solutions can be joined through curves of low error (Garipov et al., 2018). MC sheds light on the effec-

tiveness of weight-space ensembling, known to improve generalization (Izmailov et al., 2018; Ramé et al., 2022; Wortsman et al., 2022) and has been applied to adversarial robustness (Zhao et al., 2020), generalization theory (Pittorino et al., 2022; Juneja et al., 2023; Lubana et al., 2023), loss landscape geometry (Gotmare et al., 2018; Vlaar & Frankle, 2022; Lucas et al., 2021), and more recently, continual learning (Wen et al., 2023; Kozal et al., 2024; Chen et al., 2023a) and ensemble methods (Kanoh & Sugiyama, 2025; Kim et al., 2025).

**Attention Mechanism and Positional Encoding.** The attention mechanism is inherently permutation invariant, necessitating positional encoding (PE) to capture token order (Vaswani et al., 2017). Early models employed Absolute PEs (APEs), either sinusoidal or learnable embeddings (Gehring et al., 2017), which became standard in seminal architectures such as BERT (Devlin et al., 2019), GPT-2 (Radford et al., 2019), and ViT (Dosovitskiy et al., 2021). While effective, APEs treat absolute positions as the sole signal, limiting robustness under local reordering. Relative PEs (RPEs) address this by encoding pairwise distances into attention weights (Shaw et al., 2018), a design later adopted in many models (Dai et al., 2019; He et al., 2021; Raffel et al., 2020). Among recent advances, Rotary PE (RoPE) (Su et al., 2024) encodes relative position via angular rotations of query–key vectors, preserving dot-product structure and enabling both translation equivariance and long-sequence extrapolation. RoPE is now widely adopted in state-of-the-art models (Touvron et al., 2023a; Chowdhery et al., 2023; Nijkamp et al., 2023; DeepSeek-AI, 2024; 2025; OpenAI, 2025; Bai et al., 2025; Yang et al., 2025), attesting to its robustness in large-scale settings.

**Functional Equivalence.** A major difficulty in characterizing LMC lies in the *permutation invariance* of neural networks: reordering hidden units does not alter the underlying function (Brea et al., 2019; Novak et al., 2018), yet such symmetries can cause functionally identical models to appear distant in parameter space (Allen-Zhu et al., 2019; Du et al., 2019; Frankle & Carbin, 2019; Belkin et al., 2019; Neyshabur et al., 2018). This phenomenon is subsumed under the broader framework of *functional equivalence* (Hecht-Nielsen, 1990; Fefferman & Markel, 1993; Kurková & Kainen, 1994; Albertini & Sontag, 1993b;a), which seeks to describe when distinct parameterizations realize the same input–output mapping. To address this issue, recent studies have examined LMC *up to permutation*, where low-loss paths are revealed once hidden units are properly aligned (Singh & Jaggi, 2020; Ainsworth et al., 2023; Guerrero-Peña et al., 2023; Ito et al., 2025a;b; Zhao et al., 2025). Theoretical results show that dropout-stable networks naturally exhibit mode connectivity (Kuditipudi et al., 2019; Shevchenko & Mondelli, 2020), while LMC under permutation alignment may already emerge at initial-

ization in the NTK regime (Entezari et al., 2022; Jacot et al., 2021), with rigorous guarantees recently established (Ferbach et al., 2024). These developments lend support to the convexity conjecture (Entezari et al., 2022), which views the SGD solution set as approximately convex once symmetries are accounted for. This view is strengthened by Sharma et al. (2024), who propose simultaneous linear connectivity, where a single model aligns linearly with multiple others. Additional studies explore the geometry of the solution space (Ainsworth et al., 2023; Xiao et al., 2024) and identify star-shaped regions conducive to LMC (Sonthalia et al., 2025).

**Alignment Algorithms.** These algorithms align parameters to establish LMC. (Entezari et al., 2022) proposed a simulated annealing-based algorithm. Singh & Jaggi (2020) employed Optimal Transport, while Akash et al. (2022) utilized the Wasserstein Barycenter. Ainsworth et al. (2023) introduced three methods: activation matching (using intermediate activations), weight matching (being data-independent), and the Straight-Through Estimator (minimizing interpolation loss via gradients); all are based on solving the Linear Assignment Problem (Kuhn, 2010; Jonker & Volgenant, 1987; Crouse, 2016). Guerrero-Peña et al. (2023) developed Sinkhorn re-basin, a differentiable method that improves alignment but struggles with residual connections due to layer-independent optimization.

**Contribution.** Recent work on the symmetry of vanilla attention (Tran et al., 2025; Knyazev et al., 2025) shows that head permutations and linear group actions capture all symmetries. Meanwhile, Theus et al. (2025) proposed a Transformer matching method, but it overlooks symmetry in the query-key and key-value components. In this paper, we study LMC in attention-based models, focusing on how PEs influence parameter symmetry of attention. The paper is organized as follows:

1. In Section 2, we recall the notion of Multihead Attention and its parameter space, together with the result characterizing functional equivalence in the vanilla case.

2. In Section 3, we analyze how positional encodings alter the internal structure of attention. We focus primarily on the most widely used encodings, Absolute PE and Relative PE. In particular, we study sinusoidal PE as a representative of APE and rotary PE as a representative of RPE, and show why results from the vanilla case do not extend directly to these settings.

3. In Section 4, we present the main result of the paper, which characterizes the full symmetry of attention with widely used positional encodings. This characterization underlies the matching algorithm for Multihead Attention described in Section 5.

4. In Section 6, we present empirical evidence of LMC

across a wide range of models and tasks, under diverse settings and across datasets of varying scales and modalities. We also evaluate the effectiveness of our proposed matching algorithms and conduct detailed ablation studies to validate their individual components.

A table of notations, theoretical foundations, and experimental details is included in the Appendix. Given the technical nature of our proofs, Appendix A provides a *consolidated overview* to help readers grasp the overall structure of our work without delving into all technical details.

## 2. Parameter Space of Multihead Attention

We present the formal definition of the Multihead Attention, describe its associated parameter space, and review results in the literature concerning its parameter space symmetry.

**Multihead Attention and its Parameter Space.** Let $d$, $L$, and $h$ be positive integers denoting the token dimension, sequence length, and number of heads, respectively. Define the space of all sequences of $d$-dimensional tokens by $\mathcal{S} := \sqcup_{L=1}^{\infty} \mathbb{R}^{L \times d}$. Given a fixed head dimension $d_h$, consider $W_i^Q, W_i^K, W_i^V, W_i^O \in \mathbb{R}^{d \times d_h}$ for each $i \in [h]$. For an input sequence $\mathbf{x} = (x_1, \ldots, x_L)^\top \in \mathbb{R}^{L \times d} \subset \mathcal{S}$, the Multihead Attention with $h$ heads is defined by

$$\text{MHA}\big(\mathbf{x}; \{W_i^Q, W_i^K, W_i^V, W_i^O\}_{i=1}^h\big) \qquad (1)$$
$$= \sum_{i=1}^h \text{softmax}\left(\left(\mathbf{x}W_i^Q\right)\left(\mathbf{x}W_i^K\right)^\top\right) \cdot \left(\mathbf{x}W_i^V\right)\left(W_i^O\right)^\top.$$

Here, the operator softmax is applied row-wise to the similarity matrix $(\mathbf{x}W_i^Q)(\mathbf{x}W_i^K)^\top \in \mathbb{R}^{L \times L}$, yielding the *attention matrix* associated with $\mathbf{x}$. Each row of this matrix represents a probability distribution that specifies the relative contributions of all input tokens to a given output token. The parameters and the parameter space of the MHA map is thus denoted as $\theta$ and $\Theta$, respectively, and given by

$$\theta := \big(W_i^Q, W_i^K, W_i^V, W_i^O\big)_{i=1}^h$$
$$\in \Theta(d, d_h, h) := \big(\mathbb{R}^{d \times d_h}\big)^{4h}. \quad (2)$$

Typically, the head dimension is set to $d_h = d/h$.

**Symmetry Group.** Define the following group

$$G_{\text{Att}}(d_h, h) := S_h \times \big(\text{GL}(d_h) \times \text{GL}(d_h)\big)^h.$$

This is precisely the direct product between the permutation group $S_h$ and $h$ copies of $\text{GL}(d_h) \times \text{GL}(d_h)$. Each element $g$ of $G_{\text{Att}}(d_h, h)$ has the form

$$g = (\sigma, (U_i, V_i)_{i=1}^h), \text{ where } \sigma \in S_h \text{ and } U_i, V_i \in \text{GL}(d_h).$$

The group $G_{\text{Att}}(d_h, h)$ acts naturally on the parameter space

$\Theta(d, d_h, h)$ via head permutations and linear transformations of the weight matrices, as follows:

$$g\theta := \big(W_{\sigma(i)}^Q \cdot U_i^\top, \; W_{\sigma(i)}^K \cdot U_i^{-1},$$
$$W_{\sigma(i)}^V \cdot V_i^\top, \; W_{\sigma(i)}^O \cdot V_i^{-1}\big)_{i=1}^h.$$

This action preserves the functionality of MHA maps: For every $\theta \in \Theta(d, d_h, h)$ and every $g \in G_{\text{Att}}(d_h, h)$, one has

$$\text{MHA}(\cdot; \theta) = \text{MHA}(\cdot; g\theta).$$

The general linear action cancels in matrix multiplications, while the permutation action induced by $\sigma$ commutes with addition. Together, these actions determine the symmetry of multihead attention, as stated in the following result.

**Theorem 2.1** (Tran et al. (2025)). *Given two* MHA *maps with $h$ and $\bar{h}$ heads, parameterized by*

$$\theta = (W_i^Q, W_i^K, W_i^V, W_i^O)_{i=1}^h \in G_{Att}(d_h, h), \text{ and}$$
$$\bar{\theta} = (\bar{W}_i^Q, \bar{W}_i^K, \bar{W}_i^V, \bar{W}_i^O)_{i=1}^{\bar{h}} \in G_{Att}(d_h, \bar{h}),$$

*respectively. Assume that*

*1. All matrices $W_i^Q, W_i^K, W_i^V, W_i^O$ and $\bar{W}_i^Q, \bar{W}_i^K, \bar{W}_i^V, \bar{W}_i^O$, for all feasible $i$, are of rank $d_h$; and,*

*2. From $\theta$, the matrices $\{W_i^Q(W_i^K)^\top\}_{i=1}^h$ are pairwise distinct. The same condition holds for $\bar{\theta}$.*

*If the two* MHA *maps are identical, then $h = \bar{h}$, and there exists $g \in G_{\text{Att}}(d_h, h)$ such that $\bar{\theta} = g\theta$.*

**Remark 2.2.** While the theorem requires mild genericity assumptions on the MHA parameters, these hold almost surely. Hence, outside a negligible subset of the parameter space (e.g., measure zero or a non-dense set), functional equivalence is completely characterized by the symmetry group. Such assumptions are standard in the literature on functional equivalence of neural architectures (Hecht-Nielsen, 1990; Fefferman & Markel, 1993; Phuong & Lampert, 2020), and we will adopt the same perspective in our results.

## 3. How Positional Encoding Alters Architectural Symmetry

We investigate how positional encodings (PEs) modify the internal structure of the attention mechanism. Our analysis primarily focuses on *sinusoidal* and *rotary* encodings, which are two widely used PEs. These serve as representatives of the two principal paradigms of positional encoding: absolute and relative, respectively. We examine how the formulation of Multihead Attention is altered under these schemes, and how the architectural symmetries are consequently affected. For now, we follow the standard implementation practice of assuming that both $d$ and $d_h$ are even.

### 3.1. Absolute Positional Encoding

**Sinusoidal Encoding.** In Absolute PEs, let $\mathbf{p} = \{p_i\}_{i=1}^{\infty} \subset \mathbb{R}^d$ denote the sequence of positional vectors, which encodes positional information. In the case of *sinusoidal encoding* from the original Transformer (Vaswani et al., 2017), the components of $p_m \in \mathbb{R}^d$ are defined as

$$p_{m,2k} = \sin\left(m \,/\, 10000^{2k/d}\right), \text{ and}$$
$$p_{m,2k+1} = \cos\left(m \,/\, 10000^{2k/d}\right),$$

for $0 \leq k < d/2$. For an input sequence $\mathbf{x} = (x_1, \ldots, x_L)^{\top}$ of length $L$, the positional encoding is incorporated by addition, namely $\mathbf{x} + \mathbf{p} = (x_1 + p_1, \ldots, x_L + p_L)^{\top}$ (this is an abuse of notation), which is then supplied as input to the multihead attention, yielding

$$\mathrm{MHA}_{\mathrm{SinusoidalPE}}(\mathbf{x}\,;\theta) = \mathrm{MHA}(\mathbf{x} + \mathbf{p}\,;\theta).$$

**Symmetry Group.** In this formulation, PE does not alter the internal structure of the MHA map; it merely applies a shift to the input. Moreover, the encoding map $\mathcal{S} \to \mathcal{S}$, defined by $\mathbf{x} \mapsto \mathbf{x} + \mathbf{p}$, is bijective. Consequently, the introduction of sinusoidal PE has no effect on the analysis of parameter symmetry for multihead attention. Thus, the functional equivalence classes in the presence of sinusoidal PE coincide exactly with those in the absence of PE.

### 3.2. Relative Positional Encoding

**Rotary Positional Encoding.** We next recall the *Rotary Positional Encoding* (RoPE) (Su et al., 2024). For a token at position $n$, define the block-diagonal rotation matrix $R_n \in \mathbb{R}^{d_h \times d_h}$ by

$$R_n = \mathrm{diag}\left( \begin{bmatrix} \cos(n\varphi_i) & -\sin(n\varphi_i) \\ \sin(n\varphi_i) & \cos(n\varphi_i) \end{bmatrix} \,:\, i \in [d_h/2] \right),$$

where $\varphi_i = 10000^{-2(i-1)/d}$ for $i \in [d_h/2]$. For brevity, we omit the explicit subscript indicating the head dimension $d_h$. Note that $R_n = (R_1)^n$. The multihead attention with RoPE is defined as

$$\mathrm{MHA}_{\mathrm{RoPE}}(\mathbf{x};\theta) = \sum_{i=1}^{h} \mathrm{softmax}$$
$$\left[ x_m W_i^Q R_{m-n} (W_i^K)^{\top} x_n^{\top} \right]_{m,n\in[L]} \cdot \mathbf{x} W_i^V (W_i^O)^{\top}.$$

**Effect on Internal Structure and Symmetry Group.** The parameterization and parameter space of $\mathrm{MHA}_{\mathrm{RoPE}}$ coincide with those of the standard MultiHead map defined in Equation (2). However, in contrast to the vanilla case, the action of $G_{\mathrm{Att}}(d_h, h)$ on $\Theta(d, d_h, h)$ no longer preserves functionality. Specifically, for $\theta \in \Theta(d, d_h, h)$ and $g \in G_{\mathrm{Att}}(d_h, h)$, it generally holds that

$$\mathrm{MHA}_{\mathrm{RoPE}}(\cdot;\theta) \neq \mathrm{MHA}_{\mathrm{RoPE}}(\cdot;g\theta).$$

The essential reason is as follows. While the interaction between $W_i^V$ and $W_i^O$ remains purely multiplicative and thus structurally consistent with the vanilla case, the matrices $W_i^Q$ and $W_i^K$ are now separated by the relative rotary matrix $R_{m-n}$. This insertion prevents the cancellation of group actions induced by $\mathrm{GL}(d_h)$, thereby violating the invariance property.

**Symmetry Group.** To define the symmetry group of $\mathrm{MHA}_{\mathrm{RoPE}}$, we first introduce, for each $i \in [d_h/2]$, the matrices $P_i, J_i \in \mathbb{R}^{d_h \times d_h}$. These are block-diagonal matrices with $d_h/2$ consecutive $2 \times 2$ blocks, where only the $i$-th block is nonzero:

$$P_i = \mathrm{diag}\Big(0, \ldots, 0, \underbrace{\begin{bmatrix} 1 & 0 \\ 0 & 1 \end{bmatrix}}_{i\text{-th block}}, 0, \ldots, 0\Big), \text{ and}$$
$$J_i = \mathrm{diag}\Big(0, \ldots, 0, \underbrace{\begin{bmatrix} 0 & -1 \\ 1 & 0 \end{bmatrix}}_{i\text{-th block}}, 0, \ldots, 0\Big).$$

Now define the following group

$$\mathrm{H}(d_h) := \Big\{ U = \textstyle\sum_{i=1}^{d_h/2}(a_i P_i + b_i J_i) \in \mathbb{R}^{d_h \times d_h} \,:\,$$
$$(a_i, b_i) \in \mathbb{R}^2 \setminus \{(0,0)\},\ i \in [d_h/2] \Big\}.$$

It is straightforward to verify that $\mathrm{H}(d_h)$ is an abelian subgroup of $\mathrm{GL}(d_h)$, and moreover isomorphic to $(\mathbb{C}^{\times})^{d_h/2}$, where $\mathbb{C}^{\times}$ denotes the multiplicative group of nonzero complex numbers. In particular, the rotary matrices $R_n$ belong to $\mathrm{H}(d_h)$ for all $n$. We then define

$$G_{\mathrm{RoPE}}(d_h, h) := S_h \times \big(\mathrm{H}(d_h) \times \mathrm{GL}(d_h)\big)^h.$$

It follows immediately that $G_{\mathrm{RoPE}}(d_h, h)$ is a subgroup of $G_{\mathrm{Att}}(d_h, h)$. Furthermore, the canonical action of $G_{\mathrm{Att}}(d_h, h)$ on $\Theta(d, d_h, h)$ restricts to a well-defined group action of $G_{\mathrm{RoPE}}(d_h, h)$ on $\Theta(d, d_h, h)$. Crucially, this restricted action preserves the functionality of the $\mathrm{MHA}_{\mathrm{RoPE}}$ map. In particular, for every $\theta \in \Theta(d, d_h, h)$ and every $g \in G_{\mathrm{RoPE}}(d_h, h)$, one has

$$\mathrm{MHA}_{\mathrm{RoPE}}(\cdot\,;\theta) = \mathrm{MHA}_{\mathrm{RoPE}}(\cdot\,;g\theta).$$

The justification is as follows. Compared to the standard MHA map, aside from the head permutation $\sigma$ and the interaction between $W_i^V$ and $W_i^O$, the only structural difference lies in the interaction between $W_i^Q$ and $W_i^K$. Since $\mathrm{H}(d_h)$ is abelian and $R_n$ belongs to $\mathrm{H}(d_h)$, one obtains

$$(W_i^Q U^{\top}) R_n (W_i^K U^{-1})^{\top} = W_i^Q U^{\top} R_n (U^{-1})^{\top} (W_i^K)^{\top}$$
$$= W_i^Q R_n U^{\top} (U^{-1})^{\top} (W_i^K)^{\top} = W_i^Q R_n (W_i^K)^{\top}.$$

Thus the similarity matrix inside the softmax of the $\mathrm{MHA}_{\mathrm{RoPE}}$ map remains invariant under $G_{\mathrm{RoPE}}$.

**Remark 3.1.** Our main result, presented next, shows that $G_{\mathrm{RoPE}}$ fully characterizes the symmetry structure of the

MHA$_{\text{RoPE}}$ map. Since $\text{H}(d_h)$ is substantially smaller than $\text{GL}(d_h)$, the function class represented by MHA$_{\text{RoPE}}$ is strictly larger than that of MHA or MHA$_{\text{SinusoidalPE}}$. *This finding offers a theoretical rationale for the increasing use of RoPE in attention-based models.*

# 4. Parameter Symmetry of Multihead Attention with RoPE

In this section, we examine the symmetry of multihead attention under a general formulation, of which the RoPE-based attention mechanism constitutes a special case.

## 4.1. A General Formulation of Multihead Attention

**General Multihead Attention.** Define a general MHA map with $h$ heads, parameterized by $\{\{A_i^{m,n}\}_{m,n\geq 1}\}_{i=1}^h$ and $\{B_i\}_{i=1}^h$, where $A_i^{m,n}, B_i \in \mathbb{R}^{d\times d}$, as follows: For an input sequence $\mathbf{x} = (x_1, \ldots, x_L)^\top \in \mathbb{R}^{L\times d}$,

$$\text{MHA}\left(\mathbf{x}\,;\left\{\{A_i^{m,n}\}_{m,n}, B_i\right\}_{i=1}^h\right) \tag{3}$$
$$= \sum_{i=1}^h \text{softmax}\left[x_m A_i^{m,n} x_n^\top\right]_{m,n=1,\ldots,L} \cdot \mathbf{x}B_i.$$

To facilitate the subsequent analysis, we impose two structural conditions:

**1. (Stationarity)** for all $m, n \geq 1$ and all shifts $k \geq 0$, we assume $A_i^{m,n} = A_i^{m+k,n+k}$, reflecting the natural shift-invariance induced by relative positional encodings; and,

**2. (Self-similarity symmetry)** for each $m \geq 1$, $A_i^{m,m}$ parameterizes the self-similarity score of the $m$-th token in head $i$. Since any quadratic form is uniquely represented by a symmetric matrix, we may replace $A_i^{m,m}$ by its symmetrization $\text{sym}(A_i^{m,m}) := \left(A_i^{m,m} + (A_i^{m,m})^\top\right)/2$ without changing the functionality, i.e.

$$x_m A_i^{m,m} x_m^\top = x_m \text{sym}(A_i^{m,m}) x_m^\top.$$

Henceforth, we assume that all $A_i^{m,m}$ are symmetric.

From now on, these two conditions will be imposed whenever the general MHA formulation is considered.

**Functional Equivalence of General MHA.** We now study the case where two general Multi-Head Attention maps, with $h$ and $\bar{h}$ heads respectively, yield identical functions:

$$\text{MHA}\left(\mathbf{x}\,;\left\{\{A_i^{m,n}\}_{m,n}, B_i\right\}_{i=1}^h\right)$$
$$= \text{MHA}\left(\mathbf{x}\,;\left\{\{\bar{A}_i^{m,n}\}_{m,n}, \bar{B}_i\right\}_{i=1}^{\bar{h}}\right),$$

which is equivalent to the fact that the following MHA map

with $h + \bar{h}$ heads is identically zero

$$0 = \text{MHA}\left(\mathbf{x}\,;\left\{\{A_i^{m,n}\}_{m,n}\right\}_{i=1}^h \sqcup \left\{\{\bar{A}_i^{m,n}\}_{m,n}\right\}_{i=1}^{\bar{h}},\right.$$
$$\left.\left\{B_i\right\}_{i=1}^h \sqcup \left\{-\bar{B}_i\right\}_{i=1}^{\bar{h}}\right).$$

Before presenting our result, we introduce the following notion. Two families $\{X_i\}_{i\in I}$ and $\{Y_i\}_{i\in I}$ are said to be *distinct* if there exists index $i \in I$ such that $X_i \neq Y_i$. The following theorem constitutes the main result of this section, offering a fundamental insight into the symmetry structure of general MHA.

**Theorem 4.1.** *Consider the* MHA *map with $h$ heads, parameterized by families of matrices*

$$\left\{\{A_i^{m,n}\}_{m,n}\right\}_{i=1}^h \text{ and } \left\{B_i\right\}_{i=1}^h \text{ in } \mathbb{R}^{d\times d},$$

*as in Equation (3). Assume that*

**1.** *The $h$ parameter families $\{A_1^{m,n}\}_{m,n}, \ldots, \{A_h^{m,n}\}_{m,n}$, are pairwise distinct,*

**2.** $A_i^{m,n}$ *is nonzero for all $i \in [h]$ and $m, n \geq 1$.*

*If the* MHA *map is identical to zero, then all matrices $B_1, \ldots, B_h$ are equal to zero.*

The proof of Theorem 4.1, provided in Appendix D.2, may be interpreted as a statement on the linear independence of attention heads. It proceeds by rewriting the identically vanishing MHA map – after clearing the softmax denominators – as an exponential polynomial that is identically zero, and then invoking tools from the theory of exponential polynomials. Although the proof is somewhat lengthy, we believe that the intuition behind Theorem 4.1 can be understood even without going through all technical details. In particular, the symmetry of MHA$_{\text{RoPE}}$ follows immediately as a corollary, requiring only additional arguments concerning the rotary matrices $R_n$.

## 4.2. The case of Multihead Attention with RoPE

The MHA$_{\text{RoPE}}$ map is subsumed by the general formulation in Equation (3). Indeed, define

$$A_i^{m,m} := \text{sym}\left(W_i^Q (W_i^K)^\top\right),$$
$$A_i^{m,n} := W_i^Q R^{m-n} (W_i^K)^\top \text{ if } m \neq n,$$
$$B_i := W_i^V (W_i^O)^\top.$$

Then MHA$_{\text{RoPE}}$ is precisely a special case of the general MHA formulation:

$$\text{MHA}_{\text{RoPE}}\left(\mathbf{x}\,;\left\{W_i^Q, W_i^K, W_i^V, W_i^O\right\}_{i=1}^h\right)$$
$$= \text{MHA}\left(\mathbf{x}\,;\left\{\{A_i^{m,n}\}_{m,n}, B_i\right\}_{i=1}^h\right). \tag{4}$$

The following result characterizes the symmetry of Multihead Attention with RoPE.

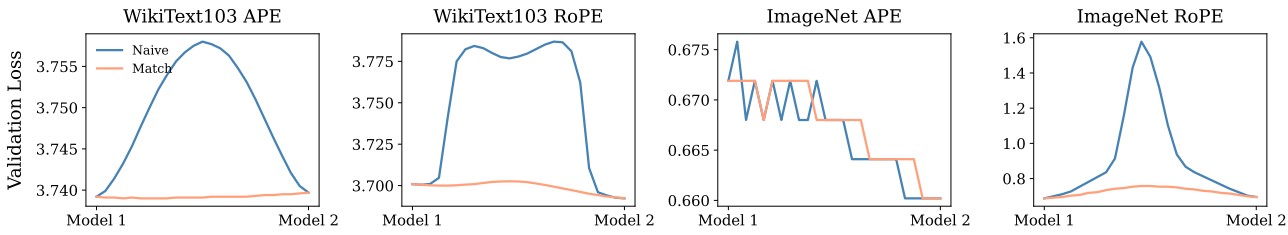

*Figure 2.* LMC interpolation plots for ViT on ImageNet-1K (subplots 3 and 4) and GPT-2 on WikiText103 (subplots 1 and 2), with APE and RoPE under first attention layer re-initialization.

**Theorem 4.2.** *Given two* $\mathrm{MHA_{RoPE}}$ *maps with $h$ and $\bar{h}$ heads, parameterized by*

$$\theta = (W_i^Q, W_i^K, W_i^V, W_i^O)_{i=1}^{h} \in G_{Att}(d_h, h), \text{ and}$$
$$\bar{\theta} = (\bar{W}_i^Q, \bar{W}_i^K, \bar{W}_i^V, \bar{W}_i^O)_{i=1}^{\bar{h}} \in G_{Att}(d_h, \bar{h}),$$

*respectively. Define*

$$A_i^0 := \mathrm{sym}\big(W_i^Q(W_i^K)^\top\big), \text{ and}$$
$$A_i^n := W_i^Q R_n(W_i^K)^\top \text{ if } n \neq 0.$$

*Assume that*

**1.** *From $\theta$, for each $i \in [h]$, the family $\{A_i^n\}_{n \in \mathbb{Z}}$ consist solely of nonzero matrices. Moreover, these form $h$ pairwise distinct families. The same condition holds for $\bar{\theta}$.*

**2.** *The matrices $W_i^Q$, $W_i^K$, $W_i^V$, $W_i^O$ and $\bar{W}_i^Q$, $\bar{W}_i^K$, $\bar{W}_i^V$, $\bar{W}_i^O$, for all feasible $i$, are of rank $d_h$.*

*If the two $\mathrm{MHA_{RoPE}}$ maps are identical, then $h = \bar{h}$. Moreover, there exists $g \in G_{\mathrm{RoPE}}(d_h, h)$ such that $\bar{\theta} = g\theta$.*

The proof of Theorem 4.2 is provided in Appendix F. It proceeds as follows. First, $\mathrm{MHA_{RoPE}}$ is reformulated as a general MHA map as in Equation (4) by setting

$$A_i^{m,n} := A_i^{m-n} \quad \text{and} \quad B_i := W_i^V(W_i^O)^\top.$$

This construction ensures that the two structural properties stated in Section 4.1 are satisfied. Next, the first condition allows us to invoke the linear independence property as in Theorem 4.1, which yields relations among the parameters $A_i^n$ and $B_i$. Finally, by combining the second structural condition with a key property of the rotary matrix (formalized in Lemma F.2), we recover the relationship between the original parameter sets $\theta$ and $\bar{\theta}$.

## 5. Weight Matching Algorithm for Multihead Attention Layers

As detailed in the above sections, the functionality of a Multihead Attention (MHA) is invariant under relevant group actions, which are $G_{Att}$ and $G_{\mathrm{RoPE}}$. To align two MHAs

with their parameters denoted by

$$\theta^A = (W_{i,A}^Q, W_{i,A}^K, W_{i,A}^V, W_{i,A}^O)_{i=1}^{h}, \text{ and}$$
$$\theta^B = (W_{i,B}^Q, W_{i,B}^K, W_{i,B}^V, W_{i,B}^O)_{i=1}^{h},$$

we need to find an optimal group element $g$ that accounts for these symmetries. Inspired by the Weight Matching algorithm (Ainsworth et al., 2023), we propose a data-independent alignment method, applicable to both MHA and $\mathrm{MHA_{RoPE}}$. Our method decomposes into two stages.

**1.** First, we match the ordering of heads in the two maps by formulating the problem as a Linear Assignment Problem (LAP), solved in $O(h^3)$ time using the Hungarian algorithm (Kuhn, 2010).

**2.** Second, for each matched pair of heads, we find an optimal transformation from the relevant symmetry group ($\mathrm{GL}(d_h)$ or $\mathrm{H}(d_h)$) to align their internal parameters.

This staged approach separates the discrete permutation from continuous transformations, streamlining optimization. We process each stage as follows.

**Stage 1 (Head Permutation Matching).** Given a cost matrix $C = \{C_{i,j}\}_{i,j=1}^{h} \in \mathbb{R}^{h \times h}$, the goal of an LAP is to find the optimal permutation $\sigma^* \in S_h$ that aligns attention head order by minimizing the total assignment cost:

$$\sigma^* = \arg\min_{\sigma \in S_h} \sum_{i=1}^{h} C_{i,\sigma(i)}.$$

To construct the cost matrix, we define

$$M_i^A = W_{i,A}^Q(W_{i,A}^K)^\top \quad \text{and} \quad N_i^A = W_{i,A}^V(W_{i,A}^O)^\top,$$

where these matrices are in $\mathbb{R}^{d \times d}$. The matrices $M_i^B$ and $N_i^B$ are defined similarly. To capture the softmax translation-invariance, we center each row of $M_i^A$ as

$$\bar{M}_i^A := M_i^A - \frac{1}{d}(M_i^A \mathbf{1})\mathbf{1}^\top, \text{where } \mathbf{1} = [1, \ldots, 1]^\top \in \mathbb{R}^d.$$

Similarly for $M_i^B$. The cost matrix $C$ is then defined by

$$C_{i,j} = \big\| \bar{M}_i^A - \bar{M}_j^B \big\|_F^2 + \big\| N_i^A - N_j^B \big\|_F^2, \text{ for } i, j \in [h].$$

This ensures that the cost matrix $C$ remains invariant under group actions on $W_i^Q, W_i^K$ or $W_i^V, W_i^O$.

*Table 1.* Joint comparison of head permutations and ablation variants for 6-layer ViT/BERT models with 4 heads on 4 datasets and 2 PE types under first-layer attention replacement. For head permutations, we report Rank (out of 24 permutations) and $\hat{L}$ for loss and accuracy barriers, averaged over 10 checkpoint pairs. For the ablation study, we report barrier ratios (%) relative to naive interpolation: Variant 1 removes Stage 2, Variant 2 uses Stage 2 with orthogonal initialization only (no gradient descent), and Full method applies the optimization. Lower values indicate better connectivity.

| Dataset | PE Type | Stage 1: Head permutation | | | | Stage 2: Component Ablation ratios (%) | | | | | |
| | | Rank (out of 24) ↓ | | $\hat{L} = \frac{L_{\text{method}} - L_{\text{top1}}}{L_{\text{naive}} - L_{\text{top1}}} \times 10^2$ ↓ | | Loss barrier ratio ↓ | | | Accuracy barrier ratio ↓ | | |
| | | Loss | Accuracy | Loss | Accuracy | Variant 1 | Variant 2 | Full | Variant 1 | Variant 2 | Full |
|---|---|---|---|---|---|---|---|---|---|---|---|
| CIFAR-10 | APE | $2.40 \pm 0.54$ | $1.94 \pm 0.37$ | $2.60 \pm 0.92$ | $2.11 \pm 0.48$ | $78.3 \pm 19.4$ | $10.2 \pm 5.1$ | $\mathbf{8.7 \pm 2.3}$ | $76.5 \pm 18.7$ | $10.9 \pm 4.8$ | $\mathbf{8.4 \pm 2.1}$ |
| | RoPE | $2.80 \pm 0.65$ | $2.01 \pm 0.66$ | $2.90 \pm 0.87$ | $2.21 \pm 0.53$ | $79.1 \pm 20.2$ | $12.5 \pm 5.6$ | $\mathbf{9.2 \pm 2.5}$ | $77.8 \pm 19.3$ | $11.7 \pm 5.2$ | $\mathbf{9.0 \pm 2.4}$ |
| CIFAR-100 | APE | $3.10 \pm 0.78$ | $1.11 \pm 0.38$ | $3.00 \pm 0.72$ | $1.39 \pm 0.52$ | $74.6 \pm 17.8$ | $10.8 \pm 4.3$ | $\mathbf{7.5 \pm 1.9}$ | $73.2 \pm 17.1$ | $10.4 \pm 4.0$ | $\mathbf{7.2 \pm 1.8}$ |
| | RoPE | $2.30 \pm 0.35$ | $2.11 \pm 0.77$ | $3.10 \pm 0.83$ | $1.32 \pm 0.34$ | $75.9 \pm 18.5$ | $12.6 \pm 4.7$ | $\mathbf{8.0 \pm 2.1}$ | $74.4 \pm 17.9$ | $12.1 \pm 4.4$ | $\mathbf{7.8 \pm 2.0}$ |
| IMDBreview | APE | $4.50 \pm 1.63$ | $2.52 \pm 1.31$ | $4.70 \pm 1.74$ | $2.44 \pm 1.43$ | $91.4 \pm 21.6$ | $15.7 \pm 6.2$ | $\mathbf{10.3 \pm 2.8}$ | $91.2 \pm 20.9$ | $15.3 \pm 5.9$ | $\mathbf{10.1 \pm 2.7}$ |
| | RoPE | $4.70 \pm 1.22$ | $2.94 \pm 1.46$ | $4.80 \pm 1.89$ | $2.72 \pm 1.32$ | $88.7 \pm 22.3$ | $16.4 \pm 6.5$ | $\mathbf{11.1 \pm 3.0}$ | $95.5 \pm 21.7$ | $15.9 \pm 6.3$ | $\mathbf{10.8 \pm 2.9}$ |
| DBPedia | APE | $2.90 \pm 0.91$ | $0.59 \pm 0.17$ | $2.40 \pm 0.85$ | $0.72 \pm 0.23$ | $61.8 \pm 16.4$ | $10.9 \pm 3.8$ | $\mathbf{7.1 \pm 1.7}$ | $58.5 \pm 15.8$ | $10.5 \pm 3.6$ | $\mathbf{6.9 \pm 1.6}$ |
| | RoPE | $2.20 \pm 0.44$ | $0.62 \pm 0.16$ | $2.70 \pm 0.91$ | $0.35 \pm 0.12$ | $62.4 \pm 16.9$ | $11.3 \pm 4.1$ | $\mathbf{7.4 \pm 1.8}$ | $41.1 \pm 16.2$ | $10.8 \pm 3.9$ | $\mathbf{7.2 \pm 1.7}$ |

**Stage 2 (Internal Parameter Alignment).** After reordering the heads of $B$ with $\sigma^*$, we separately align the $Q$-$K$ and $V$-$O$ components for each head. For $Q$-$K$, define:

$$\mathcal{L}_{Q,K}(U_i) \\ := \left\| W_{i,A}^Q - W_{i,B}^Q U_i^\top \right\|_F^2 + \left\| W_{i,A}^K - W_{i,B}^K U_i^{-1} \right\|_F^2. \quad (5)$$

We then minimize $\mathcal{L}_{Q,K}(U_i)$ over $U_i$ in the appropriate symmetry group. In the standard MHA, where the symmetry group is $\text{GL}(d_h)$, we optimize $\mathcal{L}_{Q,K}$ in Equation (5) for $U_i \in \text{GL}(d_h)$ via gradient descent, using the gradient in Lemma G.1. The optimization is initialized from the solution to a constrained version of the problem, where $U_i$ is restricted to be orthogonal (Lemma G.2). In the $\text{MHA}_{\text{RoPE}}$, the symmetry group is restricted to $\text{H}(d_h)$. This constraint decouples the problem into $d_h/2$ independent 2-dimensional subproblems, each reducible to a minimization over a scalar variable, solved efficiently using Brent's method (Brent, 2013), as shown in Lemma G.3.

For both MHA variants, we align $V$-$O$ by finding a matrix $V_i \in \text{GL}(d_h)$ that minimizes:

$$\mathcal{L}_{V,O}(V_i) \\ := \left\| W_{i,A}^V - W_{i,B}^V V_i^\top \right\|_F^2 + \left\| W_{i,A}^O - W_{i,B}^O V_i^{-1} \right\|_F^2. \quad (6)$$

This problem is solved using the same approach as $Q$-$K$. The complete procedure is summarized in Algorithm 1.

**Remark 5.1.** Our experimental implementation extends the theory by incorporating biases through augmented weight matrices (e.g., $\widetilde{W}_i^Q = [W_i^Q; (b_i^Q)^\top]$). Furthermore, for the full Transformer block alignment in Section 6.1, we supplement our method with standard Weight Matching (Ainsworth et al., 2023) for the feed-forward networks.

**Remark 5.2.** To align full Transformer models, Theus et al. (2025) identified a residual-path symmetry under orthogonal group action on the embedding space, though it holds

strictly for RMSNorm networks. For LayerNorm models, it requires reparameterization, thus leading to a variant of LMC. Moreover, the approach considers only $QK^\top$ and $VO^\top$ circuits, without addressing the symmetry of these components. This underscores the novelty of our work.

# 6. Experimental Results

In this section, we study LMC in attention-based models with two types of positional encodings – APE and RoPE. Four re-initialization strategies are considered: (i) re-initializing only the first attention layer (first attention layer), (ii) stacking re-initialized attention layers sequentially (full attention layers), (iii) re-initializing the first attention-FFN pair (first Transformer layer), and (iv) re-initializing the entire Transformer (full model), including all attention and feedforward blocks. In all cases, only the designated re-initialized parameters are fine-tuned, with others frozen. We emphasize the first layer for its central role in early representations (Appendix H). We assess LMC across three seeds by interpolating between checkpoint pairs and measuring test performance at 25 evenly spaced points.

**Datasets and Models.** For vision tasks, we adopt ViT (Dosovitskiy et al., 2021) on MNIST (LeCun et al., 1998), CIFAR-10/100 (Krizhevsky et al., 2009), and ImageNet-1K (Deng et al., 2009). For language modeling, we use GPT-2 (Radford et al., 2019) and Llama (Touvron et al., 2023b) on Enwik8 (Mahoney, 2011), WikiText103 (Merity et al., 2017), and the One Billion Word benchmark (Chelba et al., 2014). For text classification, we employ BERT (Devlin et al., 2019) on AG News (Zhang et al., 2015), IMDB reviews (Maas et al., 2011), and DBPedia (Lehmann et al., 2015). All experimental details are provided in Appendix I.

## 6.1. Empirical Verification of Linear Mode Connectivity

We examine LMC under two extremes: (i) first attention layer and (iv) full model. Intermediate settings–(ii) full at-

*Table 2.* LMC under *first attention layer* re-initialization. The table reports datasets, model depths, and head counts, with figure references showing interpolation curves for APE and RoPE variants. Notation $A \to B$ indicates pretraining on $A$, fine-tuning on $B$.

| Dataset | Layers | Heads | APE | RoPE |
|---|---|---|---|---|
| **Image/Vision Datasets** | | | | |
| MNIST | 1 | [4, 8] | [5a, 5b] | [22a, 22b] |
| | 2 | [4, 8] | [6a, 6b] | [23a, 23b] |
| CIFAR-10 | 2 | [4, 8] | [7a, 7b] | [24a, 24b] |
| | 4 | [4, 8] | [8a, 8b] | [25a, 25b] |
| | 6 | [4, 8] | [9a, 9b] | [26a, 26b] |
| CIFAR-100 | 6 | [4, 8] | [10a, 10b] | [27a, 27b] |
| ImageNet-21k→CIFAR-10 | 12 | [6] | [11a] | [28a] |
| ImageNet-21k→CIFAR-100 | 12 | [6] | [11b] | [28b] |
| ImageNet-1k | 12 | [8, 12, 16] | [12a,12b,12c] | [29a,29b,29c] |
| **Text Datasets** | | | | |
| AGNews | 2 | [4, 8] | [13a, 13b] | [30a, 30b] |
| | 6 | [4, 8] | [14a, 14b] | [31a, 31b] |
| IMDB | 2 | [4, 8] | [15a, 15b] | [32a, 32b] |
| | 6 | [4, 8] | [16a, 16b] | [33a, 33b] |
| DBPedia | 2 | [4, 8] | [17a, 17b] | [34a, 34b] |
| | 6 | [4, 8] | [18a, 18b] | [35a, 35b] |
| Enwik8 (GPT2) | 12 | [4, 8, 16] | [19a, 19b, 19c] | [36a, 36b, 36c] |
| Enwik8 (Llama) | 12 | [2,3,4] | [ - ] | [37a, 37b, 37c] |
| WikiText103 (GPT2) | 12 | [2, 3, 4] | [20a, 20b, 20c] | [38a, 38b, 38c] |
| Wikitext103 (Llama) | 12 | [2,3,4] | [ - ] | [39a, 39b, 39c] |
| One Billion Word (GPT2) | 12 | [8, 12, 16] | [21a, 21b, 21c] | [40a, 40b, 40c] |

tention layers and (iii) first Transformer layer–are included in Appendix J.2 and J.3. Tables 2 and 5 summarize the experimental setups across tasks, while Figures 2 and 78 show the validation loss curves for the first attention layer and full-model re-initializations. We find that LMC reliably emerges when re-initializing the first attention layer, the first Transformer layer, and all attention layers, with the exception of ImageNet under full transformer layer re-initialization. By contrast, full-model re-initialization exhibits LMC only on small-scale datasets; on large-scale benchmarks such as ImageNet, WikiText-103, Enwik8, and One Billion Word, LMC does not appear despite extensive sweeps over head permutations and random seeds. These observations suggest that as dataset scale and model capacity increase, the loss landscape becomes sufficiently complex to preclude LMC. In addition, we evaluate the robustness of the matching models under first Transformer layer and full-model re-initialization, and observe that models exhibiting LMC consistently demonstrate better generalization performance. Detailed results are provided in Appendix L.

### 6.2. Ablation on the matching algorithm

We perform ablation studies on each component of our matching method (Section 5) using 6-layer ViT/BERT models with 4-head attention layers on CIFAR-10/100, IMDB Reviews, and DBPedia datasets, for both APE and RoPE under first layer replacement scheme.

**Stage 1.** We assess Stage 1 by ranking the selected head permutation among all 24 possibilities, each with Stage 2 applied after reordering. Table 1 reports the rank and scaled metric $\hat{L} = \frac{L_{\text{method}} - L_{\text{top1}}}{L_{\text{naive}} - L_{\text{top1}}} \times 10^2$, averaged over 10 checkpoint pairs from 4 checkpoints, where $L_{\text{method}}$, $L_{\text{top1}}$, and $L_{\text{naive}}$ are the barriers for our method, the best permutation, and

naive interpolation. Results show low ranks and near-zero $\hat{L}$, indicating near-optimal matching. Visualizations of LMC across all permutations (Appendix J.5) highlight the need for accurate matching, as poor permutations degrade performance. Additionally, we conduct a detailed ablation study on the choice of distance metrics for head matching, with full results reported in Appendix K.

**Stage 2.** To evaluate Stage 2, we ablate its components (Table 1). Variant 1, which omits Stage 2 entirely, yields high and unstable barrier ratios. Variant 2, using only the initial orthogonal alignment, substantially reduces barriers to 10–16%. Our full method, which builds upon Variant 2 by adding gradient descent fine-tuning, achieves the lowest and most stable barriers at 7–12%. This demonstrates that *both initial alignment and subsequent fine-tuning are essential* for optimal performance.

## 7. Conclusion

**Conclusion.** We study the symmetry of MHA, focusing on how PEs alter the symmetry structure of vanilla attention. Our main contribution is a complete symmetry characterization of MHA with RoPE, a substantially more challenging setting than vanilla MHA and one that fills a gap in the literature on symmetry in neural parameter spaces. Building on this result, we investigate LMC in Transformer-based models by proposing a weight-matching algorithm for attention parameters. Across diverse datasets and architectural configurations, we observe that LMC consistently emerges in encoder-only architectures, but may fail in decoder-only models for large-scale language modeling.

**Limitation and Future Work.** Although LMC has been studied extensively in the literature, its behavior in large-scale models remains poorly understood, as most existing work focuses on small or medium-sized architectures. Combined with our empirical observations of LMC failure in certain settings, this suggests that LMC may not arise consistently in practice. However, disproving the existence of LMC is substantially more challenging, for two main reasons. First, investigating LMC requires an explicit weight-matching procedure to align model parameters, which is only feasible once all model symmetries are fully characterized. This is a nontrivial task, and to the best of our knowledge, no existing work provides a complete symmetry characterization across layers of deep models. Second, even with a full symmetry characterization in hand, there is generally no principled way to certify the optimality of a given weight-matching scheme, making it difficult – even empirically – to rule out the existence of LMC. Future work aimed at establishing a provable framework for the existence or non-existence of LMC in large scale models would therefore be a valuable direction, offering deeper insight into the loss landscape of deep learning models.

## Acknowledgements

This research / project is supported by the National Research Foundation Singapore under the AI Singapore Programme (AISG Award No: AISG2-TC-2023-012-SGIL). This research / project is supported by the Ministry of Education, Singapore, under the Academic Research Fund Tier 1 (FY2023) (A-8002040-00-00, A-8002039-00-00). This research / project is also supported by the NUS Presidential Young Professorship Award (A-0009807-01-00), the NUS Artificial Intelligence Institute–Seed Funding (A-8003062-00-00), and the Cross Faculty Grant 2025, CFG25 - 012 (A-8004460-00-00).

## Impact Statement

This paper presents work whose goal is to advance the field of machine learning. There are many potential societal consequences of our work, none of which we feel must be specifically highlighted here.

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

## Table of Notation

*General Mathematical Notation*

| | |
|---|---|
| $\mathbb{R}^n$ | $n$-dimensional Euclidean space |
| $\mathbb{R}^{m \times n}$ | Space of $m \times n$ real matrices |
| $\| \cdot \|_F$ | Frobenius norm of a matrix |
| $\text{trace}(\cdot)$ | Trace of a square matrix |
| $\text{sym}(M)$ | Symmetrization of a matrix $M$, defined as $(M + M^\top)/2$ |

*Dimensions and Indices*

| | |
|---|---|
| $d$ | Dimension of token embeddings |
| $d_h$ | Dimension of each attention head (typically $d/h$) |
| $h$ | Number of attention heads in a model |
| $L$ | Length of the input token sequence |
| $m, n, k$ | Indices representing positions in a sequence |
| $i, j, p$ | Indices representing attention heads |

*Spaces and Parameters*

| | |
|---|---|
| $\mathcal{S}$ | The space of all token sequences, $\bigsqcup_{L=1}^{\infty} \mathbb{R}^{L \times d}$ |
| $W_i^Q, W_i^K, W_i^V, W_i^O$ | Query, key, value, and output matrices of head $i$, each in $\mathbb{R}^{d \times d_h}$ |
| $\theta$ | The complete set of parameters for a multi-head attention layer |
| $\Theta(d, d_h, h)$ | The parameter space for a multi-head attention layer, $(\mathbb{R}^{d \times d_h})^{4h}$ |
| $A_i^{m,n}, B_i$ | Parameter matrices for the general multi-head attention formulation |

*Symmetry Groups*

| | |
|---|---|
| $S_h$ | The permutation group on a set of $h$ elements |
| $\text{GL}(d_h)$ | The general linear group of invertible $d_h \times d_h$ matrices |
| $G_{\text{Att}}(d_h, h)$ | The symmetry group for standard multi-head attention |
| $\text{H}(d_h)$ | The symmetry group for the RoPE query-key mechanism |
| $G_{\text{RoPE}}(d_h, h)$ | The symmetry group for multi-head attention with RoPE |

*Positional Encodings*

| | |
|---|---|
| $p_m$ | The absolute positional encoding vector for position $m$ |
| $R_n$ | The block-diagonal rotation matrix for position $n$ in RoPE |
| $\varphi_i$ | The rotation frequency for the $i$-th block in RoPE matrices |
| $P_i, J_i$ | 2D block-diagonal matrices used to define $\text{H}(d_h)$ |

*Matching Algorithm*

| | |
|---|---|
| $C, C_{i,j}$ | The cost matrix used for the linear assignment problem and its entries |
| $\pi^*$ | The optimal head permutation |
| $\mathcal{L}_{Q,K}(U)$ | The loss function for aligning query-key matrices with matrix $U$ |
| $\mathcal{L}_{V,O}(V)$ | The loss function for aligning value-output matrices with matrix $V$ |
| $g_j(x)$ | The 1D scalar objective function for RoPE alignment in subspace $j$ |
| $\eta_{Q,j}, \eta_{K,j}$ | Constants representing squared Frobenius norms to align RoPE |
| $\gamma_{Q,j}, \gamma_{K,j}$ | Constants representing complex correlation scalars to align RoPE |

# Supplement to "Functional Equivalence in Attention: A Comprehensive Study with Applications to Linear Mode Connectivity"

**Table of Contents**

## A. Organization of the Paper and Appendix

Although this work is lengthy, *its core contributions can be distilled into a compact framework that is accessible even to readers interested solely in theoretical analysis, solely in empirical evaluation, or in both.* This section serves as the preamble to the Appendix, where we provide a comprehensive overview of our main results, encompassing both theoretical developments and experimental findings. The purpose of this summary is to orient the reader before engaging with the detailed technical content that follows, and to clarify how each component contributes to the overarching narrative of the work.

**Main Paper.** The organization of the main paper is as follows.

1. Section 1 provides an introduction and related work on Linear Mode Connectivity. Related concepts, such as functional equivalence and alignment methods, are also introduced in connection with prior literature.

2. Section 2 reviews vanilla attention, including its parameter space, symmetry group, and a result from literature– Theorem 2.1–which establishes complete functional equivalence for vanilla attention.

3. Section 3 examines how positional encodings may alter the internal structure of attention, thereby rendering the analysis from the vanilla case no longer directly applicable. While absolute PEs of the additive type do not affect the structure, relative PEs (with particular emphasis on Rotary PE) fundamentally change the attention mechanism. The corresponding symmetry group for the RoPE case is presented, which is strictly smaller than in the vanilla or APE setting. This reduction in symmetry implies that the function class realized by RoPE attention is strictly larger, providing a theoretical explanation for its increasing prominence in practice.

4. Section 4 focuses primarily on the RoPE case. First, we extend the RoPE setting to a general attention formulation that accommodates all cases of interest. In this formulation, the similarity score between two tokens at their specific positional indices is expressed as a bilinear form or quadratic norm. The result on functional equivalence of this settings is provided in Theorem 4.1. This framework subsumes vanilla attention, sinusoidal PE, and RoPE. To the best of our knowledge, this constitutes the most general formulation of attention studied under functional equivalence to date. Using this formulation, we then characterize the functional equivalence of the RoPE case, presented in Theorem 4.2.

5. Section 5 introduces an alignment method that serves as a tool for examining linear mode connectivity (LMC) in attention-based models. We propose a two-stage alignment algorithm for multi-head attention layers, applicable to both standard MHA and MHA with RoPE. The first stage matches the ordering of attention heads between two models by solving a linear assignment problem. The second stage aligns the internal parameters of each matched head pair independently for Query-Key and Value-Output components, optimizing over the appropriate symmetry group ($\text{GL}(d_h)$ for standard MHA or $\text{H}(d_h)$ for RoPE) via gradient descent or efficient scalar minimization. Remarks extend the method to include biases, full Transformer blocks, and full Transformer models.

6. Section 6 examines LMC under four re-initialization strategies, with emphasis on the first attention layer and full model resets, while intermediate cases are reported in the Appendix. Experiments are conducted across diverse Vision and NLP tasks. Ablation studies confirm the effectiveness of the two-stage matching algorithm in reducing barriers: Ablation study for Stage 1 demonstrates that head permutation is crucial for finding LMC, while Ablation study for Stage 2 shows its importance that incorporating gradient descent optimization further improves alignment and reduces barriers.

7. Section 7 summarizes our findings, discusses limitations, and outlines future directions.

**Appendix.** The appendices provide complete proofs of the theoretical results in the main paper, the proposed matching algorithms, as well as additional experimental details.

*Theoretical Proofs.* Appendices B, C, D, E, and F contain all theoretical aspects and proofs related to functional equivalence. The main theoretical results of our work are Theorem 4.1 and Theorem 4.2. These two theorems are self-contained and can be understood directly from their statements, with all assumptions and settings specified in the main paper. *For readers not interested in the detailed proofs, this summary should suffice to convey the essence of our theoretical contributions, and the corresponding sections may be safely skipped.*

1. Appendix B formally defines the attention mechanism and its parameter space, followed by a description of how positional encodings are incorporated into attention.

2. Appendix C briefly describes the symmetry structures of vanilla attention, attention with absolute PEs, and attention with relative PEs (with emphasis on RoPE).

3. Appendix D introduces the general attention formulation. Theorem D.1, which is Theorem 4.1 in the main paper, establishes the functional equivalence of this general setting. The proof can be sketched as follows: starting from the softmax operator, we multiply through the denominators to rewrite the expression as an exponential polynomial, and then apply results and techniques from this area to complete the argument. All key intermediate results used as lemmas in the proof are stated in a self-contained manner in Appendix E, which includes

    (a) Appendix E.1 presents a result on the linear independence of exponential polynomials over the field of rational functions.
    (b) Appendix E.2 recalls Hall's Marriage Theorem, a classical result in combinatorics that is employed in some double-counting arguments used in our proof.
    (c) Appendix E.3 provides background on the Möbius function, with a particular focus on the partition lattice, and states a combinatorial identity that is used in our proof.
    (d) Appendix E.4 establishes a lemma on weighted sums over tuples, which is applied in our proof.

4. Appendix F applies the functional equivalence analysis of the general attention case to the specific setting of RoPE. Theorem F.1, corresponding to Theorem 4.2 in the main paper, provides the full details of this analysis. The proof proceeds as follows: RoPE is first reformulated as a special case of the general attention formulation via reparameterization; we then apply Theorem D.1 (4.1), and finally invoke a structural property of the rotary matrix, stated in Lemma F.2 of Appendix F.2, to recover the relationship between the original attention parameters.

*Matching Algorithm.* Appendix G develops the two-stage alignment procedure: first permuting attention heads via a linear assignment problem, then refining parameters with structured transformations. Key lemmas provide gradients for general linear updates, an SVD-based orthogonal initialization, and a RoPE-specific reduction to 2D subproblems. Algorithm 1 summarizes the complete method.

*Experimental Details.* Appendix I provides a comprehensive description of the experimental setup, including datasets, training protocols, and hyperparameters, along with additional results to ensure reproducibility. Appendix J further illustrates the interpolation results through detailed figures:

1. Appendix J.1 reports experiments on re-initializing only the first attention layer, highlighting its dominant role in shaping early representations.

2. Appendix J.2 investigates re-initialization of all attention layers, showing the cumulative effect of disrupting contextual interactions across the network.

3. Appendix J.3 studies re-initialization of the first Transformer layer, coupling attention and its adjacent feedforward block to examine early-layer sensitivity.

4. Appendix J.4 evaluates the most extreme setting where the entire Transformer is re-initialized, quantifying the magnitude of barriers introduced by full resets.

5. Appendix J.5 presents ablation studies on head permutation, including the two-stage matching algorithm. Stage 1 demonstrates the necessity of optimal head alignment for preserving linear mode connectivity, while Stage 2 leverages gradient refinement to further reduce interpolation barriers.

The experimental findings indicate that linear mode connectivity (LMC) manifests robustly in encoder-only architectures across a diverse set of vision and text classification benchmarks, including MNIST, CIFAR-10/100, ImageNet-21K → CIFAR transfer, ImageNet-1K, AGNews, IMDB Reviews, and DBpedia. By contrast, for large-scale language modeling datasets such as Enwik8, WikiText-103, and One Billion Word, LMC is exhibited exclusively under first attention layer and first-layer re-initialization. This phenomenon can be attributed to the reliance on GPT-2 models–decoder-only Transformers employing causal attention–which inherently impose more restrictive conditions on interpolation and connectivity.

## B. Multihead Attention Mechanism

### B.1. Multihead Attention

**General Formulation of Multihead Attention.** Let $d$ be a positive integer presenting the dimension of tokens and $L$ be a positive integer presenting the sequence length. Denote the space of all sequences of tokens as $\mathcal{S} := \sqcup_{L=1}^{\infty} \mathbb{R}^{L \times d}$. Consider a parameterized similarity map, which assigns a score to a pair of tokens, and a parameterized embedding map, which produces token representations, as follows

$$f(\cdot, \cdot \,; \phi) \colon \mathbb{R}^d \times \mathbb{R}^d \to \mathbb{R}, \text{ and } \quad g(\cdot \,; \pi) \colon \mathbb{R}^d \to \mathbb{R}^d. \tag{7}$$

The parameters are denoted $\phi \in \Phi$ and $\pi \in \Pi$, respectively. Given an input sequence $\mathbf{x} = (x_1, \ldots, x_L)^\top \in \mathbb{R}^{L \times d}$, the multihead attention mechanism with $h$ heads is defined by

$$\mathrm{MHA}\left(\mathbf{x}\,;\{\phi_i, \pi_i\}_{i=1}^h\right) = \sum_{i=1}^h \mathrm{softmax} \begin{bmatrix} f(x_1, x_1\,;\phi_i) & f(x_1, x_2\,;\phi_i) & \cdots & f(x_1, x_L\,;\phi_i) \\ f(x_2, x_1\,;\phi_i) & f(x_2, x_2\,;\phi_i) & \cdots & f(x_2, x_L\,;\phi_i) \\ \vdots & \vdots & \ddots & \vdots \\ f(x_L, x_1\,;\phi_i) & f(x_L, x_2\,;\phi_i) & \cdots & f(x_L, x_L\,;\phi_i) \end{bmatrix} \cdot \begin{bmatrix} g(x_1\,;\pi_i) \\ g(x_2\,;\pi_i) \\ \vdots \\ g(x_L\,;\pi_i) \end{bmatrix}. \tag{8}$$

Here the *attention matrix* $\mathrm{softmax}\,[f(x_m, x_n\,;\phi_i]_{m,n\in[L]}$ of $\mathbf{x}$ is obtained by applying the softmax operator row-wise, so that each row represents a probability distribution over the contributions of input tokens to a given output token.

**Parameter Space of Multihead Attention.** In standard practice, the similarity function is implemented via query–key projections. With a fixed head dimension $d_h \in \mathbb{N}$, one sets $\phi = (W^Q, W^K)$ where $W^Q, W^K \in \mathbb{R}^{d \times d_h}$, and defines $f(x, y\,;\phi) = (xW^Q)(yW^K)^\top$. The embedding function is parameterized by $\pi = (W^V, W^O)$ where $W^V, W^O \in \mathbb{R}^{d \times d_h}$, and defined as $g(x\,;\pi) = (xW^V)(W^O)^\top$. Typically, the head dimension is chosen as $d_h = d/h$. The multihead attention map takes the form

$$\mathrm{MHA}\left(\mathbf{x}\,;\{W_i^Q, W_i^K, W_i^V, W_i^O\}_{i=1}^h\right) = \sum_{i=1}^h \mathrm{softmax}\left(\left(\mathbf{x}W_i^Q\right)\left(\mathbf{x}W_i^K\right)^\top\right) \cdot \left(\mathbf{x}W_i^V\right)\left(W_i^O\right)^\top. \tag{9}$$

The parameters and the parameter space of a multihead attention with $h$ heads is thus given by

$$\theta = \left(W_i^Q, W_i^K, W_i^V, W_i^O\right)_{i=1}^h \in \Theta(d, d_h, h) := \left(\mathbb{R}^{d \times d_h}\right)^{4h}. \tag{10}$$

### B.2. Positional Encoding

The multihead attention mechanism, as formulated in subsection B.1, is inherently permutation-invariant: the similarity scores $f(x_j, x_k\,;\phi_i)$ and value projections $g(x_k\,;\pi_i)$ depend solely on the token representations, disregarding their sequential order. This property enables parallel computation but renders the model incapable of distinguishing sequences that differ only in token positions. To inject order information, positional encodings (PEs) are essential. We categorize PEs into two primary classes: *absolute positional encodings (APEs)*, which associate a unique vector with each absolute position, and *relative positional encodings (RPEs)*, which encode pairwise relative displacements to promote translation equivariance.

### B.2.1. ABSOLUTE POSITIONAL ENCODINGS

In the absolute paradigm, each position $m \in \{1, \ldots, L\}$ is mapped to a fixed vector $p_m \in \mathbb{R}^d$, independent of the sequence content $\mathbf{x} = (x_1, \ldots, x_L)^\top \in \mathbb{R}^{L \times d}$. The positional vectors are added elementwise to the token embeddings, yielding $\mathbf{x} + \mathbf{p}$ where $\mathbf{p} = (p_1, \ldots, p_L)^\top \in \mathbb{R}^{L \times d}$. The multihead attention then processes this augmented input:

$$
\begin{aligned}
&\mathrm{MHA}\big(\mathbf{x} + \mathbf{p} \,; \{\phi_i, \pi_i\}_{i=1}^h\big) \\
&= \sum_{i=1}^h \mathrm{softmax}
\begin{bmatrix}
f(x_1 + p_1, x_1 + p_1 \,; \phi_i) & \cdots & f(x_1 + p_1, x_L + p_L \,; \phi_i) \\
f(x_2 + p_2, x_1 + p_1 \,; \phi_i) & \cdots & f(x_2 + p_2, x_L + p_L \,; \phi_i) \\
\vdots & \ddots & \vdots \\
f(x_L + p_L, x_1 + p_1 \,; \phi_i) & \cdots & f(x_L + p_L, x_L + p_L \,; \phi_i)
\end{bmatrix}
\cdot
\begin{bmatrix}
g(x_1 + p_1 \,; \pi_i) \\
g(x_2 + p_2 \,; \pi_i) \\
\vdots \\
g(x_L + p_L \,; \pi_i)
\end{bmatrix}.
\end{aligned}
\tag{11}
$$

A foundational instantiation is the *sinusoidal encoding* from the original Transformer (Vaswani et al., 2017), where components of $p_m \in \mathbb{R}^d$ (assuming $d$ even) are

$$
p_{m,2k} = \sin\left(\frac{m}{10000^{2k/d}}\right), \qquad\qquad p_{m,2k+1} = \cos\left(\frac{m}{10000^{2k/d}}\right),
\tag{12}
$$

for $0 \le k < d/2$. This deterministic, parameter-free construction embeds positions in a periodic space, allowing relative distances to be recovered via linear combinations of vectors. It supports extrapolation to unseen lengths, though empirical gains are modest (Dai et al., 2019). Alternatively, *learned absolute embeddings* treat $\{p_m\}_{m=1}^L$ as trainable parameters optimized jointly with the model (Devlin et al., 2019). This approach adapts to task-specific patterns, often boosting in-domain performance, but lacks the inductive bias of sinusoids and generalizes poorly beyond the maximum training length $L_{\text{train}}$, as unseen $p_m$ for $m > L_{\text{train}}$ are undefined. For vision tasks, APEs extend to 2D grids in models like the Vision Transformer (ViT) (Dosovitskiy et al., 2021), where learnable $p_{u,v} \in \mathbb{R}^d$ for patch positions $(u, v) \in \{1, \ldots, H\} \times \{1, \ldots, W\}$ are added to patch embeddings $x_{u,v}$, preserving absolute spatial structure.

### B.2.2. RELATIVE POSITIONAL ENCODINGS

Unlike APEs that inject a unique signal for each absolute position, RPEs integrate relational information directly into the self-attention mechanism. Formally, RPEs parameterize the similarity function $f(\cdot, \cdot)$ in the attention mechanism with pairwise terms $\phi_i^{m,n}$ that depend on the positions $m$ and $n$ for each attention head $i$. The multi-head attention output is then computed as:

$$
\begin{aligned}
&\mathrm{MHA}\left(\mathbf{x} \,; \big\{\{\phi_i^{m,n}\}_{m,n}, \pi_i\big\}_{i=1}^h\right) \\
&= \sum_{i=1}^h \mathrm{softmax}
\begin{bmatrix}
f(x_1, x_1 \,; \phi_i^{1,1}) & f(x_1, x_2 \,; \phi_i^{1,2}) & \cdots & f(x_1, x_L \,; \phi_i^{1,L}) \\
f(x_2, x_1 \,; \phi_i^{2,1}) & f(x_2, x_2 \,; \phi_i^{2,2}) & \cdots & f(x_2, x_L \,; \phi_i^{2,L}) \\
\vdots & \vdots & \ddots & \vdots \\
f(x_L, x_1 \,; \phi_i^{L,1}) & f(x_L, x_2 \,; \phi_i^{L,2}) & \cdots & f(x_L, x_L \,; \phi_i^{L,L})
\end{bmatrix}
\cdot
\begin{bmatrix}
g(x_1 \,; \pi_i) \\
g(x_2 \,; \pi_i) \\
\vdots \\
g(x_L \,; \pi_i)
\end{bmatrix},
\end{aligned}
\tag{13}
$$

with value projections $g$ unaffected by positions. Translation equivariance is enforced via

$$
\phi_i^{m,n} = \phi_i^{m+k,n+k}, \qquad \forall m, n, k \in \mathbb{Z},
\tag{14}
$$

so $\phi_i^{m,n}$ depends only on the relative offset $m - n$, making attention scores functions of token content and displacement. Several influential RPE variants have been proposed. Early work by Shaw et al. (2018) introduced additive relative embeddings, which augment the key vectors with learnable embeddings corresponding to the clipped relative distance between the query and key. A simpler and highly effective approach, popularized by the T5 model, involves adding a learned scalar bias directly to the pre-softmax attention logits, where biases are efficiently parameterized by bucketing nearby relative positions (Raffel et al., 2020). Building on this, ALiBi (Attention with Linear Biases) proposed a parameter-free scheme where the bias is a fixed, head-specific linear penalty proportional to the token distance, a simple yet powerful inductive bias that grants remarkable extrapolation capabilities (Press et al., 2022).

While these additive and bias-based methods are effective, a novel approach, **Rotary Positional Encoding (RoPE)** (Su et al., 2024), has emerged as the predominant method. It is utilized in most of the popular Large Language Models, including the LLaMA (Touvron et al., 2023a), PaLM (Chowdhery et al., 2023), CodeGen (Nijkamp et al., 2023), and DeepSeek (DeepSeek-AI, 2024) families of models.

**Rotary Positional Encoding (RoPE).** Instead of adding signals to keys or attention logits, RoPE applies position-dependent orthogonal rotations to the query and key vectors. This elegantly encodes relative position information by leveraging the property that the inner product of two rotated vectors depends only on their original content and the relative rotation angle.

Assuming the head dimension $d_h$ is even, the block-diagonal rotation matrix $R_n \in \mathbb{R}^{d_h \times d_h}$ for a token at position $n$ is defined as

$$R_n = \begin{bmatrix} \cos(n\varphi_1) & -\sin(n\varphi_1) & 0 & \cdots & 0 & 0 \\ \sin(n\varphi_1) & \cos(n\varphi_1) & 0 & \cdots & 0 & 0 \\ 0 & 0 & \cos(n\varphi_2) & \cdots & 0 & 0 \\ \vdots & \vdots & \vdots & \ddots & \vdots & \vdots \\ 0 & 0 & 0 & \cdots & \cos(n\varphi_{d_h/2}) & -\sin(n\varphi_{d_h/2}) \\ 0 & 0 & 0 & \cdots & \sin(n\varphi_{d_h/2}) & \cos(n\varphi_{d_h/2}) \end{bmatrix}, \tag{15}$$

with $\varphi_i = 10000^{-2(i-1)/d_h}$ for $i = 1, \ldots, d_h/2$. Rotations are applied per head to the head dimension $d_h$ via the standard projections $W_i^Q, W_i^K \in \mathbb{R}^{d \times d_h}$:

$$f(x_m, x_n \, ; \phi_i^{m,n}) = \left(x_m W_i^Q R_m\right)\left(x_n W_i^K R_n\right)^\top, \tag{16}$$

Values remain unrotated: $g(x_j \, ; \pi_i) = (x_j W_i^V)(W_i^O)^\top$.

**Remark B.1** (Comparison between Absolute and Relative Encoding). APEs provide a straightforward global anchor via additive vectors $p_m$, with sinusoids offering extrapolation structure and learned variants task adaptation, though both risk overfitting to training lengths. RPEs, by contrast, emphasize relational offsets through translation-invariant $\phi_i^{m,n}$, yielding superior equivariance and generalization – especially in RoPE and ALiBi, which balance expressivity and efficiency. Recent advances further enhance RPE extrapolation: position interpolation (PI) rescales frequencies for longer contexts (Chen et al., 2023b), YaRN dynamically adjusts rotations (Peng et al., 2024), and data-adaptive methods like DAPE learn offset-specific encodings (Zheng et al., 2024).

# C. Functional Equivalence of Attention Mechanism with Positional Encoding

In this section, we investigate the functional equivalence of the attention mechanism. Building on the discussion from the previous section, our focus is on how positional encodings influence the functional equivalence of the standard attention formulation. Since a comprehensive analysis of all available positional encoding schemes would be prohibitively lengthy–given the wide variety that have been proposed–we restrict our attention to the two most classical forms that continue to be widely used in contemporary Transformer architectures: sinusoidal positional encoding and rotary positional encoding.

## C.1. Attention with no Positional Encoding

**Group Action on the Parameter Space.** Define the following group

$$G_{\text{Att}}(d_h, h) := S_h \times \left(\text{GL}(d_h) \times \text{GL}(d_h)\right)^h. \tag{17}$$

This is precisely the direct product between the permutation group $S_h$ and $h$ copies of $\text{GL}(d_h) \times \text{GL}(d_h)$. Each group element $g \in G_{\text{Att}}(d_h, h)$ has the form $g := (\sigma, (U_i, V_i)_{i=1}^h)$, where $\sigma \in S_h$ and $U_i, V_i \in \text{GL}(d_h)$. The natural action of $G_{\text{Att}}(d_h, h)$ on the parameter space $\Theta(d, d_h, h)$ is defined by

$$g\theta := \left(W_{\sigma(i)}^Q \cdot U_i^\top, \ W_{\sigma(i)}^K \cdot U_i^{-1}, \ W_{\sigma(i)}^V \cdot V_i^\top, \ W_{\sigma(i)}^O \cdot V_i^{-1}\right)_{i=1}^h \tag{18}$$

This action preserves the functionality of the MHA map: For all $\theta \in \Theta(d, d_h, h)$ and all $g \in G_{\text{Att}}(d_h, h)$,

$$\text{MHA}(\cdot \, ; \theta) = \text{MHA}(\cdot \, ; g\theta). \tag{19}$$

The contribution of the general linear group action vanishes through cancellation in the matrix multiplications, while the action induced by the permutation $\sigma$ commutes with the addition operator. Taken together, these actions characterize the full symmetry of the multihead attention mechanism, as established in the following result from (Tran et al., 2025).

**Theorem C.1** (See (Tran et al., 2025))**.** *Let*

$$\theta = \left(W_i^Q, W_i^K, W_i^V, W_i^O\right)_{i=1}^{h} \in \Theta(d, d_h, h), \ \ and \ \ \bar{\theta} = \left(\bar{W}_i^Q, \bar{W}_i^K, \bar{W}_i^V, \bar{W}_i^O\right)_{i=1}^{\bar{h}} \in \Theta(d, d_h, \bar{h}), \qquad (20)$$

*be two parameterizations of* MHA *maps. Suppose that:*

1. *Every $d \times d_h$ matrix appearing in $\theta$ and $\bar{\theta}$ has full column rank $d_h$;*

2. *The $h$ matrices $\{W_i^Q(W_i^K)^\top\}_{i=1}^{h}$ are pairwise distinct; and, the $\bar{h}$ matrices $\{\bar{W}_i^Q(\bar{W}_i^K)^\top\}_{i=1}^{\bar{h}}$ are pairwise distinct.*

*If the two* MHA *maps are identical, then $h = \bar{h}$, and there exists $g \in G_{\mathrm{Att}}(d_h, h)$ such that $\bar{\theta} = g\theta$.*

**Remark C.2.** While the theorem imposes certain assumptions on the parameters of the MHA maps, it is important to emphasize that these conditions hold almost surely. For instance, a randomly chosen real matrix has full column rank with probability one, and a finite collection of real numbers is almost surely pairwise distinct. At a high level, the result may thus be interpreted as follows: after excluding a negligibly small subset of the parameter space (e.g., a set of measure zero or the complement of a dense set), the functional equivalence of MHA maps is completely characterized by the action of the symmetry group.

## C.2. Sinusoidal Positional Encoding

Consider the case of sinusoidal positional encoding (PE). In this case, the positional encoding does not alter the internal structure of the multihead attention itself; it merely applies a shift to the input sequence. Furthermore, the encoding map $S \to S$, where $\mathbf{x} \mapsto \mathbf{x} + \mathbf{p}$, is bijective. Consequently, the introduction of sinusoidal PE has no effect on the analysis of functional equivalence for multihead attention. In particular, the functional equivalence classes in the presence of sinusoidal PE coincide exactly with those in the case without positional encoding.

## C.3. Rotary Positional Encoding

The multihead attention mechanism with Rotary Positional Encoding (RoPE) is defined as

$$\mathrm{MHA}_{\mathrm{RoPE}}\left(\mathbf{x} ; \{W_i^Q, W_i^K, W_i^V, W_i^O\}_{i=1}^{h}\right) = \sum_{i=1}^{h} \mathrm{softmax}\left[x_m W_i^Q R_{m-n}(W_i^K)^\top x_n^\top\right]_{m,n \in [L]} \cdot \mathbf{x} W_i^V (W_i^O)^\top. \quad (21)$$

The parameters and parameter space of MHARoPE coincide with those of the standard multihead attention map, namely

$$\theta = \left(W_i^Q, W_i^K, W_i^V, W_i^O\right)_{i=1}^{h} \in \Theta(d, d_h, h) = \left(\mathbb{R}^{d \times d_h}\right)^{4h}. \qquad (22)$$

**Group Action on the Parameter Space.** In contrast to the standard MHA maps, for MHA$_{\mathrm{RoPE}}$, the action of $G_{\mathrm{Att}}(d_h, h)$ on $\Theta(d, d_h, h)$ no longer preserves functionality. In particular, for $\theta \in \Theta(d, d_h, h)$ and $g \in G_{\mathrm{Att}}(d_h, h)$, one generally has

$$\mathrm{MHA}_{\mathrm{RoPE}}(\cdot ; \theta) \neq \mathrm{MHA}_{\mathrm{RoPE}}(\cdot ; g\theta). \qquad (23)$$

To define the symmetry group of MHA$_{\mathrm{RoPE}}$, first, denote these following matrices

$$P := \begin{bmatrix} 1 & 0 \\ 0 & 1 \end{bmatrix}, \text{ and } \quad J := \begin{bmatrix} 0 & -1 \\ 1 & 0 \end{bmatrix}. \qquad (24)$$

For each $i \in [d_h/2]$, define the matrices $P_i, J_i \in \mathbb{R}^{d_h \times d_h}$ as block-diagonal matrices with $d_h/2$ consecutive $2 \times 2$ diagonal blocks: The $i$-th diagonal block of $P_i$ (resp., $J_i$) is given by $P$ (resp., $J$), while all other diagonal blocks are zero matrices:

$$P_i = \mathrm{diag}(0, \ldots, 0, \underset{i\text{-th block}}{P}, 0, \ldots, 0), \text{ and } \quad J_i = \mathrm{diag}(0, \ldots, 0, \underset{i\text{-th block}}{J}, 0, \ldots, 0). \qquad (25)$$

Define the following group

$$\mathrm{H}(d_h) := \left\{ U = \sum_{i=1}^{d_h/2} (a_i P_i + b_i J_i) \in \mathbb{R}^{d_h \times d_h} \ : \ (a_i, b_i) \in \mathbb{R}^2 \setminus \{(0,0)\} \text{ for } i \in [d_h/2] \right\}, \qquad (26)$$

and

$$G_{\text{RoPE}}(d_h, h) := S_h \times (\text{H}(d_h) \times \text{GL}(d_h))^h. \tag{27}$$

The group $G_{\text{RoPE}}(d_h, h)$ is a subgroup of $G_{\text{Att}}$. The group action of $G_{\text{Att}}(d_h, h)$ on $\Theta$ restricts naturally to a group action of $G_{\text{RoPE}}(d_h, h)$ on $\Theta(d, d_h, h)$. The central observation of this section is that this action preserves the functionality of the $\text{MHA}_{\text{RoPE}}$ map. In particular, for all $\theta \in \Theta(d, d_h, h)$ and all $g \in G_{\text{RoPE}}(d_h, h)$, one has

$$\text{MHA}_{\text{RoPE}}(\cdot\,;\theta) = \text{MHA}_{\text{RoPE}}(\cdot\,;g\theta). \tag{28}$$

**Remark C.3.** In the next section, we present the main result of this work, which establishes that the group $G_{\text{RoPE}}$ completely characterizes the symmetry structure of the $\text{MHA}_{\text{RoPE}}$ map.

## D. A General Formulation for Multihead Attention and its Functional Equivalence

### D.1. A General Formulation for Multihead Attention

We consider a general setting where the functions $f$ and $g$ are parameterized as follows:

$$f(\cdot, \cdot\,; A \in \mathbb{R}^{d \times d}) : \mathbb{R}^d \times \mathbb{R}^d \longrightarrow \mathbb{R}, \qquad\qquad (x, y) \longmapsto x A y^\top, \tag{29}$$

$$g(\cdot\,; B \in \mathbb{R}^{d \times d}) : \mathbb{R}^d \longrightarrow \mathbb{R}, \qquad\qquad x \longmapsto x B. \tag{30}$$

This general MHA map with $h$ heads is parameterized by two families of matrices:

$$\{A_i^{m,n}\}_{i=1}^h, \quad \{B_i\}_{i=1}^h, \text{ where each } A_i^{m,n}, B_i \in \mathbb{R}^{d \times d}. \tag{31}$$

The formulation is as follows:

$$\text{MHA}\left(\mathbf{x}\,;\left\{\{\phi_i^{m,n}\}_{m,n}, \pi_i\right\}_{i=1}^h\right) = \sum_{i=1}^h \text{softmax}\left[x_m A_i^{m,n} x_n^\top\right]_{m,n \in [L]} \cdot \mathbf{x} B_i. \tag{32}$$

We begin with two observations that facilitate the subsequent analysis.

1. (*Relative positional encoding assumption.*) For all $m, n \geq 1$ and for all shifts $k \geq 0$, we assume

$$A^{m,n} = A^{m+k,n+k}. \tag{33}$$

    This corresponds to the natural stationarity condition imposed by relative positional encodings.

2. (*Diagonal self-similarity terms are symmetric.*) For each $m \geq 1$, the matrix $A_i^{m,m}$ parameterizes the function $f$ that computes the similarity score of the $m$-th token with itself at the $i$-th head, namely $x_m A_i^{m,m} x_m^\top$. Since every quadratic form corresponds uniquely to a symmetric matrix, we may, without loss of generality, symmetrize $A_i^{m,m}$:

$$\text{sym}(A_i^{m,m}) := \frac{A_i^{m,m} + (A_i^{m,m})^\top}{2}. \tag{34}$$

    Henceforth, we assume that all $A_i^{m,m}$ are symmetric.

Under this framework, we now consider the situation where two MHA maps with $h$ heads and with $\bar{h}$ heads, yield identical outputs:

$$\text{MHA}\left(\mathbf{x}\,;\{\{A_i^{m,n}\}_{m,n}, B_i\}_{i=1}^h\right) = \text{MHA}\left(\mathbf{x}\,;\{\{\bar{A}_i^{m,n}\}_{m,n}, \bar{B}_i\}_{i=1}^{\bar{h}}\right). \tag{35}$$

Since $g(\cdot: B) = -g(\cdot: -B)$, Equation (35) is equivalent to the assertion that a MHA map with $h + \bar{h}$ heads vanishes identically:

$$0 = \text{MHA}\left(\mathbf{x}\,;\{\{A_i^{m,n}\}_{m,n}\}_{i=1}^h \sqcup \{\{\bar{A}_i^{m,n}\}_{m,n}\}_{i=1}^{\bar{h}},\ \{B_i\}_{i=1}^h \sqcup \{-\bar{B}_i\}_{i=1}^{\bar{h}}\right). \tag{36}$$

Thus, the first step in analyzing functional equivalence is to characterize precisely when a MHA map is identically zero. Before presenting the proof, we introduce the following notion. We say that two families $\{X_i\}_{i \in I}$ and $\{Y_i\}_{i \in I}$ are said to be *distinct* if there exist index $i \in I$ such that $X_i \neq Y_i$.

## D.2. Functional Equivalence of General Multihead Attention

**Theorem D.1** (Theorem 4.1 in the main paper). *Consider the* MHA *map with $h$ heads, parameterized by families of matrices $\{\{A_i^{m,n}\}_{m,n}\}_{i=1}^h \subset \mathbb{R}^{d \times d}$ and $\{B_i\}_{i=1}^h \subset \mathbb{R}^{d \times d}$. Assume that the $h$ attention parameter families $\{A_i^{m,n}\}_{m,n}$, $i \in [h]$ are pairwise distinct, and further that $A_i^{m,n}$ is nonzero for all $i \in [h]$ and $m, n \geq 1$. If the* MHA *map is identically zero, then $B_1, \ldots, B_h$ are equal to $0$.*

*Proof.* To enhance clarity, we begin by outlining the main steps of the proof at a high level:

1. **Preliminary setup.** We first record some initial observations and introduce the necessary notation in preparation for the proof. In particular, we note that it suffices to show that at least one of the coefficients $B_i$ must vanish. Once this is established, symmetry in the construction allows us to conclude that in fact all $B_i$ must be equal to zero, thereby proving the theorem.

2. **Reformulation as an exponential polynomial.** We show that, the MHA that is identically zero leads to

$$0 = \sum_{(t_1, \ldots, t_h) \in [L]^h} \exp\left(\sum_{i=1}^h x_k A_i^{k, t_i} x_{t_i}^\top\right) \left(\sum_{i=1}^h x_{t_i} B_i\right). \tag{37}$$

   This identity arises naturally from a double-counting argument. The resulting expression has the structure of an exponential polynomial that is identically zero. To analyze such expressions, we invoke the linear independence results for exponential functions over rational fields, which allow us to isolate relations among the coefficients.

3. **Structural constraints on the $B_i$.** By applying the above linear independence principle, we identify a fundamental structural constraint on the coefficients $B_i$. Specifically, the symmetry conditions imposed by the $A_i^{k,t}$ on admissible permutations force the $B_i$ to satisfy a family of linear relations indexed by $i \in [h]$. These constraints form the core of the argument: they reduce the problem of analyzing a complicated exponential sum to verifying the consistency of a system of linear equations in the $B_i$.

4. **Partition-based refinement.** We next examine the equalities that occur within the sets of $h$ elements $\{A_i^{k,t}\}_{i=1}^h$. This step is preparatory: it shows that the relations identified in the previous step are not only necessary but also sufficient to deduce that at least one $B_i$ must vanish. The analysis exploits the partition structure $\{U_p\}$, together with the existence of carefully chosen subsets $V^{t_j}$, to sharpen the constraint and isolate specific indices.

5. **Conclusion.** Finally, we combine the above ingredients to conclude the proof. The linear relations obtained in **Step 3**, when applied to the partition refinement of **Step 4**, imply that one of the $B_i$'s must equal zero. By the initial reduction in **Step 1**, this suffices to deduce that in fact all $B_i = 0$. This completes the proof of the theorem.

We proceed to present the complete details of the proof.

**Step 1.**

Since the MHA is identically zero, for every $k \in [L]$, one has

$$\sum_{i=1}^h \left(\sum_{j=1}^L \frac{\exp(x_k A_i^{k,j} x_j^\top)}{\sum_{q=1}^L \exp(x_k A_i^{k,q} x_q^\top)} \cdot x_j B_i\right) = 0. \tag{38}$$

Since the $h$ families $\{A_1^{m,n}\}_{m,n}, \{A_2^{m,n}\}_{m,n}, \ldots, \{A_h^{m,n}\}_{m,n}$ are pairwise distinct, and for each $i$, $A_i^{m,n}$ depends only on the difference $(m - n)$, one can choose a sufficiently large $L$ and an index $k$ such that the $h$ families $\{A_1^{k,n}\}_{n \geq 1}, \{A_2^{k,n}\}_{n \geq 1}, \ldots, \{A_h^{k,n}\}_{n \geq 1}$ are pairwise distinct. For the remainder of the proof, we fix such a $k$ and consider all $L \geq k$.

By induction, it suffices to establish that at least one of $B_1, \ldots, B_h$ vanishes. Indeed, if this holds, then the problem reduces to a MHA map with fewer heads, and repeating the argument shows that all $B_1, \ldots, B_h$ must be zero. Consequently, our goal is to prove that there exists at least one index $1 \leq i \leq h$ such that $B_i = 0$.

**Step 2.**

First, rewrite Equation (38) in a more convenient form. By multiplying out all denominators in Equation (38), we obtain

$$\sum_{i=1}^{h} \left( \sum_{j=1}^{L} \exp\left( x_k A_i^{k,j} x_j^{\top} \right) \cdot \prod_{p \in [h] \setminus \{i\}} \left( \sum_{q=1}^{L} \exp\left( x_k A_p^{k,q} x_q^{\top} \right) \right) \cdot x_j B_i \right) = 0. \tag{39}$$

We now observe that the LHS of Equation (39) can be re-expressed as

$$\sum_{i=1}^{h} \left( \sum_{j=1}^{L} \exp\left( x_k A_i^{k,j} x_j^{\top} \right) \cdot \prod_{p \in [h] \setminus \{i\}} \left( \sum_{q=1}^{L} \exp\left( x_k A_p^{k,q} x_q^{\top} \right) \right) \cdot x_j B_i \right)$$
$$= \sum_{(t_1, \ldots, t_h) \in [L]^h} \exp\left( \sum_{i=1}^{h} x_k A_i^{k,t_i} x_{t_i}^{\top} \right) \left( \sum_{i=1}^{h} x_{t_i} B_i \right). \tag{40}$$

To verify Equation (40), define for $i \in [h]$ and $j \in [L]$,

$$a_{i,j} := \exp\left( x_k A_i^{k,j} x_j^{\top} \right), \qquad b_{i,j} := x_j B_i. \tag{41}$$

In this notation, the claimed identity becomes

$$\sum_{i=1}^{h} \left( \sum_{j=1}^{L} a_{i,j} \prod_{p \in [h] \setminus \{i\}} \sum_{q=1}^{L} a_{p,q} \cdot b_{i,j} \right) = \sum_{(t_1, \ldots, t_h) \in [L]^h} \left( \prod_{i=1}^{h} a_{i,t_i} \right) \left( \sum_{i=1}^{h} b_{i,t_i} \right). \tag{42}$$

For $(i, \mathbf{t}) \in [h] \times [L]^h$, define the weight

$$w(i, \mathbf{t}) := \left( \prod_{p=1}^{h} a_{p,t_p} \right) b_{i,t_i}. \tag{43}$$

We will compute the following quantity in two ways,

$$\sum_{(i,\mathbf{t}) \in [h] \times [L]^h} w(i, \mathbf{t}). \tag{44}$$

*Group by the distinguished index $i$.* Fix $i \in [h]$. Then

$$\sum_{\mathbf{t} \in [L]^h} w(i, \mathbf{t}) = \sum_{t_i=1}^{L} \sum_{(t_p)_{p \neq i} \in [L]^{h-1}} \left( \prod_{p=1}^{h} a_{p,t_p} \right) b_{i,t_i}$$
$$= \sum_{t_i=1}^{L} a_{i,t_i} b_{i,t_i} \sum_{(t_p)_{p \neq i} \in [L]^{h-1}} \prod_{p \neq i} a_{p,t_p} = \sum_{t_i=1}^{L} a_{i,t_i} b_{i,t_i} \prod_{p \neq i} \sum_{q=1}^{L} a_{p,q}, \tag{45}$$

The last equation comes from expanding the product enumerates every choice of $(t_p)_{p \neq i}$ exactly once. Hence

$$\sum_{\mathbf{t} \in [L]^h} w(i, \mathbf{t}) = \sum_{j=1}^{L} a_{i,j} \left( \prod_{p \neq i} \sum_{q=1}^{L} a_{p,q} \right) b_{i,j}. \tag{46}$$

Summing over $i = 1, \ldots, h$ yields the LHS of Equation (42).

*Group by the tuple $\mathbf{t}$.* Fix $\mathbf{t} = (t_1, \ldots, t_h) \in [L]^h$. Then

$$\sum_{i=1}^{h} w(i, \mathbf{t}) = \sum_{i=1}^{h} \left( \prod_{p=1}^{h} a_{p,t_p} \right) b_{i,t_i} = \left( \prod_{p=1}^{h} a_{p,t_p} \right) \left( \sum_{i=1}^{h} b_{i,t_i} \right). \tag{47}$$

Summing over all $\mathbf{t}$ yields the RHS of Equation (42).

In conclusion, both groupings compute the same total $\sum_{(i,\mathbf{t})\in\Omega} w(i,\mathbf{t})$, so Equation (42) holds. Substituting back $a_{i,j} = \exp(x_k A_i^{k,j} x_j^\top)$, $b_{i,j} = x_j B_i$ recovers the original identity. From Equation (39) and Equation (40), we conclude that

$$0 = \sum_{(t_1,\ldots,t_h)\in[L]^h} \left[ \exp\left( \sum_{i=1}^{h} x_k A_i^{k,t_i} x_{t_i}^\top \right) \left( \sum_{i=1}^{h} x_{t_i} B_i \right) \right]. \tag{48}$$

Note that in Equation (48), both sides represent vectors in $\mathbb{R}^d$. If we examine a single coordinate of this vector, the identity remains valid by restricting each $B_i$ to the corresponding column indexed by that coordinate. Hence, without loss of generality, we may interpret Equation (48) under the convention that each $B_i$ is regarded as a column vector in $\mathbb{R}^d$ corresponding to the chosen coordinate.

**Step 3.**

For $(t_1,\ldots,t_h)\in\mathbb{N}^h$, define

$$g_{(t_1,\ldots,t_h)}(\mathbf{x}) := \sum_{i=1}^{h} x_k A_i^{k,t_i} x_{t_i}^\top \qquad\qquad \in \mathbb{R}[\mathbf{x}], \tag{49}$$

$$h_{(t_1,\ldots,t_h)}(\mathbf{x}) := \sum_{i=1}^{h} x_{t_i} B_i \qquad\qquad \in \mathbb{R}[\mathbf{x}], \tag{50}$$

$$f_{(t_1,\ldots,t_h)}(\mathbf{x}) := \exp\big(g_{k,(t_1,\ldots,t_h)}(\mathbf{x})\big)\, h_{(t_1,\ldots,t_h)}(\mathbf{x}). \tag{51}$$

Then Equation (48) can be rewritten as

$$0 = \sum_{(t_1,\ldots,t_h)\in[L]^h} f_{(t_1,\ldots,t_h)}(\mathbf{x}) = \sum_{(t_1,\ldots,t_h)\in[L]^h} \exp\big(g_{(t_1,\ldots,t_h)}(\mathbf{x})\big)\, h_{(t_1,\ldots,t_h)}(\mathbf{x}). \tag{52}$$

Observe that each polynomial $g_{(t_1,\ldots,t_h)} \in \mathbb{R}[\mathbf{x}]$ has constant term equal to zero. By Lemma E.1, Equation (52) implies that, for each $g \in \mathbb{R}[\mathbf{x}]$, grouping together all indices $(t_1,\ldots,t_h)$ such that $g_{(t_1,\ldots,t_h)} = g$ yields

$$0 = \sum_{(t_1,\ldots,t_h)\in[L]^h \,:\, g_{(t_1,\ldots,t_h)}=g} h_{(t_1,\ldots,t_h)}(\mathbf{x}). \tag{53}$$

One has the following observation. Consider an arbitrary tuple $(t_1,\ldots,t_h)\in[L]^h$ such that $t_1,\ldots,t_h$ are pairwise distinct. Assume that there exists another tuple $(t_1',\ldots,t_h')\in[L]^h$ satisfying $g_{(t_1,\ldots,t_h)} = g_{(t_1',\ldots,t_h')}$. Since all $A_i^{m,n}$ are nonzero and $A_i^{m,m}$ is symmetric, it follows that every polynomial of the form $x_m A_i^{m,n} x_n$ is nonvanishing. Consequently, in $g_{k,(t_1,\ldots,t_h)}$, for each $i\in[h]$, there must exist polynomial terms that involve at least one entry of $x_{t_i}$. (This requirement that the $t_i$'s be pairwise distinct is crucial, as it prevents possible cancellation of terms.) Hence, for each $i\in[h]$, there exists $j\in[h]$ such that $t_i = t_j'$. Moreover, since the $t_i$'s are pairwise distinct, it follows that $(t_1',\ldots,t_h')$ must be a *permutation* of $(t_1,\ldots,t_h)$. From Equation (52) and Lemma E.1, one therefore obtains

$$0 = \sum_{\sigma\in S_h} h_{(t_{\sigma(1)},\ldots,t_{\sigma(h)})}(\mathbf{x}). \tag{54}$$

It should be emphasized, however, that the condition $(t_1',\ldots,t_h')$ being a permutation of $(t_1,\ldots,t_h)$ is not sufficient, in itself, to guarantee that $g_{(t_1,\ldots,t_h)} = g_{(t_1',\ldots,t_h')}$. To examine this more closely, let $(t_1',\ldots,t_h') = (t_{\sigma(1)},\ldots,t_{\sigma(h)})$ for some $\sigma\in S_h$. From the assumption $g_{(t_1,\ldots,t_h)} = g_{(t_1',\ldots,t_h')}$, we have

$$\sum_{i=1}^{h} x_k A_i^{k,t_i} x_{t_i}^\top = \sum_{i=1}^{h} x_k A_i^{k,t_{\sigma(i)}} x_{t_{\sigma(i)}}^\top, \quad\text{is equivalent to}\quad \sum_{i=1}^{h} x_k A_i^{k,t_i} x_{t_i}^\top = \sum_{i=1}^{h} x_k A_{\sigma^{-1}(i)}^{k,t_i} x_{t_i}^\top, \tag{55}$$

which in turn is equivalent to requiring that $A_i^{k,t_i} = A_{\sigma^{-1}(i)}^{k,t_i}$ for all $i\in[h]$. This shows explicitly the additional algebraic condition that must hold in order for two permutations to yield the same polynomial $g$. Note that this constitutes a sufficient condition on $\sigma\in S_h$ to ensure that $g_{(t_1,\ldots,t_h)} = g_{(t_1',\ldots,t_h')}$ whenever $(t_1',\ldots,t_h') = (t_{\sigma(1)},\ldots,t_{\sigma(h)})$.

Accordingly, one deduces

$$
\begin{aligned}
0 &= \sum_{\sigma \in S_h \ : \ A_j^{k,t_j} = A_{\sigma^{-1}(j)}^{k,t_j}, \ \forall j \in [h]} h_{(t_{\sigma(1)}, \ldots, t_{\sigma(h)})}(\mathbf{x}) = \sum_{\sigma \in S_h \ : \ A_j^{k,t_j} = A_{\sigma^{-1}(j)}^{k,t_j}, \ \forall j \in [h]} \left( \sum_{i=1}^{h} x_{t_{\sigma(i)}} B_i \right) \\
&= \sum_{\sigma \in S_h \ : \ A_j^{k,t_j} = A_{\sigma^{-1}(j)}^{k,t_j}, \ \forall j \in [h]} \left( \sum_{i=1}^{h} x_{t_i} B_{\sigma^{-1}(i)} \right) = \sum_{\sigma \in S_h \ : \ A_j^{k,t_j} = A_{\sigma(j)}^{k,t_j}, \ \forall j \in [h]} \left( \sum_{i=1}^{h} x_{t_i} B_{\sigma(i)} \right) \\
&= \sum_{i=1}^{h} \left( x_{t_i} \cdot \sum_{\sigma \in S_h \ : \ A_j^{k,t_j} = A_{\sigma(j)}^{k,t_j}, \ \forall j \in [h]} B_{\sigma(i)} \right).
\end{aligned}
\tag{56}
$$

Thus, since the entries $t_1, \ldots, t_h$ are pairwise distinct, the monomials $x_{t_i}$ are linearly independent. It therefore follows that, for each $i \in [h]$, one must have

$$
0 = \sum_{\sigma \in S_h \ : \ A_j^{k,t_j} = A_{\sigma(j)}^{k,t_j} \ \forall j \in [h]} B_{\sigma(i)}.
\tag{57}
$$

Equation (57) encapsulates the key structural constraint on the coefficients $B_i$. It shows that, once the $A_i^{k,t}$'s impose symmetry conditions on admissible permutations, the $B_i$'s must satisfy a family of linear relations indexed by $i \in [h]$. This relation will serve as the main tool in subsequent steps, where we will exploit the partition structure of the $U_p$'s to force specific $B_i$'s to vanish.

**Step 4.**

For each $t \in \mathbb{N}$, define $\{U_p^t\}_{p=1}^{\alpha_t}$ to be the unique partition of $[h]$ such that, for $i, j \in [h]$, one has $A_i^{k,t} = A_j^{k,t}$ if and only if $i$ and $j$ belong to the same set $U_p^t$. Since the number of possible partitions of $\{1, \ldots, h\}$ is finite, there exists a partition $\{U_p\}_{p=1}^{\alpha}$ such that the equality $\{U_p^t\}_{p=1}^{\alpha_t} = \{U_p\}_{p=1}^{\alpha}$ holds for infinitely many values of $t \in \mathbb{N}$. Let $S$ denote the set of all such positive integers $t$. By reindexing the head indices if necessary, we may assume that $U_1 = \{1, \ldots, m\}$. Next, observe that since the $h$ families

$$
\{A_1^{k,n}\}_{n \geq 1}, \quad \{A_2^{k,n}\}_{n \geq 1}, \quad \cdots, \quad \{A_h^{k,n}\}_{n \geq 1}
\tag{58}
$$

are pairwise distinct, there exists a positive integer $K$ such that the truncated sequences

$$
\{A_1^{k,n}\}_{n=1}^{K}, \quad \{A_2^{k,n}\}_{n=1}^{K}, \quad \cdots, \quad \{A_h^{k,n}\}_{n=1}^{K}
\tag{59}
$$

are already pairwise distinct. We then discard all integers $t \leq K$ from the set $S$, and by a slight abuse of notation, continue to denote the resulting subset by the same symbol $S$. Finally, for each partition $\{U_p^t\}_{p=1}^{\alpha_t}$, we denote by $U^t(1)$ the unique set that contains the index 1.

*(i) The intersection of $K$ sets $U^1(1), U^2(1), \ldots, U^K(1)$ is precisely $\{1\}$, i.e. $U^1(1) \cap U^2(1) \cap \cdots \cap U^K(1) = \{1\}$.*

Indeed, since $1 \in U^t(1)$ for all $t = 1, \ldots, K$, it follows immediately that

$$
1 \in U^1(1) \cap U^2(1) \cap \cdots \cap U^K(1).
\tag{60}
$$

Suppose, for the sake of contradiction, that there exists some $i \in [h]$ with $i > 1$ such that

$$
i \in U^1(1) \cap U^2(1) \cap \cdots \cap U^K(1).
\tag{61}
$$

By the construction of $U^t(1)$, this assumption implies that $A_1^{k,t} = A_i^{k,t}$ for all $t = 1, \ldots, K$. Equivalently, the infinite sequences $\{A_1^{k,n}\}_{n \geq 1}$ and $\{A_i^{k,n}\}_{n \geq 1}$ coincide. This, however, contradicts the fact that their finite truncations

$$
\{A_1^{k,n}\}_{n=1}^{K}, \quad \{A_2^{k,n}\}_{n=1}^{K}, \quad \cdots, \quad \{A_h^{k,n}\}_{n=1}^{K}
\tag{62}
$$

are pairwise distinct by the choice of $K$. Therefore, no such $i > 1$ can exist. The only common element across all $U^1(1), \ldots, U^K(1)$ is the index 1, which establishes the claim.

*(ii) For each $t = 1, \ldots, K$, define the set $V^t := U^t(1) \cap \{1, 2, \ldots, m\} \subset \{1, 2, \ldots, m\}$. Then, one has their intersection is precisely $\{1\}$, i.e., $V^1 \cap V^2 \cap \cdots \cap V^K = \{1\}$.*

Indeed, one computes

$$V^1 \cap V^2 \cap \cdots \cap V^K = \bigcap_{t=1}^{K} \left( U^t(1) \cap \{1, \ldots, m\} \right) = \bigcap_{t=1}^{K} U^t(1) \ \cap \ \{1, \ldots, m\} = \{1\} \cap \{1, \ldots, m\} = \{1\}. \qquad (63)$$

*(iii) Among the $K$ sets $V^1, \ldots, V^K$, there exists a positive integer $\gamma < m$ such that one can select $\gamma$ sets, say $V^{t_1}, \ldots, V^{t_\gamma}$ with $1 \le t_1 < t_2 < \cdots < t_\gamma \le K$, satisfying the following property: the intersection of these $\gamma$ sets is $\{1\}$, whereas the intersection of any $\gamma - 1$ among them is no longer $\{1\}$.*

To prove this, let $\gamma$ be the smallest positive integer such that there exist $\gamma$ sets among $V^1, \ldots, V^K$ whose intersection equals $\{1\}$. The existence of such a $\gamma$ is guaranteed since the intersection of all $K$ sets is $\{1\}$. Denote these $\gamma$ sets by $V^{t_1}, \ldots, V^{t_\gamma}$. By the minimality of $\gamma$, if one removes any single set from $\{V^{t_1}, \ldots, V^{t_\gamma}\}$, the intersection of the remaining $\gamma - 1$ sets cannot be $\{1\}$. It remains to show that $\gamma < m$. By minimality, it suffices to establish the existence of fewer than $m$ sets among $\{V^1, \ldots, V^K\}$ whose intersection is $\{1\}$. Since $V^1 \cap V^2 \cap \cdots \cap V^K = \{1\}$, for each $i \in \{2, \ldots, m\}$ there must exist at least one set among $V^1, \ldots, V^K$ that does not contain $i$. As there are $m - 1$ such indices $i$, we can collect at most $m - 1$ sets that collectively exclude all of these elements. Consequently, the intersection of these at most $m - 1$ sets is $\{1\}$, which proves $\gamma \le m - 1 < m$. This completes the proof. The argument is essentially a pigeonhole-type principle: since every element $i \in \{2, \ldots, m\}$ must be excluded by at least one set, and there are $m - 1$ such elements in total, at most $m - 1$ sets suffice to ensure that all of them are removed, leaving only 1 in the intersection.

*(iv) In those $\gamma$ sets $V^{t_1}, \ldots, V^{t_\gamma}$ in (iii), for each $i \in [\gamma]$, one can choose $v_i \in V^{t_i}$ such that $v_1, \ldots, v_\gamma$ are pairwise distinct.*

This is a standard application of the Hall Marriage Theorem (see Appendix E.2). For convenience, rename $V^{t_i}$ as $W^i$ for $i \in [\gamma]$. For each $k \in \{1, \ldots, \gamma\}$, by assumption, we may choose

$$b_k \in \left( \bigcap_{i \ne k} W^i \right) \setminus \{1\}. \qquad (64)$$

By construction, $b_k \ne 1$, and $b_k \in W^i$ for all $i \ne k$. Moreover, $b_k \notin W^k$, since otherwise $b_k$ would belong to $\bigcap_{i=1}^{\gamma} W^i = \{1\}$, a contradiction. Let $B = \{b_1, \ldots, b_\gamma\}$. Consider the bipartite graph with left vertices $\{W^1, \ldots, W^\gamma\}$ and right vertices $\{1\} \cup B \subseteq \{1, \ldots, m\}$, with an edge $W^i \leftrightarrow x$ whenever $x \in W^i$. A system of distinct representatives (SDR) of size $\gamma$ in this graph yields the desired elements $v_i \in W^i$. By Hall's theorem, it suffices to show that for every nonempty $J \subseteq \{1, \ldots, \gamma\}$, the neighborhood $N(J)$ satisfies $|N(J)| \ge |J|$.

- If $|J| = 1$, say $J = \{i\}$, then $1 \in W^i$. Furthermore, for every $k \ne i$ we have $b_k \in W^i$. Thus

$$|N(J)| \ge 1 + (\gamma - 1) = \gamma \ge |J|. \qquad (65)$$

- If $|J| \ge 2$, fix $k \in \{1, \ldots, \gamma\}$.
    - If $k \notin J$, then $b_k \in W^i$ for every $i \in J$, hence $b_k \in N(J)$.
    - If $k \in J$, pick any $j \in J \setminus \{k\}$. Since $b_k \in W^j$, it follows that $b_k \in N(J)$.

  Thus every $b_k$ belongs to $N(J)$, and clearly $1 \in N(J)$. Hence

$$|N(J)| \ge |B| + 1 = \gamma + 1 \ge |J|. \qquad (66)$$

Since Hall's condition is satisfied, there exists a matching that assigns to each $W^i$ a distinct element of $\{1\} \cup B$ contained in $W^i$. These assigned elements provide the required representatives $v_i \in W^i$, which are pairwise distinct.

**Step 5.**

To deliver the result of this part, we now employ the token indices $t_1, \ldots, t_\gamma$ identified in *(iii)* and *(iv)* of **Step 4**, together with the token indices in the set $S$ also obtained in **Step 4**. We recall the properties of these token indices that will be used:

1. For all $t \in S$, the partition $\{U_p^t\}_{p=1}^{\alpha_t}$, defined in **Step 4**, coincides with $\{U_p\}_{p=1}^{\alpha}$. In particular, by reindexing the head indices, we may assume $U_1 = \{1, \ldots, m\}$. This guarantees that the structure of the partition is stable across infinitely many $t \in S$, providing us with a consistent reference framework.

2. For all $t_i$ with $i \in [\gamma]$, where $\gamma < m$, recall that $V^{t_i} = U^{t_i}(1) \cap \{1, \ldots, m\}$. One can select $\gamma$ head indices $v_i \in V^{t_i}$ such that they are pairwise distinct. This property will be crucial later when we need to ensure that certain representatives can be chosen without overlap.

We also recall the main result from **Step 3**, namely Equation (57): for any $(s_1, \ldots, s_h) \in [L]^h$ with pairwise distinct entries, and for each $i \in [h]$, one has

$$0 = \sum_{\substack{\sigma \in S_h \,:\, A_j^{k,s_j} = A_{\sigma(j)}^{k,s_j} \ \forall j \in [h]}} B_{\sigma(i)}. \tag{67}$$

This identity is the foundation of the argument: Under the given matching condition on the coefficients $A_j^{k,s_j}$, a nontrivial linear combination of the $B_i$'s must vanish. Now, in Equation (67), let us consider $(s_1, \ldots, s_h) \in [L]^h$ constructed as follows. First, observe that the index set $\{1, \ldots, h\}$ can be decomposed into three disjoint parts:

$$\{1, \ldots, h\} = \{v_1, \ldots, v_\gamma\} \sqcup \left(\{1, \ldots, m\} \setminus \{v_1, \ldots, v_\gamma\}\right) \sqcup \left(U_2 \sqcup U_3 \sqcup \cdots \sqcup U_\alpha\right). \tag{68}$$

The first component corresponds to the specially chosen distinct representatives $v_i$, the second to the remaining elements of $U_1$, and the third to all indices belonging to the other partition classes $U_2, \ldots, U_\alpha$. Now fix a subset $T \subset [\gamma]$. Define $(s_1, \ldots, s_h) \in [L]^h$ by setting, for each $j \in [h]$,

1. If $j = v_i$ for some $i \in T$, then set $s_j = s_{v_i} = t_i$. In other words, the positions corresponding to $T$ are aligned with the distinguished token indices $t_i$.

2. If $j \in \{1, \ldots, m\} \setminus \{v_i : i \in T\}$, take $s_j$ to be an arbitrary element of $S$. This ensures consistency with the partition structure while leaving us flexibility in the assignment.

3. If $j \in U_p$ for some $2 \le p \le \alpha$, then take $s_j$ to be an arbitrary element of $S$. Again, this choice respects the partitioning of indices into classes $U_p$.

For the chosen $(s_1, \ldots, s_h) \in [L]^h$, we analyze which $\sigma \in S_h$ satisfy the condition $A_j^{k,s_j} = A_{\sigma(j)}^{k,s_j}$ for all $j \in [h]$. We make the following observations, case by case:

1. For $j \in U_2 \sqcup U_3 \sqcup \cdots \sqcup U_\alpha$, say $j \in U_p$ with $2 \le p \le \alpha$, the condition $A_j^{k,s_j} = A_{\sigma(j)}^{k,s_j}$ implies $\sigma(j) \in U_p$. Hence

$$\sigma(U_2 \sqcup U_3 \sqcup \cdots \sqcup U_\alpha) = U_2 \sqcup U_3 \sqcup \cdots \sqcup U_\alpha, \tag{69}$$

and consequently $\sigma(U_1) = U_1$. In particular, if $j \in U_1$, then $\sigma(j) \in U_1$.

2. For $j \in \{1, \ldots, m\} \setminus \{v_i : i \in T\}$, if $A_j^{k,s_j} = A_{\sigma(j)}^{k,s_j}$, then necessarily $\sigma(j) \in U_1 = \{1, \ldots, m\}$. Thus the entire set $U_1$ is stable under $\sigma$, but the specific images of these indices may vary within $U_1$.

3. For $j = v_i$ with $i \in T$, if $A_j^{k,s_j} = A_{\sigma(j)}^{k,s_j}$, then $\sigma(j) \in U^{s_{v_i}}(1) = U^{t_i}(1)$. From the previous point, we also know $\sigma(j) \in U_1$. Taken together, these conditions imply that $\sigma(j) \in V^{t_i} = U^{t_i}(1) \cap U_1$. In other words, the image of $v_i$ under $\sigma$ is constrained to lie inside the restricted set $V^{t_i}$.

Therefore, specifying a $\sigma \in S_h$ that satisfies $A_j^{k,s_j} = A_{\sigma(j)}^{k,s_j}$ for all $j \in [h]$ is equivalent to:

1. For each $j = v_i$ with $i \in T$, choosing $\sigma(j) = \sigma(v_i) \in V^{t_i}$,

2. For each $j \in \{1, \ldots, m\} \setminus \{v_i : i \in T\}$, choosing $\sigma(j) \in U_1 \setminus \{\sigma(v_i) : i \in T\}$ arbitrarily,

3. For each $j \in U_p$ with $2 \leq p \leq \alpha$, choosing $\sigma(j) \in U_p$.

In conclusion, the structure of admissible permutations $\sigma$ in Equation (67) is fully determined by the subset $T \subset [\gamma]$ and the representatives $v_i \in V^{t_i}$ chosen in **Step 4**. This description clarifies how the constraints arising from the partition classes $U_p$ and the distinguished representatives $v_i$ together restrict the allowed form of $\sigma$. Consequently, the sum in Equation (67) can be partitioned into contributions indexed by subsets $T \subset [\gamma]$, which will be the key mechanism for deriving vanishing conditions on the $B_i$'s in the subsequent step.

With these observations in hand, we now perform explicit computations. Fix one choice of $(s_1, \ldots, s_h) \in [L]^h$ satisfying the above construction, and in Equation (67) take $i = v_i$ for some $i \in T$. The equation then specializes to

$$0 = \sum_{\sigma \in S_h : A_j^{k,t_j} = A_{\sigma(j)}^{k,t_j} \ \forall j \in [h]} B_{\sigma(v_i)}$$

$$= \sum_{v \in V^{t_i}} B_v \cdot \Big(\text{the number of } h\text{-tuples in the Cartesian product} \prod_{j \in T} V^{t_j} \times U_1^{m-|T|} \times \prod_{p=2}^{\alpha} U_p^{|U_p|},$$

$$\text{such that all } h \text{ entries are pairwise distinct, and the coordinate corresponding to } V^{t_i} \text{ is fixed to be } v\Big). \quad (70)$$

The interpretation is as follows: each valid permutation $\sigma$ contributes one admissible tuple, and the contribution is grouped according to which element $v \in V^{t_i}$ is assigned to the coordinate corresponding to $V^{t_i}$. The factor multiplying $B_v$ therefore counts exactly the number of such admissible tuples. Now, observe that once the coordinates corresponding to the $V^{t_j}$'s are chosen, all the remaining coordinates can be filled freely within their respective partition blocks. In particular:

- The indices in $\{1, \ldots, m\} \setminus \{v_i : i \in T\}$ may be permuted arbitrarily within $U_1$, yielding a factor of $(m - |T|)!$.

- For each $p \in \{2, \ldots, \alpha\}$, the indices in $U_p$ may also be permuted arbitrarily, contributing a factor of $|U_p|!$.

Hence the above expression simplifies to

$$0 = \sum_{v \in V^{t_i}} B_v \cdot (m - |T|)! \cdot \prod_{p=2}^{\alpha} |U_p|! \cdot \Big(\text{the number of } h\text{-tuples in the Cartesian product} \prod_{j \in T} V^{t_j},$$

$$\text{such that all entries are pairwise distinct, and the coordinate corresponding to } V^{t_i} \text{ equals } v\Big). \quad (71)$$

Since the factorial factors are nonzero constants independent of the choice of $v$, we may divide them out to obtain the equivalent condition

$$0 = \sum_{v \in V^{t_i}} B_v \cdot \Big(\text{the number of } h\text{-tuples in the Cartesian product} \prod_{j \in T} V^{t_j},$$

$$\text{such that all entries are pairwise distinct, and the coordinate corresponding to } V^{t_i} \text{ equals } v\Big). \quad (72)$$

This identity holds for every choice of subset $T \subset [\gamma]$ and for every $v \in V^{t_i}$ with $i \in [\gamma]$. The key point is that the coefficients $B_v$ appear only through such linear relations, weighted by combinatorial counts of admissible tuples. By applying Corollary E.8, we deduce that

$$0 = \sum_{i \in V^{t_1} \cap V^{t_2} \cap \cdots \cap V^{t_\gamma}} B_i. \quad (73)$$

Finally, recall from the construction in *(iii)* of **Step 4** that the intersection $V^{t_1} \cap V^{t_2} \cap \cdots \cap V^{t_\gamma}$ is exactly $\{1\}$. Therefore, the above equation reduces to $B_1 = 0$. We have established that $B_1 = 0$. By the preceding argument at the beginning of the proof, this immediately implies that all $B_i$ vanish identically. Hence, we conclude that $B_i = 0$ for every $i$, which completes the proof. $\qquad \square$

**Remark D.2.** Theorem D.1 can be viewed as a statement about the linear independence of attention heads. Although the theorem is formulated under specific assumptions on the parameters of the MultiHead maps, these conditions are satisfied with probability one. In essence, the result asserts that – except for a negligibly small subset of the parameter space (e.g., a measure-zero set or the complement of a dense subset) – the functional equivalence of general MultiHead maps can be completely characterized. The probabilistic nature of these assumptions aligns with those commonly made in prior studies on the functional equivalence of deep neural networks.

We have the following corollary of Theorem D.1.

**Corollary D.3.** *Consider two* MHA *maps with $h$ and $\bar{h}$ heads, parameterized by families of matrices*

$$\left\{\{A_i^{m,n}\}_{m,n}\right\}_{i=1}^{h}, \left\{B_i\right\}_{i=1}^{h}, \ \ and \ \ \left\{\{\bar{A}_i^{m,n}\}_{m,n}\right\}_{i=1}^{\bar{h}}, \left\{\bar{B}_i\right\}_{i=1}^{\bar{h}} \ \ in \ \mathbb{R}^{d\times d}, \tag{74}$$

*respectively. Assume that $A_i^{m,n}$ and $\bar{A}_i^{m,n}$ are nonzero for all feasible triples $(i, m, n)$. If the two* MHA *maps are identical, then for every parameter family $\{A^{m,n}\}_{m,n}$ in $\mathbb{R}^{d\times d}$, we have the identity*

$$\sum_{i\in[h] \,:\, \{A_i^{m,n}\}_{m,n}=\{A^{m,n}\}_{m,n}} B_i = \sum_{i\in[\bar{h}] \,:\, \{\bar{A}_i^{m,n}\}_{m,n}=\{A^{m,n}\}_{m,n}} \bar{B}_i. \tag{75}$$

# E. Key Lemmas for the Functional Equivalence of General MultiHead Attention

In this section, we introduce the preliminary concepts and fundamental results that will serve as the foundation for the proofs of our main theorems.

## E.1. A Result on the Linear Independence of Exponential Polynomials over the Field of Rational Functions

Let $n$ be a positive integer. Recall that $\mathbb{R}[\mathbf{x}] = \mathbb{R}[x_1, \ldots, x_n]$ denotes the polynomial ring in $n$ variables over $\mathbb{R}$. Its field of fractions is denoted by $\mathbb{R}(\mathbf{x})$, that is,

$$\mathbb{R}(\mathbf{x}) = \left\{\frac{p}{q} \,:\, p, q \in \mathbb{R}[\mathbf{x}], \ q \neq 0\right\}, \tag{76}$$

the field of all rational functions in the variables $x_1, \ldots, x_n$ with real coefficients. We now state and prove a standard result concerning the linear independence of exponential polynomials over $\mathbb{R}(\mathbf{x})$.

**Lemma E.1.** *Let $p_1, \ldots, p_m$ be polynomials in $\mathbb{R}[\mathbf{x}]$ such that $p_i - p_j$ is nonconstant whenever $i \neq j$. Suppose $q_1, \ldots, q_m$ are rational functions in $\mathbb{R}(\mathbf{x})$ satisfying $q_1 \cdot e^{p_1} + \cdots + q_m \cdot e^{p_m} = 0$. Then necessarily $q_1 = \cdots = q_m = 0$.*

*Proof.* We proceed by induction on $m$.

*Base case.* For $m = 1$, the statement is immediate. Indeed, since $e^{p_1}$ never vanishes, $q_1 \cdot e^{p_1} = 0$ implies $q_1 = 0$.

*Inductive step.* Assume the result holds for every collection of fewer than $m$ exponentials. Let $q_1, \ldots, q_m \in \mathbb{R}(\mathbf{x})$ satisfy

$$q_1 \cdot e^{p_1} + \cdots + q_m \cdot e^{p_m} = 0. \tag{77}$$

We wish to show that all $q_i$ vanish. Suppose, for contradiction, that not all $q_i$ are zero. Without loss of generality, assume $q_m \neq 0$. Dividing through Equation (77) by $q_m e^{p_m}$ yields

$$\frac{q_1}{q_m} \cdot e^{p_1 - p_m} + \cdots + \frac{q_{m-1}}{q_m} \cdot e^{p_{m-1} - p_m} + 1 = 0. \tag{78}$$

This expresses 1 as a linear combination of the exponentials $e^{p_j - p_m}$ with coefficients in $\mathbb{R}(\mathbf{x})$. Now differentiate both sides of Equation (78) with respect to each variable $x_i$ for $i = 1, \ldots, n$. Since the derivative of 1 is zero, we obtain

$$\sum_{j=1}^{m-1} \left(\frac{\partial}{\partial x_i}\left(\frac{q_j}{q_m}\right) + \frac{q_j}{q_m} \cdot \frac{\partial}{\partial x_i}(p_j - p_m)\right) e^{p_j - p_m} = 0. \tag{79}$$

Each coefficient in parentheses lies in $\mathbb{R}(\mathbf{x})$. Since $p_1 - p_m, \ldots, p_{m-1} - p_m$ are pairwise distinct and nonconstant, the corresponding exponentials $e^{p_j - p_m}$ are linearly independent over $\mathbb{R}(\mathbf{x})$ by the induction hypothesis. Therefore, each coefficient in Equation (79) must vanish, i.e.,

$$\frac{\partial}{\partial x_i}\left(\frac{q_j}{q_m}\right) + \frac{q_j}{q_m} \cdot \frac{\partial}{\partial x_i}(p_j - p_m) = 0, \tag{80}$$

for every $i = 1, \ldots, n$ and $j = 1, \ldots, m - 1$. Equivalently,

$$\frac{\partial}{\partial x_i}\left(\frac{q_j}{q_m} \cdot e^{p_j - p_m}\right) = 0. \tag{81}$$

This shows that for each $j = 1, \ldots, m - 1$, the function $q_j/q_m \cdot e^{p_j - p_m}$ is independent of all variables $x_1, \ldots, x_n$, and hence must be a constant $c_j \in \mathbb{R}$. If some $c_j \neq 0$, then $q_j \neq 0$ and we would have $e^{p_j - p_m} = c_j q_m / q_j$, which would imply that $e^{p_j - p_m}$ is a rational function, and therefore constant. This contradicts the assumption that $p_j - p_m$ is nonconstant. Thus, each $c_j = 0$, forcing $q_j = 0$ for all $j = 1, \ldots, m - 1$. Substituting into Equation (78) then yields $1 = 0$, an impossibility. Hence our assumption was false, and all $q_i = 0$. By induction, the lemma follows. $\square$

**Remark E.2.** Lemma E.1 formalizes the intuitive fact that exponential functions with distinct polynomial exponents cannot cancel each other when combined with rational-function coefficients. It can be viewed as a multivariate generalization of the classical result that functions of the form $e^{ax}$ with distinct real numbers $a$ are linearly independent over the field of rational functions in one variable. Here, the same principle extends to exponential polynomials in several variables, with the essential role played by the assumption that the differences $p_i - p_j$ are nonconstant. This generalization is crucial for arguments in Theorem D.1, involving exponential polynomials over $\mathbb{R}(\mathbf{x})$.

### E.2. Hall's Marriage Theorem and Systems of Distinct Representatives

In this section, we recall a classical result from combinatorics, known as *Hall's Marriage Theorem* (Hall, 1935), which provides necessary and sufficient conditions for the existence of a system of distinct representatives (SDR). This theorem will play a crucial role in our arguments, as our construction ultimately reduces to the problem of selecting distinct representatives from a family of subsets. Let $\mathcal{A} = \{A_1, A_2, \ldots, A_s\}$ be a finite family of subsets of a ground set $X$. A *system of distinct representatives* (SDR) for $\mathcal{A}$ is a set $\{a_1, a_2, \ldots, a_s\}$ such that $a_i \in A_i$ for each $i$ and all $a_1, \ldots, a_s$ are pairwise distinct. Equivalently, an SDR is an injective choice function assigning to each $A_i$ an element $a_i \in A_i$.

The existence of an SDR is a classical question in combinatorics, and Hall's theorem provides a complete characterization.

**Theorem E.3** (Hall's Marriage Theorem). *Let $\mathcal{A} = \{A_1, A_2, \ldots, A_s\}$ be a finite family of subsets of a set $X$. Then $\mathcal{A}$ admits a system of distinct representatives if and only if the following condition (Hall's condition) holds:*

$$\left|\bigcup_{i \in J} A_i\right| \geq |J| \quad \text{for every subset } J \subseteq \{1, 2, \ldots, s\}. \tag{82}$$

In words, Hall's condition states that for every subcollection of the sets $A_i$, the total number of available elements in their union must be at least as large as the number of sets in the subcollection. This condition is clearly necessary: if $|J|$ sets are assigned representatives, then at least $|J|$ distinct elements are required. The theorem asserts that this necessary condition is also sufficient. Hall's theorem has many applications in combinatorics, graph theory, and matching theory. In the language of bipartite graphs, it gives a necessary and sufficient condition for the existence of a perfect matching from the left vertex set into the right vertex set.

**Remark E.4.** Hall's Marriage Theorem plays a central role in the argument of Theorem D.1. Moreover, its application is closely connected to the statements of Theorem E.7 and Corollary E.8.

### E.3. The Möbius Function on the Partition Lattice

This section introduces the necessary background on incidence algebras and Möbius inversion over finite posets. We then establish an identity for the Möbius function that will serve as a fundamental tool throughout the remainder of the paper. We also present several connections between this identity and other well-studied combinatorial concepts, with the aim of providing readers with greater intuition about its significance. For comprehensive treatments of these topics, we refer the reader to (Rota, 1964; Stanley, 2011).

E.3.1. INCIDENCE ALGEBRAS AND MÖBIUS INVERSION ON FINITE POSETS

Let $(P, \leq)$ be a finite poset. The *incidence algebra* $I(P)$ over $\mathbb{C}$ consists of all functions

$$f := \{(x, y) \in P \times P : x \leq y\} \longrightarrow \mathbb{C}. \tag{83}$$

with convolution

$$(f * g)(x, y) := \sum_{x \leq z \leq y} f(x, z)\, g(z, y), \quad \text{for all } x \leq y. \tag{84}$$

The identity for convolution is the Kronecker delta $\delta(x, y)$ (i.e. $\delta(x, y) = 1$ if $x = y$, and $0$ otherwise). The *zeta function* $\zeta \in I(P)$ is $\zeta(x, y) \equiv 1$ for $x \leq y$. An element $f \in I(P)$ is invertible if and only if $f(x, x) \neq 0$ for all $x \in P$; in that case $f^{-1}$ is its inverse under convolution.

**Möbius function.** The *Möbius function* $\mu = \mu_P \in I(P)$ is defined as the convolution inverse of $\zeta$:

$$\mu * \zeta = \zeta * \mu = \delta. \tag{85}$$

Equivalently, for all $x \leq y$ in $P$, one has

$$\sum_{x \leq z \leq y} \mu(x, z) = \delta(x, y). \tag{86}$$

As a consequence, if $f, g : P \to \mathbb{C}$ satisfy

$$f(x) = \sum_{y \geq x} g(y), \quad \text{for all } x \in P, \tag{87}$$

then *Möbius inversion* yields

$$g(x) = \sum_{y \geq x} \mu(x, y)\, f(y), \quad \text{for all } x \in P. \tag{88}$$

**Products of posets.** If $P, Q$ are finite posets, their product $P \times Q$ is ordered componentwise. Define

$$(\zeta_P \otimes \zeta_Q)\big((p_1, q_1), (p_2, q_2)\big) := \zeta_P(p_1, p_2)\, \zeta_Q(q_1, q_2). \tag{89}$$

A direct computation in $I(P \times Q)$ shows

$$\zeta_{P \times Q} = \zeta_P \otimes \zeta_Q, \tag{90}$$
$$(\mu_P \otimes \mu_Q) * (\zeta_P \otimes \zeta_Q) = \delta_P \otimes \delta_Q = \delta_{P \times Q}. \tag{91}$$

Hence

$$\mu_{P \times Q}\big((p_1, q_1), (p_2, q_2)\big) = \mu_P(p_1, p_2)\, \mu_Q(q_1, q_2). \tag{92}$$

E.3.2. THE PARTITION LATTICE AND INTERVAL FACTORIZATION

Let $U$ be a finite set with $|U| = n$. The set $\Pi(U)$ of all set partitions of $U$, ordered by refinement, forms a finite lattice with minimum $\hat{0}$ (all singletons) and maximum $\hat{1}$ (one block). The goal of this section is to derive the following explicit formula, stated in the following theorem:

**Theorem E.5.** *For $\pi \in \Pi(U)$, one has:*

$$\mu_{\Pi(U)}(\hat{0}, \pi) = \prod_{B \in \pi} (-1)^{|B|-1}(|B| - 1)!. \tag{93}$$

For clarity, we begin with an outline of the proof. The reasoning unfolds in two stages.

1. **Interval factorization.** Restriction to blocks induces a canonical isomorphism:

$$[\hat{0}, \pi] \cong \prod_{B \in \pi} \Pi(B). \tag{94}$$

   By multiplicativity of the Möbius function on products, one has:

$$\mu_{\Pi(U)}(\hat{0}, \pi) = \prod_{B \in \pi} \mu_{\Pi(B)}(\hat{0}_B, \hat{1}_B). \tag{95}$$

2. **One–block evaluation.** Using the exponential formula for labelled set partitions, for all $n \geq 1$, one has:

$$\mu_{\Pi([n])}(\hat{0}, \hat{1}) = (-1)^{n-1}(n-1)!. \tag{96}$$

   Substituting into the product from Step 1 yields

$$\mu_{\Pi(U)}(\hat{0}, \pi) = \prod_{B \in \pi} (-1)^{|B|-1}(|B| - 1)!. \tag{97}$$

Having outlined the strategy, we now provide the full proof with all intermediate steps made explicit.

*Proof.* We structure the proof into several steps for the sake of clarity and readability.

**Step 1.**

A partition $\pi \in \Pi(U)$ is a set of disjoint nonempty blocks $B \subseteq U$ covering $U$. For $\sigma, \pi \in \Pi(U)$ write $\sigma \leq \pi$ if every block of $\sigma$ is contained in a block of $\pi$. For $\sigma \leq \pi$ and a block $B \in \pi$, let $\sigma|_B$ be the restriction of $\sigma$ to $B$ (intersect each block of $\sigma$ with $B$ and remove empties). Denote by $\hat{1}_B$ the one-block partition of $B$. We have the following result.

**Lemma E.6** (Interval factorization). *For $\sigma \leq \pi$ in $\Pi(U)$, restriction induces a poset isomorphism*

$$\Phi: \; [\sigma, \pi] \; \longrightarrow \; \prod_{B \in \pi} \Pi\big(\sigma|_B, \hat{1}_B\big), \qquad \Phi(\tau): \big(\tau|_B\big)_{B \in \pi}. \tag{98}$$

*Its inverse maps $(\rho_B)_{B \in \pi}$ to the join $\bigvee_{B \in \pi} \rho_B$, which coincides with the partition whose restriction to each $B$ equals $\rho_B$.*

*Proof.* If $\tau \in [\sigma, \pi]$, then $\sigma \leq \tau \leq \pi$ implies that each block of $\tau$ lies inside some block of $\pi$, so $\tau|_B$ is a partition of $B$ refining $\sigma|_B$, hence $\sigma|_B \leq \tau|_B \leq \hat{1}_B$. Thus $\Phi$ is well-defined and order-preserving. Conversely, if $(\rho_B)_{B \in \pi}$ satisfies $\sigma|_B \leq \rho_B \leq \hat{1}_B$, define $\rho$ by declaring that $x, y \in U$ lie in the same block of $\rho$ iff either $x, y \in B$ and $x \sim_{\rho_B} y$ for some $B \in \pi$, or $x, y$ lie in different blocks of $\pi$ (which never happens since we work blockwise). Then $\rho$ is a partition with $\sigma \leq \rho \leq \pi$ and $\rho|_B = \rho_B$. One checks $\Phi(\rho) = (\rho_B)$ and $\bigvee_{B \in \pi}(\tau|_B) = \tau$, hence $\Phi$ is an isomorphism. $\square$

Setting $\sigma = \hat{0}$ in Lemma E.6 yields

$$[\hat{0}, \pi] \; \cong \; \prod_{B \in \pi} \Pi(B). \tag{99}$$

Applying the multiplicativity Equation (92) to Equation (99), one has

$$\mu_{\Pi(U)}(\hat{0}, \pi) = \prod_{B \in \pi} \mu_{\Pi(B)}(\hat{0}_B, \hat{1}_B). \tag{100}$$

Therefore, to compute $\mu_{\Pi(U)}(\hat{0}, \pi)$ for arbitrary $\pi$, it suffices to evaluate the single-block quantity

$$m(n) := \mu_{\Pi_n}(\hat{0}, \hat{1}), \tag{101}$$

for $n \in \mathbb{N}$, where $\Pi_n$ denotes the partition lattice on an $n$-element set.

**Step 2.**

We now determine $m(n)$ exactly. One has a Möbius sum constraint as follows: by Equation (86), for every finite poset and any $x < y$, one has

$$\sum_{x \leq z \leq y} \mu(x, z) = 0. \tag{102}$$

In $\Pi_n$, taking $x = \hat{0}$ and $y = \hat{1}$ gives

$$\sum_{\tau \in \Pi_n} \mu_{\Pi_n}(\hat{0}, \tau) = 0, \tag{103}$$

for all $n \geq 2$. For $n = 0, 1$, the sum equals 1 (the unique element of the interval). By Equation (100) applied inside $\Pi_n$, one has

$$\mu_{\Pi_n}(\hat{0}, \tau) = \prod_{B \in \tau} m(|B|). \tag{104}$$

Define

$$F_n := \sum_{\tau \in \Pi_n} \prod_{B \in \tau} m(|B|). \tag{105}$$

Then, for $n \geq 2$, one has $F_0 = 1, F_1 = 1, F_n = 0$. A standard labeled-partition identity (the exponential formula) asserts that for any sequence $(a_k)_{k \geq 1}$,

$$\sum_{n \geq 0} \left( \sum_{\tau \in \Pi_n} \prod_{B \in \tau} a_{|B|} \right) \frac{z^n}{n!} = \exp \left( \sum_{k \geq 1} a_k \frac{z^k}{k!} \right). \tag{106}$$

Applying this with $a_k = m(k)$ yields

$$\sum_{n \geq 0} F_n \frac{z^n}{n!} = \exp \left( \sum_{k \geq 1} m(k) \frac{z^k}{k!} \right). \tag{107}$$

The LHS of Equation (107) equals $1 + z$. Taking the formal logarithm gives

$$\sum_{k \geq 1} m(k) \frac{z^k}{k!} = \log(1 + z) = \sum_{k \geq 1} (-1)^{k-1} \frac{z^k}{k}. \tag{108}$$

Equating coefficients, for $k \geq 1$, one has

$$m(k) = k! \cdot \frac{(-1)^{k-1}}{k} = (-1)^{k-1} (k-1)!. \tag{109}$$

Substituting Equation (109) into the block factorization Equation (100) gives the desired expression in Equation (93):

$$\mu_{\Pi(U)}(\hat{0}, \pi) = \prod_{B \in \pi} (-1)^{|B|-1} (|B| - 1)!. \tag{110}$$

This concludes the proof. $\qquad \square$

The identity established in Theorem E.5 plays a pivotal role in the proof of Theorem E.7, which, in turn, functions as a supporting lemma for the proof of Theorem D.1 – the main result of this work.

### E.4. A Technical Result on Weighted Sums over Distinct Tuples

We now present a result concerning weighted sums over distinct tuples. The results developed in this section form the backbone of our argument in the proof of Theorem D.1, and they encapsulate the main technical difficulty of that proof.

**Theorem E.7.** *Given positive integers $m, n \geq 1$. For each $i \in [m]$, let $A_i$ be a subset of $[n]$. Let $x_1, \ldots, x_n$ be $n$ real numbers. For any nonempty $S \subseteq [m]$, define*

$$F_S := \Big\{ (a_i)_{i \in S} : a_i \in A_i \text{ for all } i \in S, \text{ and all } a_i\text{'s are pairwise distinct} \Big\}. \tag{111}$$

*For $i \in S$ and $a \in A_i$, define the fiber*

$$F_{S,i,a} := \{ (a_j)_{j \in S} \in F_S : a_i = a \}. \tag{112}$$

*For any nonempty $T \subseteq [m]$, define $A_T := \bigcap_{i \in T} A_i$, and*

$$G(T) := \sum_{a \in A_T} x_a. \tag{113}$$

*Assume that, for every nonempty $S \subseteq [m]$ and every $i \in S$, one has*

$$\sum_{a \in A_i} |F_{S,i,a}| \, x_a = 0. \tag{114}$$

*Then, for every nonempty $T \subseteq [m]$, one has*

$$G(T) = \sum_{a \in A_T} x_a = 0. \tag{115}$$

*Proof.* Let $S$ be a nonempty finite set. Denote by $\Pi(S)$ the lattice of set partitions of $S$ ordered by refinement: For $\sigma, \pi \in \Pi(S)$, we write $\sigma \leq \pi$ if every block of $\sigma$ is contained in a block of $\pi$. Any $\pi \in \Pi(S)$ is a family of disjoint nonempty blocks whose union is $S$. For a block $B \subseteq S$ define

$$A_B := \bigcap_{j \in B} A_j, \qquad \text{and} \qquad |A_B| := \Big| \bigcap_{j \in B} A_j \Big|. \tag{116}$$

Let $\mu$ denote the Möbius function of $\Pi(S)$ (with respect to refinement). $\mu$ is determined by $\sum_{\sigma : \sigma \leq \pi} \mu(\sigma) = \mathbf{1}_{\{\pi = \hat{0}\}}$, where $\hat{0}$ is the discrete partition. It is well-known that:

$$\mu(\pi) = \prod_{B \in \pi} (-1)^{|B|-1} (|B| - 1)!. \tag{117}$$

Fix a nonempty $S \subseteq [m]$, an index $i \in S$, and an element $a \in [n]$. Let $\mathcal{G}_S$ be the set of all functions $g : S \to [n]$ satisfying $g(j) \in A_j$ for all $j \in S$. For $g \in \mathcal{G}_S$, define its equality partition $\pi(g) \in \Pi(S)$ by:

$$j \sim_{\pi(g)} k \quad \text{if and only if} \quad g(j) = g(k). \tag{118}$$

Thus $\pi(g)$ records which indices are assigned the same value by $g$. One has $g$ is injective on $S$ if and only if $\pi(g) = \hat{0}$. The set $F_S$ of injective choices can be described as:

$$F_S = \Big\{ g \in \mathcal{G}_S : \pi(g) = \hat{0} \Big\}, \tag{119}$$

and the *fiber* fixing the value at the distinguished index $i$ is:

$$F_{S,i,a} = \Big\{ g \in \mathcal{G}_S : g(i) = a, \, \pi(g) = \hat{0} \Big\}. \tag{120}$$

For $\pi \in \Pi(S)$ and $i \in S$, let $B_i(\pi)$ denote the unique block of $\pi$ containing $i$. Define:

$$N_{S,i,a}(\pi) := \Big| \big\{ g \in \mathcal{G}_S : g \text{ is constant on each block of } \pi, \, g(i) = a \big\} \Big|. \tag{121}$$

That is, $N_{S,i,a}(\pi)$ counts maps that are constant along blocks of $\pi$ (so the only equalities allowed among coordinates are those forced by $\pi$) and take the prescribed value $a$ at the index $i$. For every $\pi \in \Pi(S)$, one has:

$$N_{S,i,a}(\pi) = \mathbf{1}_{\{a \in A_{B_i(\pi)}\}} \prod_{B \in \pi \ \text{and} \ B \neq B_i(\pi)} |A_B|. \tag{122}$$

Indeed, if $g$ is constant on each block of $\pi$, the value on the block $B_i(\pi)$ must equal $g(i) = a$. This is possible exactly when $a \in \bigcap_{j \in B_i(\pi)} A_j = A_{B_i(\pi)}$, which contributes the indicator $\mathbf{1}_{\{a \in A_{B_i(\pi)}\}}$. Then, for any other block $B \in \pi$ with $B \neq B_i(\pi)$, the common value of $g$ on $B$ can be chosen arbitrarily from the intersection $A_B = \bigcap_{j \in B} A_j$, independently across distinct blocks. Therefore there are $|A_B|$ choices for each such block, and multiplying over all $B \neq B_i(\pi)$ yields the product in Equation (122). Now, for $g \in \mathcal{G}_S$, define the two indicator functions on $\Pi(S)$:

$$E(g) := \mathbf{1}_{\{\pi(g) = \hat{0}\}}, \ \text{and} \ \ C_\pi(g) := \mathbf{1}_{\{\pi(g) \geq \pi\}} \ \ (\pi \in \Pi(S)). \tag{123}$$

Here $\pi(g) \geq \pi$ means that $g$ is constant on every block of $\pi$. By general Möbius inversion on posets, one has:

$$E(g) \ = \ \sum_{\pi \in \Pi(S)} \mu(\pi) \, C_\pi(g), \ \text{since} \ \sum_{\sigma \leq \pi(g)} \mu(\sigma) = \mathbf{1}_{\{\pi(g) = \hat{0}\}}. \tag{124}$$

Now fix $i \in S$ and $a \in [n]$, multiply the last identity by $\mathbf{1}_{\{g(i) = a\}}$, and sum over all $g \in \mathcal{G}_S$, one has:

$$\left| F_{S,i,a} \right| = \sum_{g \in \mathcal{G}_S} \mathbf{1}_{\{g(i) = a\}} E(g) = \sum_{\pi \in \Pi(S)} \mu(\pi) \sum_{g \in \mathcal{G}_S} \mathbf{1}_{\{g(i) = a\}} C_\pi(g). \tag{125}$$

The inner sum is precisely $N_{S,i,a}(\pi)$ by definition. Using Equation (122), one therefore obtains the explicit expansion:

$$|F_{S,i,a}| = \sum_{\pi \in \Pi(S)} \mu(\pi) \, \mathbf{1}_{\{a \in A_{B_i(\pi)}\}} \prod_{B \in \pi \ \text{and} \ B \neq B_i(\pi)} |A_B|. \tag{126}$$

Multiply Equation (126) by $x_a$ and sum over all $a \in A_i$ (equivalently, over all $a \in [n]$, since the indicator in Equation (126) already forces $a \in A_i$ when $i \in B_i(\pi)$):

$$\sum_{a \in A_i} |F_{S,i,a}| \, x_a = \sum_{\pi \in \Pi(S)} \mu(\pi) \left( \prod_{B \in \pi \ \text{and} \ B \neq B_i(\pi)} |A_B| \right) \left( \sum_{a \in A_{B_i(\pi)}} x_a \right). \tag{127}$$

With the shorthand $G(T) := \sum_{a \in A_T} x_a$ this becomes

$$\sum_{a \in A_i} |F_{S,i,a}| \, x_a = \sum_{\pi \in \Pi(S)} \mu(\pi) \left( \prod_{B \in \pi \ \text{and} \ B \neq B_i(\pi)} |A_B| \right) G\big(B_i(\pi)\big). \tag{128}$$

By the hypothesis, the LHS of Equation (128) is 0. Hence

$$0 = \sum_{\pi \in \Pi(S)} \mu(\pi) \left( \prod_{B \in \pi \ \text{and} \ B \neq B_i(\pi)} |A_B| \right) G\big(B_i(\pi)\big), \tag{129}$$

for every nonempty $S \subseteq [m]$ and every $i \in S$. Observe that, in Equation (129), the term $G\big(B_i(\pi)\big)$ only involves nonempty subsets $B_i(\pi)$ with $i \in B_i(\pi) \subseteq S$.

Back to the problem. We now show that $G(T) = 0$ for every nonempty $T \subseteq [m]$ by induction on $k := |T|$.

*Base case.*

Let $T = \{i\}$ for some $i \in [m]$. Take $S = \{i\}$ in the given hypothesis, one has

$$\sum_{a \in A_i} |F_{S,i,a}| x_a = 0. \tag{130}$$

Since $S$ has one element, an injective choice on $S$ is just a choice of a value in $A_i$, hence $|F_{\{i\},i,a}| = \mathbf{1}_{\{a \in A_i\}}$. Therefore

$$0 = \sum_{a \in A_i} |F_{\{i\},i,a}| x_a = \sum_{a \in A_i} x_a = G(\{i\}), \tag{131}$$

which establishes the base case.

*Inductive step.*

Fix $k \geq 2$ and assume the claim holds for all nonempty $U \subseteq [m]$ with $|U| < k$, i.e., $G(U) = 0$ whenever $1 \leq |U| \leq k - 1$. Let $T \subseteq [m]$ with $|T| = k$, and fix any distinguished index $i \in T$. Apply Equation (129) with $S = T$, we analyze the sum over $\pi \in \Pi(T)$ by separating the one–block partition from the rest.

*(i) The contribution of the one–block partition.*

There is a unique partition $\pi^\star = \{T\}$ with a single block. For this partition we have $B_i(\pi^\star) = T$, and the product over $B \neq B_i(\pi^\star)$ is an empty product, hence equals 1 by convention. By Equation (117) with $|T| = k$, one has:

$$\mu(\pi^\star) = (-1)^{k-1}(k-1)!. \tag{132}$$

Thus, the term of Equation (129) corresponding to $\pi^\star$ equals

$$\mu(\pi^\star) \cdot 1 \cdot G\big(B_i(\pi^\star)\big) = (-1)^{k-1}(k-1)! G(T). \tag{133}$$

*(ii) The contribution of all other partitions.*

Let $\pi \in \Pi(T)$ with $\pi \neq \pi^\star$. Then $B_i(\pi)$ is a proper, nonempty subset of $T$ (it still contains $i$ but does not equal $T$). Consequently $|B_i(\pi)| \leq k - 1$. By the inductive hypothesis,

$$G\big(B_i(\pi)\big) = 0.$$

Hence every summand in Equation (129) with $\pi \neq \pi^\star$ vanishes, regardless of the multiplicative factor $\prod_{B \neq B_i(\pi)} |A_B|$ and the value of $\mu(\pi)$.

Collecting (a) and (b), identity Equation (129) with $S = T$ reduces to

$$0 = (-1)^{k-1}(k-1)! G(T). \tag{134}$$

Since $(-1)^{k-1}(k-1)! \neq 0$, we conclude $G(T) = 0$.

By induction on $k$, the relation $G(T) = 0$ holds for every nonempty $T \subseteq [m]$. $\square$

We have a direct corollary of Theorem E.7.

**Corollary E.8.** *Given positive integers $m, n \geq 1$. For each $i \in [m]$, let $A_i$ be a subset of $[n]$. Let $x_1, \ldots, x_n$ be $n$ real numbers. For any nonempty $S \subseteq [m]$, define*

$$F_S := \Big\{(a_i)_{i \in S} : a_i \in A_i \text{ for all } i \in S, \text{ and all } a_i\text{'s are pairwise distinct}\Big\}. \tag{135}$$

*For $i \in S$ and $a \in A_i$, define the fiber*

$$F_{S,i,a} := \{(a_j)_{j \in S} \in F_S : a_i = a\}. \tag{136}$$

*Assume that, for every nonempty $S \subseteq [m]$ and every $i \in S$, one has*

$$\sum_{a \in A_i} |F_{S,i,a}| x_a = 0. \tag{137}$$

*Then, one has*

$$G(T) = \sum_{a \in A_1 \cap \ldots \cap A_m} x_a = 0. \tag{138}$$

*Proof.* By taking $T = [m]$ in Theorem E.7, one obtains the asserted main conclusion. $\square$

# F. Functional Equivalence of Multihead Attention with Rotary Positional Encoding

## F.1. Main Result on Functional Equivalence

**Theorem F.1** (Theorem 4.2 in the main paper). *Given two positive integers $d$ and $d_h$ with $d > d_h$. Consider two $\mathrm{MHA}_{\mathrm{RoPE}}$ maps with $h$ and $\bar{h}$ heads, with rotary positional encoding. They are parameterized by families of matrices*

$$\theta = \left(W_i^Q, W_i^K, W_i^V, W_i^O\right)_{i=1}^{h} \in \Theta(d, d_h, h), \ \ and \ \ \bar{\theta} = \left(\bar{W}_i^Q, \bar{W}_i^K, \bar{W}_i^V, \bar{W}_i^O\right)_{i=1}^{\bar{h}} \in \Theta(d, d_h, \bar{h}), \tag{139}$$

*respectively. Denote*

$$A_i^0 := \mathrm{sym}\big(W_i^Q(W_i^K)^\top\big), \ \ and \ \ A_i^n := W_i^Q R_n (W_i^K)^\top \ \ if \ n \neq 0. \ \ Same \ for \ \bar{A}_i^n.$$

*Assume that*

1. *All matrices $A_i^n$ and $\bar{A}_i^n$, for feasible $i$ and $n \in \mathbb{Z}$, are nonzero.*

2. *From $\theta$, the $h$ families $\{A_1^n\}_{n\in\mathbb{Z}}, \ldots, \{A_h^n\}_{n\in\mathbb{Z}}$ are pairwise distinct. The same condition holds for $\bar{\theta}$.*

3. *All matrices $W_i^Q, W_i^K, W_i^V, W_i^O$ and $\bar{W}_i^Q, \bar{W}_i^K, \bar{W}_i^V, W_i^O$, for feasible $i$, are of rank $d_h$.*

*If the two $\mathrm{MHA}_{\mathrm{RoPE}}$ maps are identical, then $h = \bar{h}$. Moreover, there exists $g \in G_{\mathrm{RoPE}}(d_h, h)$ such that $\bar{\theta} = g\theta$.*

*Proof.* For $i \in [h]$ and $m, n \geq 1$, define $A_i^{m,n} = A_i^{m-n}$ and $B_i := W_i^V(W_i^O)^\top$. Same for $\bar{A}_i^{m,n}$ and $\bar{B}_i$. Then, one has

$$\mathrm{MHA}\Big(\mathbf{x}; \{\{A_i^{m,n}\}_{m,n}, B_i\}_{i=1}^h\Big) = \mathrm{MHA}_{\mathrm{RoPE}}\Big(\mathbf{x}; \theta\Big) = \mathrm{MHA}_{\mathrm{RoPE}}\Big(\mathbf{x}; \bar{\theta}\Big) = \mathrm{MHA}\Big(\mathbf{x}; \{\{\bar{A}_i^{m,n}\}_{m,n}, \bar{B}_i\}_{i=1}^{\bar{h}}\Big). \tag{140}$$

From the condition 2, the property of parameters from these maps fit to the setting of Corollary D.3, which is that $A_i^{m,n}$ and $\bar{A}_i^{m,n}$ are nonzero for all feasible triples $(i, m, n)$. Thus, for every parameter family $\{A^{m,n}\}_{m,n} \subset \mathbb{R}^{d \times d}$, the following identity holds

$$\sum_{i\in[h]\,:\,\{A_i^{m,n}\}_{m,n}=\{A^{m,n}\}_{m,n}} B_i = \sum_{i\in[\bar{h}]\,:\,\{\bar{A}_i^{m,n}\}_{m,n}=\{A^{m,n}\}_{m,n}} \bar{B}_i. \tag{141}$$

The $h$ families $\{A_1^{m,n}\}_{m,n\geq 1}, \{A_2^{m,n}\}_{m,n\geq 1}, \ldots, \{A_h^{m,n}\}_{m,n\geq 1}$ are pairwise distinct. Together with Equation (141), consider $\{A^{m,n}\}_{m,n} = \{A_i^{m,n}\}_{m,n}$, one has the LHS of Equation (141) is equal to $B_i$. Thus,

$$B_i = \sum_{j\in[\bar{h}]\,:\,\{\bar{A}_j^{m,n}\}_{m,n}=\{A_i^{m,n}\}_{m,n}} \bar{B}_j. \tag{142}$$

Note that, since all the matrices $W_i^V$ and $W_i^O$ have rank $d_h$, it implies that all $B_i$ are non-zero. From Equation (142), for each $i \in [h]$, since the left-hand side is non-zero, the right-hand side has at least one index $j \in [\bar{h}]$ such that $\bar{B}_j$ is non-zero and $\{\bar{A}_j^{m,n}\}_{m,n} = \{A_i^{m,n}\}_{m,n}$. Since $h$ families $\{A_1^{m,n}\}_{m,n\geq 1}, \{A_2^{m,n}\}_{m,n\geq 1}, \ldots, \{A_h^{m,n}\}_{m,n\geq 1}$, are pairwise distinct, one implies that each $i$ has its corresponding $j$'s distinctly from others. Thus, $h \leq \bar{h}$. By a symmetric argument, one also has $h \geq \bar{h}$. In conclusion, one has $h = \bar{h}$. Moreover, by the above argument, for each $i$, there exists exactly one $j \in [h]$ such that $\{\bar{A}_j^{m,n}\}_{m,n} = \{A_i^{m,n}\}_{m,n}$. Moreover, this also implies that $B_j = B_i$. In conclusion, there exists a permutation $\sigma \in S_h$ such that

$$\bar{A}_i^{m,n} = A_{\sigma(i)}^{m,n}, \ \ for \ all \ m, n \geq 1, \ \ and \ \ \bar{B}_{\sigma(i)} = B_i. \tag{143}$$

From Lemma F.2, there exists matrices $\{U_i\}_{i=1}^h \subset \mathrm{H}(d_h)$ such that

$$\bar{W}_i^Q = W_{\sigma(i)}^Q \cdot U_i^\top, \ \ \ \bar{W}_i^K = W_{\sigma(i)}^K \cdot (U_i)^{-1}. \tag{144}$$

From the rank factorization (Piziak & Odell, 1999), there exists matrices $\{V_i\}_{i=1}^h \subset \mathrm{GL}(d_h)$ such that

$$\bar{W}_i^V = W_{\sigma(i)}^V \cdot V_i^\top, \ \ \ \bar{W}_i^O = W_{\sigma(i)}^O \cdot (V_i)^{-1}. \tag{145}$$

This concludes the proof. $\qquad \square$

## F.2. A Lemma Concerning the Rotary Matrix

Given $d = 2m$ be an even integer. Consider the RoPE matrix at position 1 as

$$R = \text{diag}\big(R(\theta_1), \ldots, R(\theta_{d/2})\big) \in \mathbb{R}^{d \times d}, \quad \text{where } R(\theta) = \begin{bmatrix} \cos\theta & -\sin\theta \\ \sin\theta & \cos\theta \end{bmatrix}. \tag{146}$$

Denote the $n \times n$ identity matrix as $I_n$. For $i = 1, \ldots, m$, define the 2-dimensional coordinate plane

$$E_i := \text{span}\{e_{2i-1}, e_{2i}\} \subset \mathbb{R}^d, \text{ where } e_{2i-1}, e_{2i} \text{ are the } (2i-1)\text{-th and } 2i\text{-th coordinate basis vectors.} \tag{147}$$

Define $P_i := e_{2i-1}e_{2i-1}^\top + e_{2i}e_{2i}^\top$ and $J_i := e_{2i}e_{2i-1}^\top - e_{2i-1}e_{2i}^\top$ in $\mathbb{R}^{d \times d}$. In words, $P_i$ and $J_i$ are the $d \times d$ matrices that the $i$-th $2 \times 2$ diagonal block is

$$I := \begin{bmatrix} 1 & 0 \\ 0 & 1 \end{bmatrix}, \qquad J := \begin{bmatrix} 0 & -1 \\ 1 & 0 \end{bmatrix}, \tag{148}$$

respectively. The matrix $R$ now can be written as $R = \sum_{i=1}^m (\cos\theta_i P_i + \sin\theta_i J_i)$. We have the following result.

**Lemma F.2.** *Given an integer $D \geq d$. Consider matrices $X, Z \in \mathbb{R}^{D \times d}$ and $Y, T \in \mathbb{R}^{d \times D}$. Assume that, for all non zero interger $n$, we have $XR^nY = ZR^nT$. Assume that all the angles $\theta_i \in (0, \pi)$ are pairwise distinct, and $XP_i$ and $P_iY$ are of rank 2 for $i \in [m]$. Then, there exists an invertible matrix $U \in \mathbb{R}^{d \times d}$ of the form*

$$U = \sum_{i=1}^m (a_i P_i + b_i J_i) \text{ with } (a_i, b_i) \in \mathbb{R}^2 \setminus \{(0,0)\} \text{ for } i = 1, \ldots, m, \tag{149}$$

*such that $Z = XU$ and $T = U^{-1}Y$.*

*Proof.* We structure the proof into several steps for the sake of clarity and readability.

**Step 1.**

Define $A_{1,i} := XP_iY, B_{1,i} := XJ_iY, A_{2,i} := ZP_iT, B_{2,i} := ZJ_iT$ in $\mathbb{R}^{D \times D}$. One has

$$XR^nY = \sum_{i=1}^m X\left(\cos(n\theta_i)P_i + \sin(n\theta_i)J_i\right)Y$$

$$= \sum_{i=1}^m \left(\cos(n\theta_i)XP_iY + \sin(n\theta_i)XJ_iY\right) = \sum_{i=1}^m \left(\cos(n\theta_i)A_{1,i} + \sin(n\theta_i)B_{1,i}\right), \tag{150}$$

$$ZR^nT = \sum_{i=1}^m Z\left(\cos(n\theta_i)P_i + \sin(n\theta_i)J_i\right)T$$

$$= \sum_{i=1}^m \left(\cos(n\theta_i)ZP_iT + \sin(n\theta_i)ZJ_iT\right) = \sum_{i=1}^m \left(\cos(n\theta_i)A_{2,i} + \sin(n\theta_i)B_{2,i}\right). \tag{151}$$

Since $XR^nY = ZR^nT$ for all $n \neq 0$, and $\theta_1, \theta_2, \ldots, \theta_m$ are pairwise distinct, one has $A_{1,i} = A_{2,i}$ and $B_{1,i} = B_{2,i}$ for all $i \in [m]$, which are $XP_iY = ZP_iT$ and $XJ_iY = ZJ_iT$.

**Step 2.**

Now fix an number $i \in \{1, \ldots, m\}$. Let $X_i$ is the $D \times 2$ matrix constructed by concating the $(2i-1)$-th and $2i$-th columns of $X$, $Y_i$ be the $2 \times D$ matrix constructed by concating the $(2i-1)$-th and $2i$-th rows of $Y$. Similarly, we construct $Z_i, T_i$ for $Z, T$, respectively. By the second assumption, we have both $X_i$ and $Y_i$ have rank 2. Moreover, from $XP_iY = ZP_iT$ and $XJ_iY = ZJ_iT$, one has $X_iY_i = Z_iT_i$ and $X_iJY_i = Z_iJT_i$. Let $V_X \in \mathbb{R}^{2 \times D}$ be the left inverse matrix of $X_i$ and $V_Y \in \mathbb{R}^{D \times 2}$ be the right inverse matrix of $Y_i$, i.e., $V_XX_i = Y_iV_Y = I_2$. One has

$$I_2 = (V_XX_i)(Y_iV_Y) = V_X(X_iY_i)V_Y = V_X(Z_iT_i)V_Y = (V_XZ_i)(T_iV_Y). \tag{152}$$

Let $U_i = V_X Z_i$. Then $U_i^{-1} = T_i V_Y$. Moreover, one has

$$X_i = X_i(Y_i V_Y) = (X_i Y_i)V_Y = (Z_i T_i)V_Y = Z_i(T_i V_Y) = Z_i U_i^{-1}, \tag{153}$$

so $Z_i = X_i U_i$. Similarly,

$$Y_i = (V_X X_i)Y_i = V_X(X_i Y_i) = V_X(Z_i T_i) = (V_X Z_i)T_i = U_i T_i, \tag{154}$$

so $T_i = U_i^{-1} Y_i$. Now, from $X_i J Y_i = Z_i J T_i$, one has

$$J = (V_X X_i)J(Y_i V_Y) = V_X(X_i J Y_i)V_Y = V_X(Z_i J T_i)V_Y = (V_X Z_i)J(T_i V_Y) = U_i J U_i^{-1}. \tag{155}$$

In other words, one has $U_i J = J U_i$. Then, there exists $(a_i, b_i) \in \mathbb{R}^2 \setminus \{(0,0)\}$ such that $U_i = a_i I_2 + b_i J$. In conclusion, one has $Z_i = X_i U_i$ and $T_i = U_i^{-1} Y_i$, where $U_i = a_i I_2 + b_i J$ with $(a_i, b_i) \in \mathbb{R}^2 \setminus \{(0,0)\}$.

**Step 3.**

Define $U = \text{diag}(U_1, \ldots, U_m)$. From the property of $U_i$'s, we have

$$U = \sum_{i=1}^{m}(a_i P_i + b_i J_i) \text{ with } (a_i, b_i) \in \mathbb{R}^2 \setminus \{(0,0)\} \text{ for } i = 1, \ldots, m, \tag{156}$$

and $Z = XU$ and $T = U^{-1}Y$. This concludes the proof. □

This result will be invoked in the proof of Theorem F.1.

**Remark F.3** (On the assumptions of Lemma F.2). If angles are not distinct or some equal 0 or $\pi$, first merge blocks with equal $\theta$ and repeat the argument within each frequency class; the conclusion remains that $U$ must commute with $R$ (hence with each $J_i$) on the active subspaces. If $\text{rank}(X P_i) < 2$ or $\text{rank}(P_i Y) < 2$ for some $i$, the same derivation shows $C_i$ must commute with $J_i$ on the image subspace; $C_i$ may be non-unique, but the global relation $Z = XU$, $T = U^{-1}Y$ with $U$ commuting with $R$ still describes the solution set restricted to the active coordinates.

**Remark F.4** (Concrete matrix forms). We provide the explicit form of the matrices used in the above argument for the case $d = 6$ (i.e., $m = 3$), expressed in the standard basis $(e_1, \ldots, e_6)$, to facilitate readability.

$$P_1 = \begin{bmatrix} 1 & 0 & 0 & 0 & 0 & 0 \\ 0 & 1 & 0 & 0 & 0 & 0 \\ 0 & 0 & 0 & 0 & 0 & 0 \\ 0 & 0 & 0 & 0 & 0 & 0 \\ 0 & 0 & 0 & 0 & 0 & 0 \\ 0 & 0 & 0 & 0 & 0 & 0 \end{bmatrix}, \quad P_2 = \begin{bmatrix} 0 & 0 & 0 & 0 & 0 & 0 \\ 0 & 0 & 0 & 0 & 0 & 0 \\ 0 & 0 & 1 & 0 & 0 & 0 \\ 0 & 0 & 0 & 1 & 0 & 0 \\ 0 & 0 & 0 & 0 & 0 & 0 \\ 0 & 0 & 0 & 0 & 0 & 0 \end{bmatrix}, \quad P_3 = \begin{bmatrix} 0 & 0 & 0 & 0 & 0 & 0 \\ 0 & 0 & 0 & 0 & 0 & 0 \\ 0 & 0 & 0 & 0 & 0 & 0 \\ 0 & 0 & 0 & 0 & 0 & 0 \\ 0 & 0 & 0 & 0 & 1 & 0 \\ 0 & 0 & 0 & 0 & 0 & 1 \end{bmatrix}$$

$$J_1 = \begin{bmatrix} 0 & -1 & 0 & 0 & 0 & 0 \\ 1 & 0 & 0 & 0 & 0 & 0 \\ 0 & 0 & 0 & 0 & 0 & 0 \\ 0 & 0 & 0 & 0 & 0 & 0 \\ 0 & 0 & 0 & 0 & 0 & 0 \\ 0 & 0 & 0 & 0 & 0 & 0 \end{bmatrix}, \quad J_2 = \begin{bmatrix} 0 & 0 & 0 & 0 & 0 & 0 \\ 0 & 0 & 0 & 0 & 0 & 0 \\ 0 & 0 & 0 & -1 & 0 & 0 \\ 0 & 0 & 1 & 0 & 0 & 0 \\ 0 & 0 & 0 & 0 & 0 & 0 \\ 0 & 0 & 0 & 0 & 0 & 0 \end{bmatrix}, \quad J_3 = \begin{bmatrix} 0 & 0 & 0 & 0 & 0 & 0 \\ 0 & 0 & 0 & 0 & 0 & 0 \\ 0 & 0 & 0 & 0 & 0 & 0 \\ 0 & 0 & 0 & 0 & 0 & 0 \\ 0 & 0 & 0 & 0 & 0 & -1 \\ 0 & 0 & 0 & 0 & 1 & 0 \end{bmatrix},$$

$$R = \begin{bmatrix} \cos\theta_1 & -\sin\theta_1 & 0 & 0 & 0 & 0 \\ \sin\theta_1 & \cos\theta_1 & 0 & 0 & 0 & 0 \\ 0 & 0 & \cos\theta_2 & -\sin\theta_2 & 0 & 0 \\ 0 & 0 & \sin\theta_2 & \cos\theta_2 & 0 & 0 \\ 0 & 0 & 0 & 0 & \cos\theta_3 & -\sin\theta_3 \\ 0 & 0 & 0 & 0 & \sin\theta_3 & \cos\theta_3 \end{bmatrix},$$

$$U = \begin{bmatrix} a_1 & -b_1 & 0 & 0 & 0 & 0 \\ b_1 & a_1 & 0 & 0 & 0 & 0 \\ 0 & 0 & a_2 & -b_2 & 0 & 0 \\ 0 & 0 & b_2 & a_2 & 0 & 0 \\ 0 & 0 & 0 & 0 & a_3 & -b_3 \\ 0 & 0 & 0 & 0 & b_3 & a_3 \end{bmatrix}, \quad U^{-1} = \begin{bmatrix} \frac{a_1}{a_1^2+b_1^2} & \frac{b_1}{a_1^2+b_1^2} & 0 & 0 & 0 & 0 \\ -\frac{b_1}{a_1^2+b_1^2} & \frac{a_1}{a_1^2+b_1^2} & 0 & 0 & 0 & 0 \\ 0 & 0 & \frac{a_2}{a_2^2+b_2^2} & \frac{b_2}{a_2^2+b_2^2} & 0 & 0 \\ 0 & 0 & -\frac{b_2}{a_2^2+b_2^2} & \frac{a_2}{a_2^2+b_2^2} & 0 & 0 \\ 0 & 0 & 0 & 0 & \frac{a_3}{a_3^2+b_3^2} & \frac{b_3}{a_3^2+b_3^2} \\ 0 & 0 & 0 & 0 & -\frac{b_3}{a_3^2+b_3^2} & \frac{a_3}{a_3^2+b_3^2} \end{bmatrix}.$$

## G. Supplementary Details on the Matching Algorithm

### G.1. Lemmas and Proofs for the Algorithm

**Lemma G.1.** *Given matrices $X, X', Y, Y' \in \mathbb{R}^{m \times n}$ with $m \geq n$, find a matrix $A \in GL(n)$ that minimizes the following objective function:*

$$f(A) = \|X - X'A^\top\|_F^2 + \|Y - Y'A^{-1}\|_F^2, \tag{157}$$

*where $\|\cdot\|_F$ denotes the Frobenius norm, and $GL(n)$ is the general linear group of invertible $n \times n$ matrices. Then $\frac{\partial f}{\partial A}$, the gradient w.r.t $A$, is:*

$$2(AX'^\top - X^\top)X' + 2(A^{-1})^T Y'^T (Y - Y'A^{-1})(A^{-1})^T. \tag{158}$$

*Proof.* We adopt the matrix convention where the gradient is represented as a column vector. We aim to find an invertible matrix $A \in GL(n)$ that minimizes the objective function

$$f(A) = \|X - X'A^\top\|_F^2 + \|Y - Y'A^{-1}\|_F^2, \tag{159}$$

where $X, X', Y, Y' \in \mathbb{R}^{m \times n}$ with $m \geq n$, and $\|\cdot\|_F$ denotes the Frobenius norm.

**Step 1. Express the Objective Function Using the Trace**

Since the Frobenius norm satisfies $\|M\|_F^2 = \text{trace}(M^T M)$, we can write

$$\begin{aligned} f(A) &= \|X - X'A^\top\|_F^2 + \|Y - Y'A^{-1}\|_F^2 \\ &= \text{trace}\left((X - X'A^\top)^\top(X - X'A^\top)\right) + \text{trace}\left((Y - Y'A^{-1})^\top(Y - Y'A^{-1})\right). \end{aligned} \tag{160}$$

**Step 2. Compute the Gradient**

To find the minimum, we compute the gradient of $f(A)$ with respect to $A$ and set it to zero. Define $g_1(A) = \|X - X'A^\top\|_F^2$ and $g_2(A) = \|Y - Y'A^{-1}\|_F^2$, so $f(A) = g_1(A) + g_2(A)$. To compute the gradient of $g_1(A) = \|X - X'A^\top\|_F^2$ with respect to $A$, we first expand the expression using the trace property $\|M\|_F^2 = \text{trace}(M^\top M)$:

$$\begin{aligned} g_1(A) &= \text{trace}\left((X - X'A^\top)^\top(X - X'A^\top)\right) = \text{trace}\left((X^\top - AX'^\top)(X - X'A^\top)\right) \\ &= \text{trace}\left(X^\top X - X^\top X'A^\top - AX'^\top X + AX'^\top X'A^\top\right) \\ &= \text{trace}(X^\top X) - 2\text{trace}(X^\top X'A^\top) + \text{trace}(AX'^\top X'A^\top). \end{aligned} \tag{161}$$

Now, we compute the gradient of each term with respect to $A$:

$$\frac{\partial}{\partial A}\text{trace}(X^\top X) = 0, \qquad \frac{\partial}{\partial A}(-2\text{trace}(X^\top X'A^\top)) = -2X^\top X', \qquad \frac{\partial}{\partial A}\text{trace}(AX'^\top X'A^\top) = 2AX'^\top X'. \tag{162}$$

Summing these results, we obtain the gradient of $g_1(A)$:

$$\frac{\partial g_1(A)}{\partial A} = 0 - 2X^\top X' + 2AX'^\top X' = 2(AX'^\top - X^\top)X'. \tag{163}$$

For the second term $g_2(A)$, since it involves $A^{-1}$, we use the differential. Note that $d(A^{-1}) = -A^{-1}dA A^{-1}$. The differential of $g_2(A)$ is

$$dg_2 = d\left[\text{trace}\left((Y - Y'A^{-1})^T(Y - Y'A^{-1})\right)\right] = 2\text{trace}\left((Y - Y'A^{-1})^T d(Y - Y'A^{-1})\right)$$

$$= 2\text{trace}\left((Y - Y'A^{-1})^T Y'(A^{-1}dAA^{-1})\right) = 2\text{trace}\left((Y - Y'A^{-1})^T Y'A^{-1}dAA^{-1}\right). \tag{164}$$

Using the cyclic property of the trace, $\text{trace}(PQRS) = \text{trace}(SPQR)$, we adjust the expression:

$$dg_2 = 2\text{trace}\left((Y - Y'A^{-1})^T Y'A^{-1}dAA^{-1}\right) = 2\text{trace}\left(A^{-1}(Y - Y'A^{-1})^T Y'A^{-1}dA\right). \tag{165}$$

Since $dg_2 = \text{trace}\left(\left(\frac{\partial g_2}{\partial A}\right)^\top dA\right)$, we identify

$$\frac{\partial g_2}{\partial A} = (2A^{-1}(Y - Y'A^{-1})^T Y'A^{-1})^T = 2(A^{-1})^T Y'^T(Y - Y'A^{-1})(A^{-1})^T. \tag{166}$$

Thus, the total gradient of $f(A)$ is

$$\frac{\partial f}{\partial A} = \frac{\partial g_1}{\partial A} + \frac{\partial g_2}{\partial A} = 2(AX'^\top - X^\top)X' + 2(A^{-1})^T Y'^T(Y - Y'A^{-1})(A^{-1})^T. \tag{167}$$

This completes the proof. $\square$

**Lemma G.2.** *Given matrices $X, X', Y, Y' \in \mathbb{R}^{m \times n}$ with $m \geq n$, the orthogonal matrix $A \in \mathbb{R}^{n \times n}$ satisfying $A^\top A = I$ that minimizes the objective function:*

$$f(A) = \|X - X'A^\top\|_F^2 + \|Y - Y'A^{-1}\|_F^2, \tag{168}$$

*is $A = UV^\top$, where $U, \Sigma, V$ are from the SVD of $B = U\Sigma V^\top$, with $B = X^\top X' + Y^\top Y'$.*

*Proof.* Since $A$ is orthogonal, $A^{-1} = A^\top$, so the objective can be rewritten as:

$$f(A) = \|X - X'A^\top\|_F^2 + \|Y - Y'A^\top\|_F^2. \tag{169}$$

**Step 1.**

The Frobenius norm squared is $\|M\|_F^2 = \text{trace}(M^\top M)$. We expand the first term of $f(A)$:

$$\begin{aligned}
\|X - X'A^\top\|_F^2 &= \text{trace}\left((X - X'A^\top)^\top(X - X'A^\top)\right) = \text{trace}\left((X^\top - AX'^\top)(X - X'A^\top)\right) \\
&= \text{trace}\left(X^\top X - X^\top X'A^\top - AX'^\top X + AX'^\top X'A^\top\right) \\
&= \text{trace}(X^\top X) - 2\text{trace}(X^\top X'A^\top) + \text{trace}(AX'^\top X'A^\top).
\end{aligned} \tag{170}$$

For an orthogonal matrix $A$, since $A^\top A = AA^\top = I$ and the trace is invariant under cyclic permutations, we have:

$$\text{trace}(A^\top X'^\top X'A) = \text{trace}(X'^\top X'AA^\top) = \text{trace}(X'^\top X'). \tag{171}$$

Thus,

$$\|X - X'A^\top\|_F^2 = \text{trace}(X^\top X) - 2\text{trace}(X^\top X'A^\top) + \text{trace}(X'^\top X'). \tag{172}$$

Similarly, for the second term regarding $Y$:

$$\|Y - Y'A^\top\|_F^2 = \text{trace}(Y^\top Y) - 2\text{trace}(Y^\top Y'A^\top) + \text{trace}(Y'^\top Y'). \tag{173}$$

Substituting into $f(A)$:

$$f(A) = \text{trace}(X^\top X) + \text{trace}(X'^\top X') + \text{trace}(Y^\top Y) + \text{trace}(Y'^\top Y') - 2\left(\text{trace}(X^\top X'A^\top) + \text{trace}(Y^\top Y'A^\top)\right). \tag{174}$$

The terms $\text{trace}(X^\top X)$, $\text{trace}(X'^\top X')$, $\text{trace}(Y^\top Y)$, and $\text{trace}(Y'^\top Y')$ are constant with respect to $A$. Thus, minimizing $f(A)$ is equivalent to maximizing:

$$g(A) = \text{trace}(X^\top X'A^\top) + \text{trace}(Y^\top Y'A^\top). \tag{175}$$

Using the linearity of the trace and the property $\text{trace}(MN) = \text{trace}(NM)$:

$$g(A) = \text{trace}(A^\top(X^\top X' + Y^\top Y')). \tag{176}$$

Define $B = X^\top X' + Y^\top Y'$. Then, the problem reduces to maximizing $\text{trace}(A^\top B)$ over all orthogonal matrices $A$.

**Step 2.**

Compute the singular value decomposition of $B = U\Sigma V^\top$, where $U, V \in \mathbb{R}^{n \times n}$ are orthogonal matrices, and $\Sigma = \text{diag}(\sigma_1, \ldots, \sigma_n)$ is a diagonal matrix with non-negative singular values $\sigma_i \geq 0$. Then,

$$\text{trace}(A^\top B) = \text{trace}(A^\top U\Sigma V^\top) = \text{trace}(V^\top A^\top U\Sigma). \tag{177}$$

Define $C = V^\top A^\top U$. Since $A, U, V$ are orthogonal, $C$ is also an orthogonal matrix. Thus, $\text{trace}(C\Sigma) = \sum_{i=1}^{n} c_{ii}\sigma_i$, where $c_{ii}$ are the diagonal elements of $C$. Since $C$ is orthogonal, its columns (and rows) are orthonormal vectors, which implies $|c_{ii}| \leq 1$ for all $i$. Therefore, $\text{trace}(C\Sigma) \leq \sum_{i=1}^{n} \sigma_i$, with equality when $C = I$, i.e., $c_{ii} = 1$ for all $i$ (assuming all $\sigma_i \geq 0$).

**Step 3.**

The maximum value of $\text{trace}(A^\top B)$ is $\sum_{i=1}^{n} \sigma_i$, achieved when $C = I$, or $V^\top A^\top U = I$. This is equivalent to $A = UV^\top$. Since $A = UV^\top$ maximizes $g(A)$, and $f(A)$ is of the form $\text{constant} - 2g(A)$, this choice of $A$ minimizes $f(A)$. This completes the proof. $\qquad\square$

**Lemma G.3** (Optimal Alignment for RoPE Query-Key Matrices). *Let $W_Q^a, W_K^a \in \mathbb{R}^{d \times d_h}$ and $W_Q^b, W_K^b \in \mathbb{R}^{d \times d_h}$ be the query and key weight matrices for a single attention head from two models, denoted $a$ and $b$. The problem of finding an alignment matrix $U \in H(d_h)$ that minimizes the loss function*

$$\mathcal{L}_{Q,K}(U) = \left\|W_Q^a - W_Q^b U^\top\right\|_F^2 + \left\|W_K^a - W_K^b U^{-1}\right\|_F^2 \tag{178}$$

*over $U \in H(d_h)$ decouples into $d_h/2$ independent subproblems. For each subspace $j = 1, \ldots, d_h/2$, the subproblem of finding the optimal $2 \times 2$ matrix $U_j$ is equivalent to finding the minimizer $x_j^\star = \arg\min_{x>0} g_j(x)$ of the 1D scalar objective function*

$$g_j(x) = x\,\eta_{Q,j} + \frac{\eta_{K,j}}{x} - 4\sqrt{|\gamma_{Q,j}|^2 x + \frac{|\gamma_{K,j}|^2}{x} + 2\text{Re}(\gamma_{Q,j}\bar{\gamma}_{K,j})}, \tag{179}$$

*where the constants $\eta_{Q,j}, \eta_{K,j}, \gamma_{Q,j}, \gamma_{K,j}$ are derived from the corresponding weight submatrices as defined in the proof below (with $\eta$ denoting squared Frobenius norms and $\gamma$ denoting complex correlation scalars). The optimal matrix $U_j^\star$ is then determined by the optimal value $x_j^\star$.*

*Proof.* We proceed the proofs step-by-step.

**Step 1.**

The loss $\mathcal{L}_{Q,K}(U)$ decouples independently across the $d_h/2$ orthogonal 2D subspaces. For each subspace $j = 1, \ldots, d_h/2$, the corresponding loss term is

$$\mathcal{L}_{Q,K}^{(j)}(U_j) = \left\|Q_j^a - Q_j^b U_j^\top\right\|_F^2 + \left\|K_j^a - K_j^b U_j^{-1}\right\|_F^2, \tag{180}$$

where $Q_j^m = W_{Q,j}^m$ and $K_j^m = W_{K,j}^m \in \mathbb{R}^{d \times 2}$ are the submatrices for $m \in \{a, b\}$, and $U_j = \begin{pmatrix} a_j & -b_j \\ b_j & a_j \end{pmatrix} \in H(2)$. Each $\mathcal{L}_{Q,K}^{(j)}$ can be minimized independently. For simplicity, we omit the index $j$ in the following. The goal is to find the matrix $U = \begin{pmatrix} a & -b \\ b & a \end{pmatrix}$ that minimizes the loss:

$$\mathcal{L}(a, b) = \left\|Q^a - Q^b U^\top\right\|_F^2 + \left\|K^a - K^b U^{-1}\right\|_F^2. \tag{181}$$

**Step 2.**

The problem simplifies by identifying the matrix $U$ with a complex number $z = a + ib$. The squared magnitude is $r^2 = a^2 + b^2 = |z|^2 = \det(U)$. Key properties are $U^\top U = UU^\top = r^2 I$ and $U^{-1} = \frac{1}{r^2}U^\top$. Using the property

$\|M\|_F^2 = \mathrm{tr}(M^\top M)$, we expand the loss function: By dropping the constant terms $\|Q^a\|_F^2 + \|K^a\|_F^2$, the objective to minimize is:

$$
\begin{aligned}
\mathcal{L} &= -2\mathrm{tr}((Q^a)^\top Q^b U^\top) + \mathrm{tr}(U(Q^b)^\top Q^b U^\top) - 2\mathrm{tr}((K^a)^\top K^b U^{-1}) + \mathrm{tr}((U^{-1})^\top (K^b)^\top K^b U^{-1}) \\
&= -2\mathrm{tr}(C_Q U^\top) + r^2 \|Q^b\|_F^2 - 2\mathrm{tr}(C_K U^{-1}) + \frac{1}{r^2}\|K^b\|_F^2 = r^2 \eta_Q + \frac{\eta_K}{r^2} - 2\mathrm{tr}(C_Q U^\top) - \frac{2}{r^2}\mathrm{tr}(C_K U^\top),
\end{aligned}
\tag{182}
$$

where the constants are defined as $\eta_Q = \|Q^b\|_F^2$, $\eta_K = \|K^b\|_F^2$, $C_Q = (Q^a)^\top Q^b$, and $C_K = (K^a)^\top K^b$. To express the trace terms in complex form, note that $U^\top = aI - bJ$. This yields the identity $\mathrm{tr}(CU^\top) = a\mathrm{tr}(C) - b\mathrm{tr}(CJ) = 2\mathrm{Re}(\gamma z)$, where the complex scalar $\gamma = \frac{1}{2}(\mathrm{tr}(C) + i\mathrm{tr}(CJ))$. Applying this, the loss becomes:

$$
\mathcal{L}(z) = |z|^2 \eta_Q + \frac{\eta_K}{|z|^2} - 4\mathrm{Re}(\gamma_Q z) - \frac{4}{|z|^2}\mathrm{Re}(\gamma_K z),
\tag{183}
$$

where $\gamma_Q$ and $\gamma_K$ are complex constants derived from $C_Q$ and $C_K$ respectively. Express $z$ in polar form as $z = re^{i\theta}$, where $r = |z| > 0$. The loss function can be rewritten to isolate terms dependent on the phase angle $\theta$:

$$
\mathcal{L}(r, \theta) = r^2 \eta_Q + \frac{\eta_K}{r^2} - 4\mathrm{Re}\left(\left(r\gamma_Q + \frac{1}{r}\gamma_K\right)e^{i\theta}\right).
\tag{184}
$$

First, optimize the phase $\theta$ for a fixed magnitude $r$. The expression is minimized by maximizing the real part term. The maximum value of $\mathrm{Re}(Ce^{i\theta})$ is $|C|$, achieved when $e^{i\theta}$ has angle $-\arg(C)$. Thus, the optimal phase $\theta^\star$ for a given $r$ is:

$$
\theta^\star(r) = -\arg\left(r\gamma_Q + \frac{1}{r}\gamma_K\right).
\tag{185}
$$

Substituting $\theta^\star$ back into the loss yields a 1D scalar objective function depending only on $r$:

$$
g(r) = r^2 \eta_Q + \frac{\eta_K}{r^2} - 4\left|r\gamma_Q + \frac{1}{r}\gamma_K\right|.
\tag{186}
$$

For algebraic convenience, substitute $x = r^2 > 0$. The squared norm term expands as:

$$
\left|r\gamma_Q + \frac{1}{r}\gamma_K\right|^2 = \left(\sqrt{x}\gamma_Q + \frac{1}{\sqrt{x}}\gamma_K\right)\left(\sqrt{x}\bar{\gamma}_Q + \frac{1}{\sqrt{x}}\bar{\gamma}_K\right) = x|\gamma_Q|^2 + \frac{1}{x}|\gamma_K|^2 + 2\mathrm{Re}(\gamma_Q \bar{\gamma}_K).
\tag{187}
$$

Letting $A = |\gamma_Q|^2$, $B = |\gamma_K|^2$, and $C = 2\mathrm{Re}(\gamma_Q \bar{\gamma}_K)$, the objective function in terms of $x$ is:

$$
g(x) = x\eta_Q + \frac{\eta_K}{x} - 4\sqrt{Ax + \frac{B}{x} + C}.
\tag{188}
$$

**Step 3.**

To minimize $g(x)$ for $x > 0$, find stationary points by solving $g'(x) = 0$:

$$
g'(x) = \eta_Q - \frac{\eta_K}{x^2} - \frac{2\left(A - \frac{B}{x^2}\right)}{\sqrt{Ax + \frac{B}{x} + C}} = 0.
\tag{189}
$$

Isolating the square root term and square both sides, we have

$$
\left(\eta_Q - \frac{\eta_K}{x^2}\right)^2 = \frac{4\left(A - \frac{B}{x^2}\right)^2}{Ax + \frac{B}{x} + C}.
\tag{190}
$$

Multiplying by the denominator and clearing fractions by multiplying by $x^4$ yields:

$$
(\eta_Q x^2 - \eta_K)^2 (Ax^2 + Cx + B) = 4x(Ax^2 - B)^2.
\tag{191}
$$

The left side has degree 6 in $x$, while the right side has degree 5, so the stationarity condition corresponds to finding roots of a 6th-degree polynomial.

**Step 4.**

Since solving a 6th-degree polynomial analytically is generally infeasible and numerical root-finding can be unstable, a more robust approach is to directly minimize the scalar function $g(x)$ using a 1D optimization method. The procedure is as follows:

1. Compute the scalar constants $\eta_Q, \eta_K$ and the complex constants $\gamma_Q, \gamma_K$.

2. Define the objective function $g(x) = x\eta_Q + \frac{\eta_K}{x} - 4\sqrt{|\gamma_Q|^2 x + \frac{|\gamma_K|^2}{x} + 2\mathrm{Re}(\gamma_Q\bar{\gamma}_K)}$.

3. Find the minimizer $x^\star = \arg\min_{x>0} g(x)$ using a numerical optimization routine. Here we use the Brent's method (Brent, 2013). The optimal solution is computed as $r^\star = \sqrt{x^\star}$, $\theta^\star = -\arg\left(r^\star\gamma_Q + \frac{1}{r^\star}\gamma_K\right)$, and finally $a = r^\star\cos(\theta^\star)$, $b = r^\star\sin(\theta^\star)$.

This yields the optimal alignment matrix $U_j$ for each subspace $j$. □

### G.2. Algorithm Description

---

**Algorithm 1** Attention Layer Alignment

---

  **Input:** $\theta^A$, $\theta^B$.
  **Output:** Aligned $\theta^{B,\text{aligned}}$.
  % Stage 1: Head Permutation
  Compute cost matrix $C$.
  Solve LAP for $\pi^*$.
  Reorder $\theta^B \leftarrow \pi^*(\theta^B)$.
  % Stage 2: Internal Parameter Alignment
  **for** $i = 1$ to $h$ **do**
    % Align $Q, K$
    **if** standard MHA **then**
      Minimize $\mathcal{L}_{Q,K}(U_i)$ over $\mathrm{GL}(d_h)$.
    **else**
      Minimize $\mathcal{L}_{Q,K}(U_i)$ over $\mathrm{H}(d_h)$.
    **end if**
    Update: $W_{i,B}^Q \leftarrow W_{i,B}^Q U_i^\top$, $W_{i,B}^K \leftarrow W_{i,B}^K U_i^{-1}$.
    % Align $V, O$
    Minimize $\mathcal{L}_{V,O}(V_i)$ over $\mathrm{GL}(d_h)$.
    Update: $W_{i,B}^V \leftarrow W_{i,B}^V V_i^{-1}$, $W_{i,B}^O \leftarrow V_i W_{i,B}^O$.
  **end for**
  **return** $\theta^{B,\text{aligned}}$

---

## H. Impact of Attention Reinitialization on Pretrained Transformer Performance

We investigate the effect of targeted attention reinitialization on pretrained Transformer models. Unlike feedforward blocks, attention layers govern contextual interactions and strongly influence early representations. To assess their contribution, we reset the parameters of individual attention modules using standard initialization, while keeping embeddings, LayerNorms, and feedforward blocks fixed. Models are then evaluated directly on their pretrained tasks without fine-tuning. Our study considers ViT-Base on ImageNet-1K for image classification and GPT-2 on WikiText103 for language modeling, with performance measured in accuracy and perplexity, respectively. Figures 3 and 4 summarize the results across layers.

We find that reinitializing attention layers beyond the first generally leads to only modest degradation, whereas resetting the initial layer produces a pronounced drop in performance. This asymmetry indicates that early attention plays a uniquely

critical role in anchoring representations, while deeper layers remain more resilient due to residual connections and redundancy in the architecture. Based on these findings, subsequent experiments on linear mode connectivity focus on reinitializing the first attention layer, as it provides the most consistent and informative signal of model sensitivity.

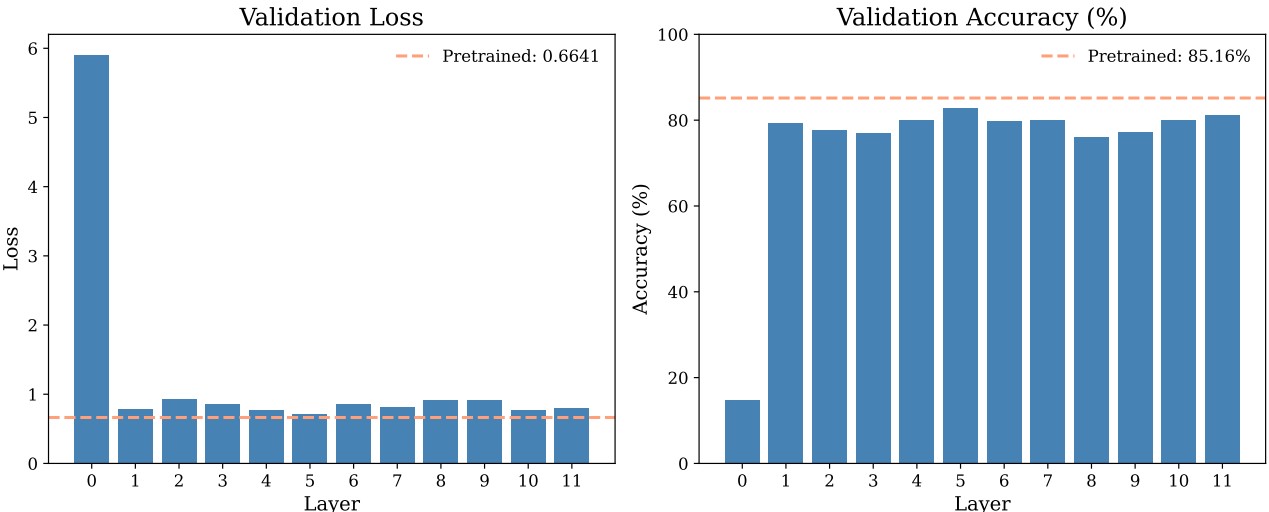

*Figure 3.* Performance degradation in ViT-Base on ImageNet due to attention reinitialization at different layers.

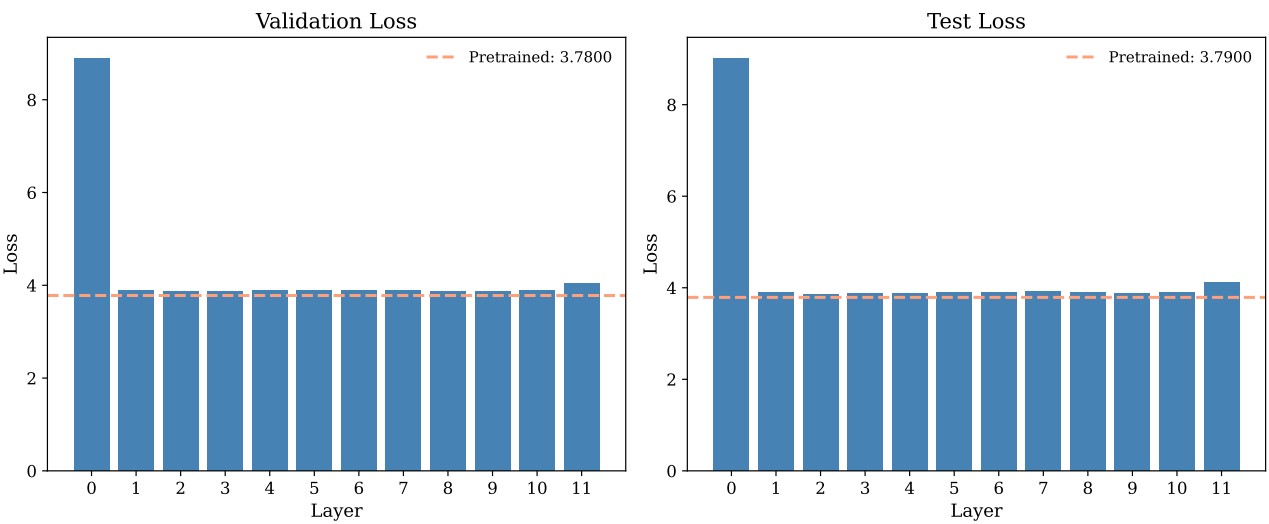

*Figure 4.* Effect of attention reinitialization on GPT-2 perplexity across layers on WikiText103.

## I. Experimental Details and Hyperparameters

Our experiments assess Linear Mode Connectivity (LMC) across a broad spectrum of benchmarks in both vision and natural language processing. The vision suite covers MNIST, CIFAR-10, CIFAR-100, ImageNet-1k, and transfer from ImageNet-21k to smaller classification datasets. For language, we include generative modeling with WikiText103, Enwik8, and the One Billion Word benchmark, together with supervised classification tasks such as AGNews, IMDB reviews, and DBpedia. Each experiment builds on pretrained Transformer architectures, where the core weights remain fixed and only selected attention modules are re-initialized for fine-tuning. Vision tasks use Vision Transformer (ViT) backbones, autoregressive language modeling relies on GPT-2, and text classification tasks are handled by BERT.

**AGNews.** For the AGNews dataset, we adopt a compact BERT-style encoder with embedding dimension 96, hidden size

384, and vocabulary size 15,000. Models are trained with depths of 2 or 6 layers and attention configurations of 4 or 8 heads. Pretraining is carried out using the Adam optimizer with a batch size of 512 and learning rate $1 \cdot 10^{-3}$, for up to 6 epochs until convergence.

**IMDBreview.** For the IMDB dataset, we adopt a compact BERT-style encoder with embedding dimension 96, hidden size 384, and vocabulary size 15,000. Models are trained with depths of 2, or 6 layers and attention configurations of 4 or 8 heads. Pretraining is performed using the Adam optimizer with a batch size of 128 and learning rate $3 \cdot 10^{-4}$, for up to 7 epochs until convergence.

**DBPedia.** For the DBPedia dataset, we adopt a compact BERT-style encoder with embedding dimension 96, hidden size 384, vocabulary size 30,522, and 219 output classes (max sequence length 256). Models are trained with depths of 2 or 6 layers and attention configurations of 4 or 8 heads. Pretraining is carried out using the Adam optimizer with a batch size of 256 and learning rate $1 \times 10^{-3}$ under a linear decay schedule, for up to 5 epochs until convergence.

**Enwik8.** For the Enwik8 dataset, we employ a GPT-2 style Transformer with 12 layers, hidden size of 512, 8 attention heads, and an intermediate size of 2048. The context length is set to 512 tokens, with memory length 512 and evaluation length 128. Pretraining is performed using the Adam optimizer with a batch size of 24 and an initial learning rate of $2.5 \cdot 10^{-4}$, following a cosine decay schedule without warmup, for a total of 60000 steps. During fine-tuning, we replace the pretrained attention modules with variants containing 4, 8, or 16 heads, and train for 60000 steps.

**WikiText103.** For the WikiText103 benchmark, we adopt a GPT-2 style Transformer with 12 layers, hidden size of 192, 3 attention heads, and an intermediate size of 768. The model uses learned attention biases, with context length, memory length, and evaluation length all set to 256 tokens. Training is conducted with the Adam optimizer using a batch size of 64 and an initial learning rate of $2.5 \cdot 10^{-4}$. A linear warmup of 2000 steps is followed by a cosine decay learning rate schedule. The pretraining phase runs for 60k steps. For fine-tuning, we replace the attention modules with variants containing 2, 3, or 4 heads, and train each configuration for 60000 steps.

**One Billion Word.** For the One Billion Word benchmark, we employ a GPT-2 style Transformer-based language model with sinusoidal positional embeddings, 12 layers, hidden size of 768, 12 attention heads, and an intermediate size of 3072. The vocabulary size is 793,470. Pretraining is performed with target sequence length 256, memory length 256, and evaluation sequence length 256. The model is trained using Adam with a batch size of 96, an initial learning rate of $2.5 \cdot 10^{-4}$, and a cosine decay learning rate schedule with 2000 warmup steps. Training is run for 500000 steps with random seed fixed at 0 for reproducibility. For fine-tuning, we replace the attention mechanism with variants containing 8, 12, or 16 heads. Each configuration is fine-tuned for 100000 steps.

**MNIST.** For the MNIST dataset, we adopt a lightweight Vision Transformer with patch size 7, embedding dimension 16, hidden size 64, and depths of 1 or 2 layers paired with 4 or 8 attention heads. Pretraining is carried out using the Adam optimizer with a learning rate of $5 \times 10^{-3}$, training to validation convergence (typically 60–80 epochs, depending on configuration).

**CIFAR-10.** For CIFAR-10, we use a Vision Transformer with patch size 4, embedding dimension 128, hidden size 512, and depths of 2, 4, or 6 layers paired with 4 or 8 attention heads. Images are normalized with CIFAR-10 statistics and augmented using random resized crop, horizontal flip, and rotation. Pretraining is performed with the Adam optimizer at a learning rate of $5 \times 10^{-3}$ for 100 epochs with batch size 100.

**CIFAR-100.** For CIFAR-100, we adopt a Vision Transformer with patch size 4, embedding dimension 128, hidden size 512, and depths of 6 layers paired with 4 or 8 attention heads. Images are normalized using standard CIFAR-100 statistics and augmented with random resized crop, horizontal flip, and rotation. Pretraining is conducted with the Adam optimizer at a learning rate of $5 \times 10^{-3}$ for 100 epochs and batch size 100.

**Imagenet21k→CIFAR10.** We adopt the ViT-Small-Patch16-224 model, pretrained on ImageNet-21k and subsequently fine-tuned on CIFAR-10. The model consists of 12 layers, a hidden size of 384, an MLP size of 1536, and 6 attention heads, resulting in approximately 22.2M parameters. It employs a patch size and stride of 16. Dropout is disabled (set to 0.0), and the activation function is `gelu`. Stochastic Gradient Descent (SGD) is employed during fine-tuning.

**Imagenet21k→CIFAR100.** We adopt the ViT-Small-Patch16-224 model, pretrained on ImageNet-21k and subsequently fine-tuned on CIFAR-100. The model consists of 12 layers, a hidden size of 384, an MLP size of 1536, and 6 attention heads, resulting in approximately 22.2M parameters. It employs a patch size and stride of 16. Dropout is disabled (set to 0.0), and the activation function is `gelu`. Stochastic Gradient Descent (SGD) is employed during fine-tuning.

**ImageNet-1k.** For ImageNet-1k, we utilize a pretrained Vision Transformer with the following configuration: hidden size of 768, 12 Transformer layers, 12 attention heads, and an intermediate size of 3072. Training is performed for 300 epochs with a batch size of 256 using the Adam optimizer and an initial learning rate of $5 \cdot 10^{-4}$. The learning rate follows a cosine decay schedule with 5 epochs of linear warmup. The gelu activation function is employed throughout the network, and both attention and hidden dropout rates are set to 0.0. During fine-tuning, we systematically replace the pretrained attention layers with variants containing 8, 12, or 16 heads. Depending on the number of re-initialized layers, the fine-tuning budget is set to 30, 50, 100, or 300 epochs, respectively.

**Runtime Environment.** All experiments were executed on NVIDIA H100 GPUs with 80GB of memory. A single GPU was sufficient for every task, except for the One Billion Word benchmark, which required two GPUs. Since training was implemented in JAX, approximately 75% of the GPU memory (about 60GB) was pre-allocated by default. For data loading and preprocessing, the number of CPU workers was limited to 10. In terms of wall-clock time, small-scale benchmarks– including MNIST, CIFAR-10, CIFAR-100, transfer learning from ImageNet-21k, and text classification datasets (AGNews, IMDB reviews, DBPedia)–each completed in under 30 minutes. For language modeling, both WikiText103 and Enwik8 required about 2 hours for pretraining and fine-tuning. The One Billion Word benchmark was more computationally demanding, requiring up to 2 days. On the vision side, ImageNet-1k fine-tuning could take as long as 6 days, depending on the configuration.

## J. Experiments

### J.1. Linear Mode Connectivity for Attention First Layer

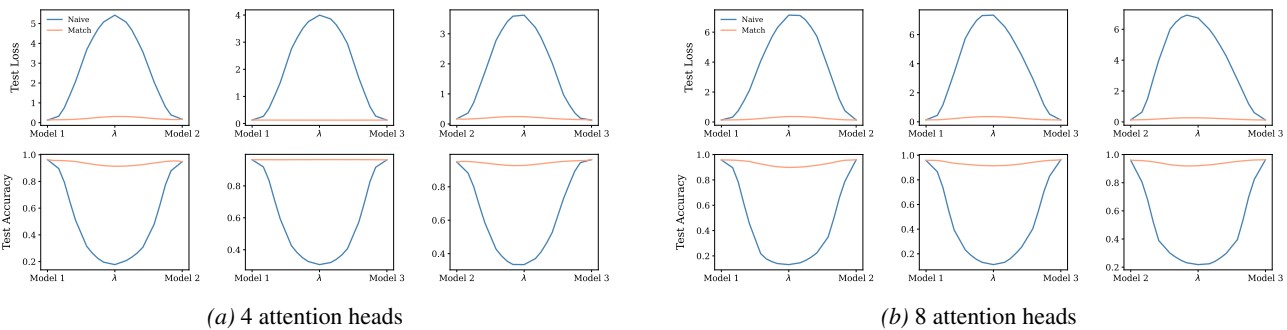

*(a)* 4 attention heads  *(b)* 8 attention heads

*Figure 5.* Linear Mode Connectivity for ViT on MNIST with 1 layer

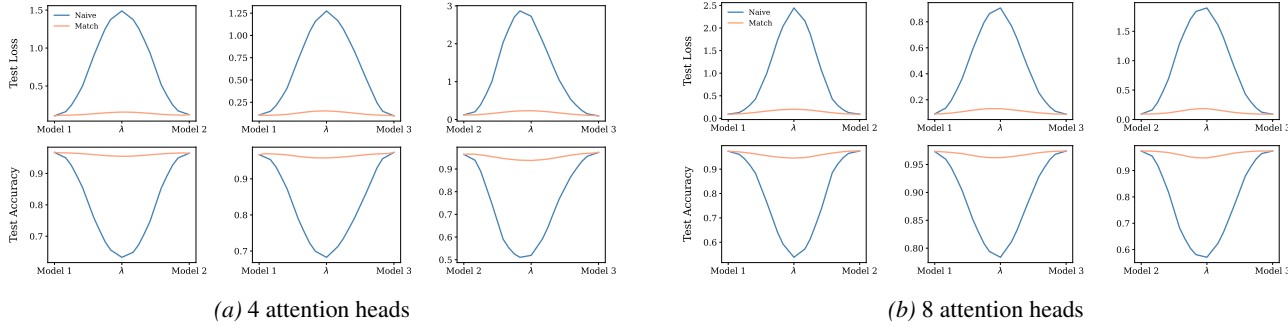

*(a)* 4 attention heads  *(b)* 8 attention heads

*Figure 6.* Linear Mode Connectivity for ViT on MNIST with 2 layers

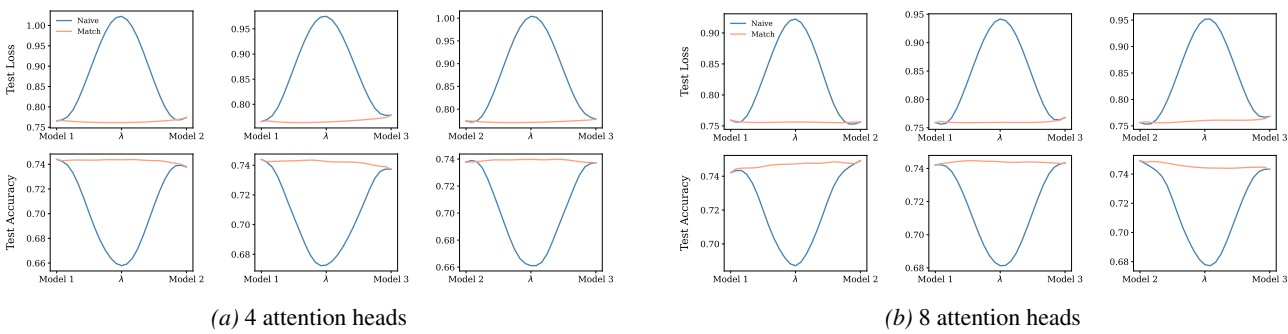

*(a)* 4 attention heads                    *(b)* 8 attention heads

*Figure 7.* Linear Mode Connectivity for ViT on CIFAR-10 with 2 layers

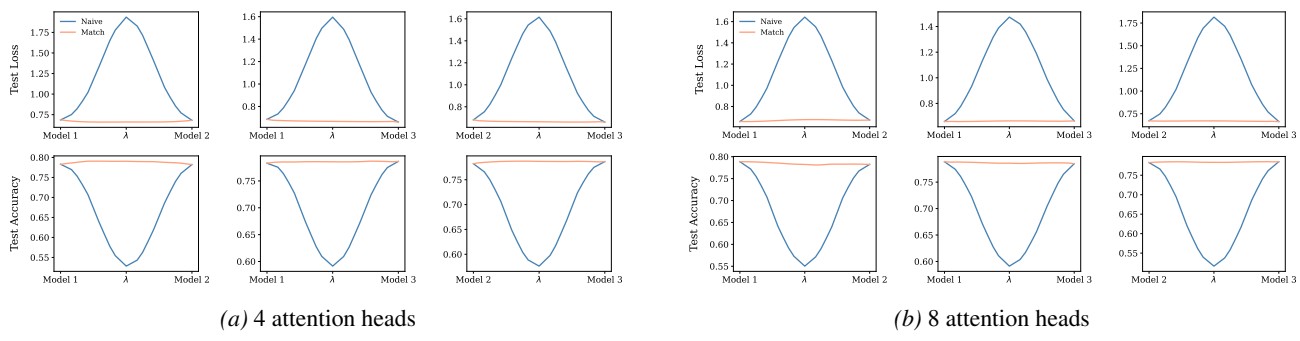

*(a)* 4 attention heads                    *(b)* 8 attention heads

*Figure 8.* Linear Mode Connectivity for ViT on CIFAR-10 with 4 layers

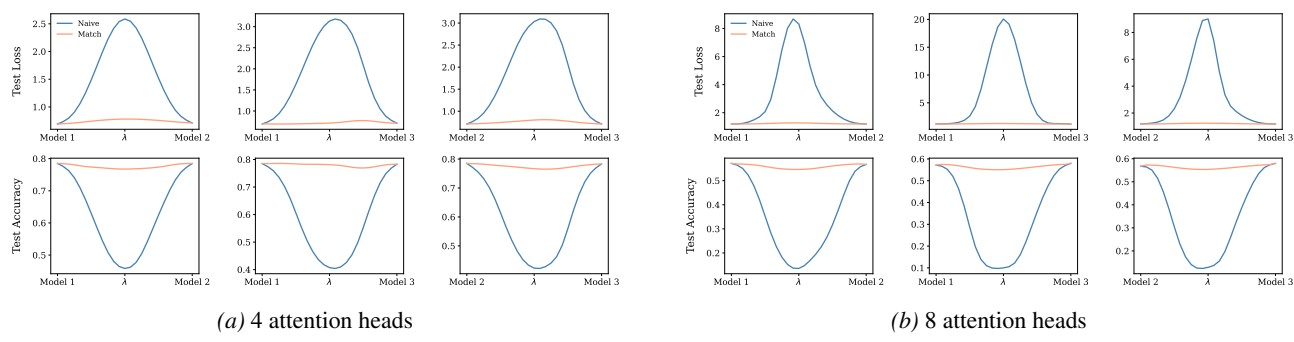

*(a)* 4 attention heads                    *(b)* 8 attention heads

*Figure 9.* Linear Mode Connectivity for ViT on CIFAR-10 with 6 layers

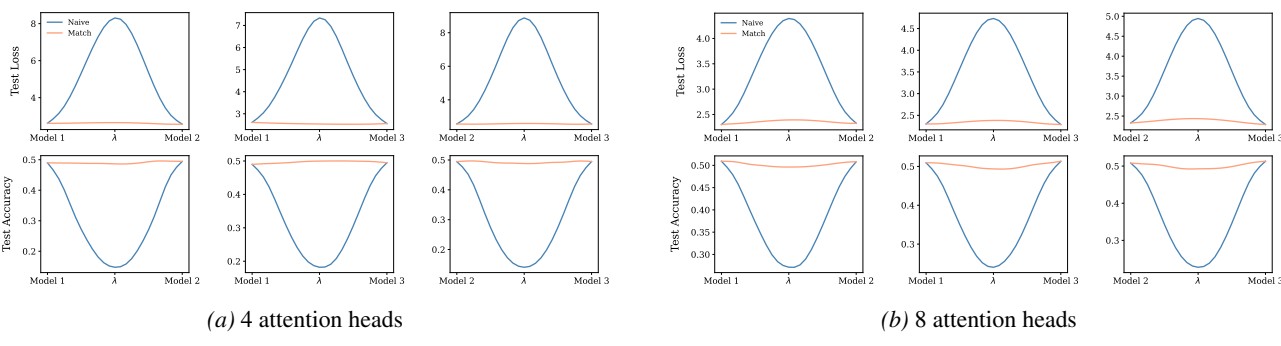

*(a)* 4 attention heads                    *(b)* 8 attention heads

*Figure 10.* Linear Mode Connectivity for ViT on CIFAR-100 with 6 layers

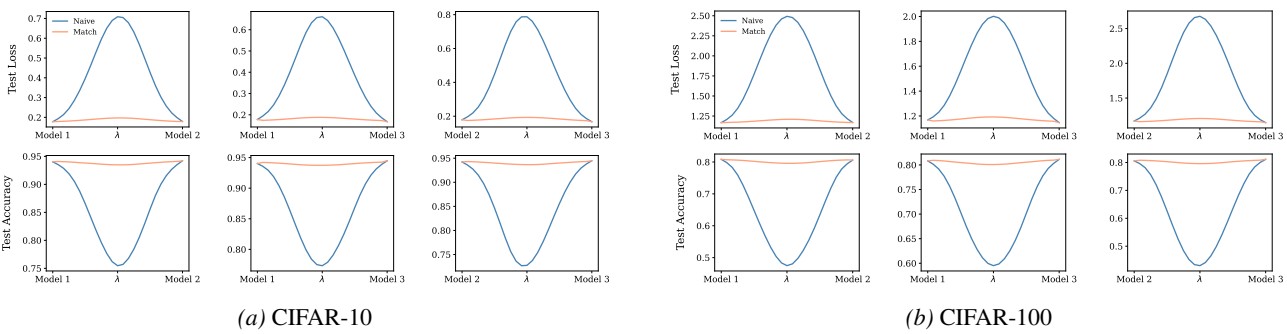

*(a)* CIFAR-10         *(b)* CIFAR-100

*Figure 11.* Linear Mode Connectivity for ViT on ImageNet21k→CIFAR-10/100 with 12 layers and 6 heads

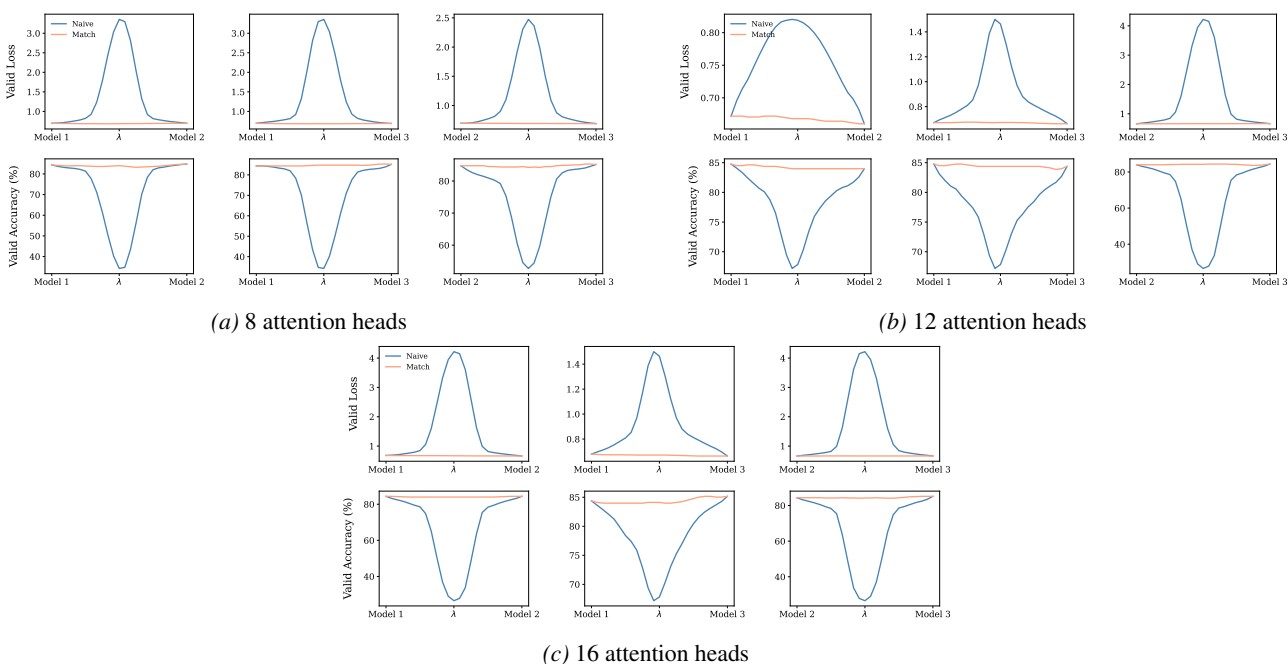

*(a)* 8 attention heads         *(b)* 12 attention heads

*(c)* 16 attention heads

*Figure 12.* Linear Mode Connectivity for ViT on ImageNet with 12 layers.

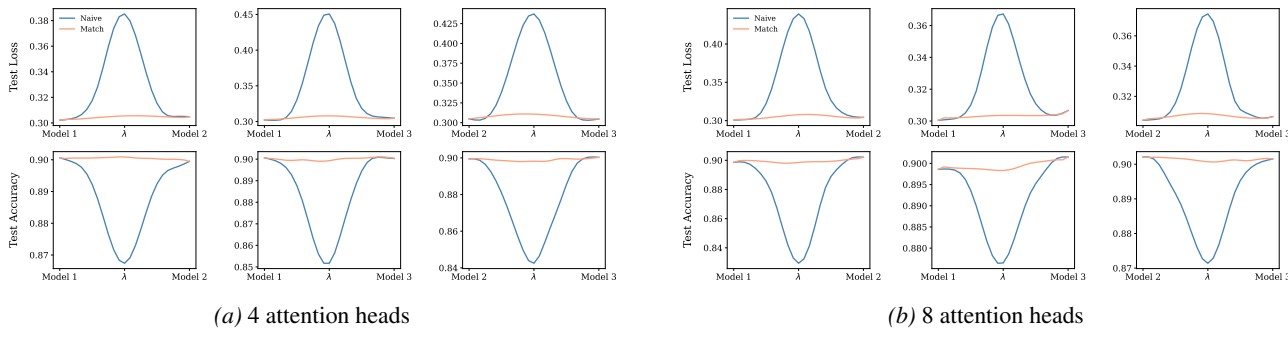

*(a)* 4 attention heads         *(b)* 8 attention heads

*Figure 13.* Linear Mode Connectivity for BERT on AGnews with 2 layers

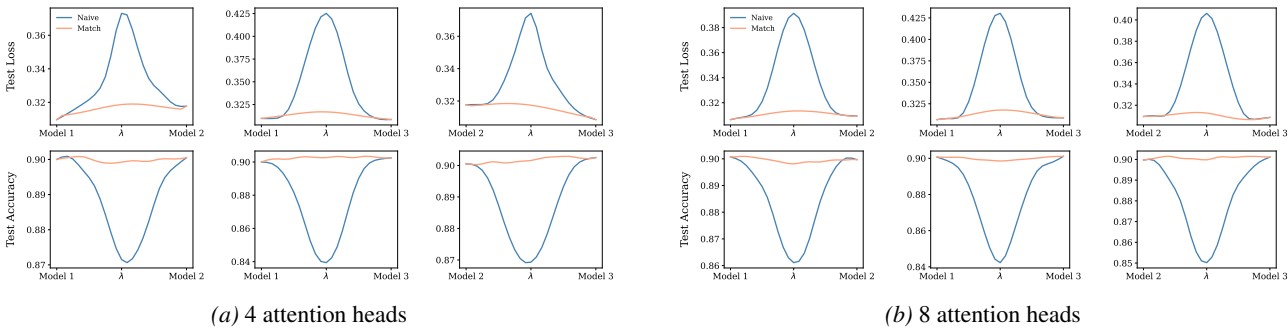

*(a)* 4 attention heads        *(b)* 8 attention heads

*Figure 14.* Linear Mode Connectivity for BERT on AGnews with 6 layers

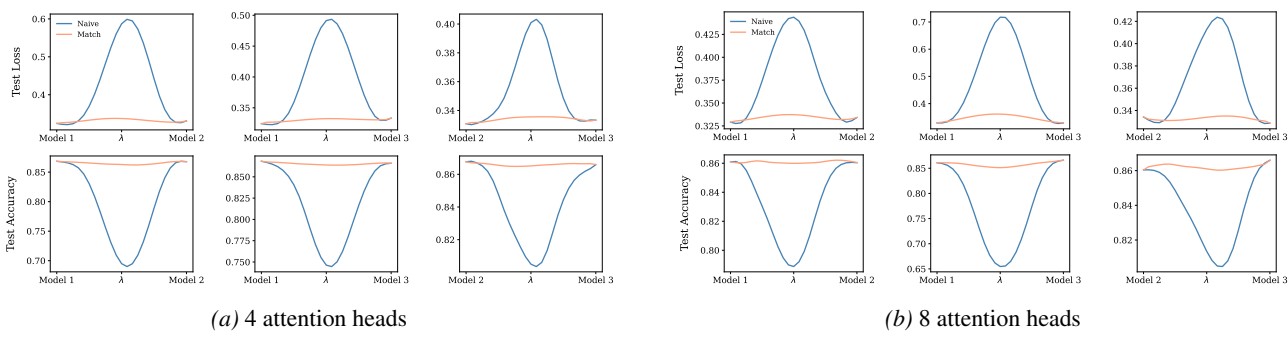

*(a)* 4 attention heads        *(b)* 8 attention heads

*Figure 15.* Linear Mode Connectivity for BERT on IMDBreview with 2 layers

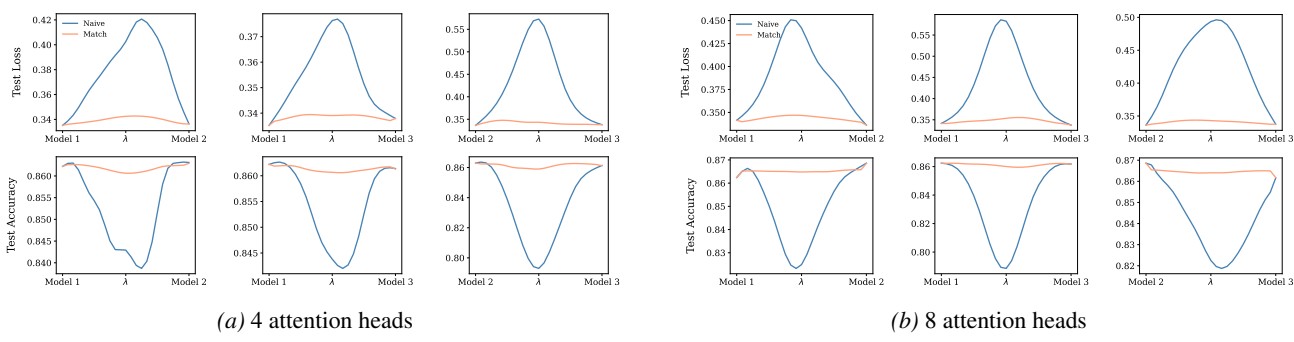

*(a)* 4 attention heads        *(b)* 8 attention heads

*Figure 16.* Linear Mode Connectivity for BERT on IMDBreview with 6 layers

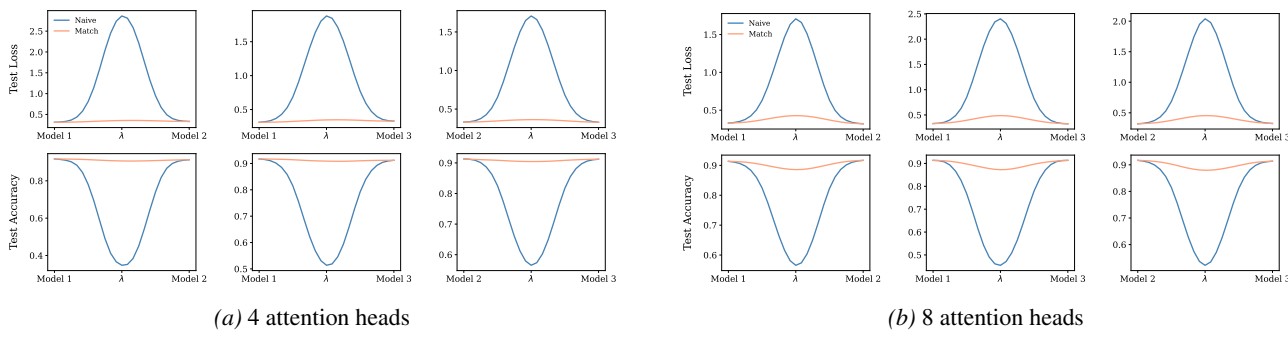

*(a)* 4 attention heads        *(b)* 8 attention heads

*Figure 17.* Linear Mode Connectivity for BERT on DBPedia with 2 layers

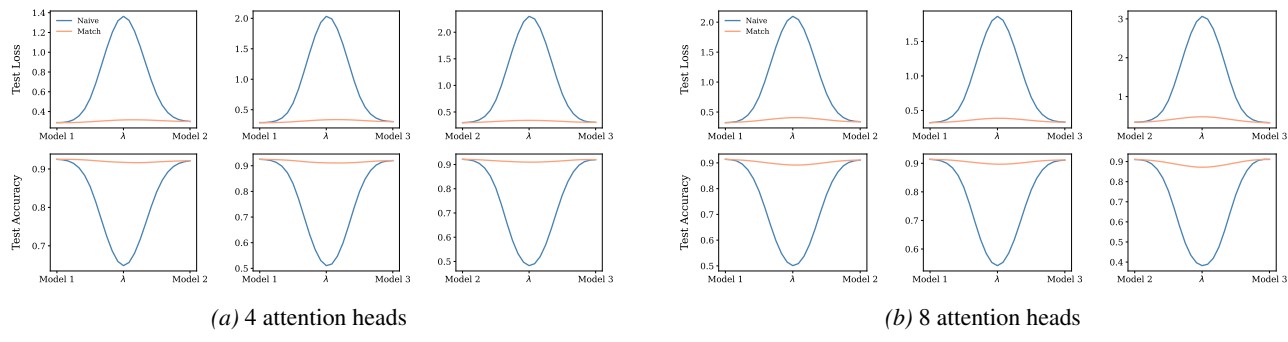

*(a)* 4 attention heads        *(b)* 8 attention heads

*Figure 18.* Linear Mode Connectivity for BERT on DBPedia with 6 layers

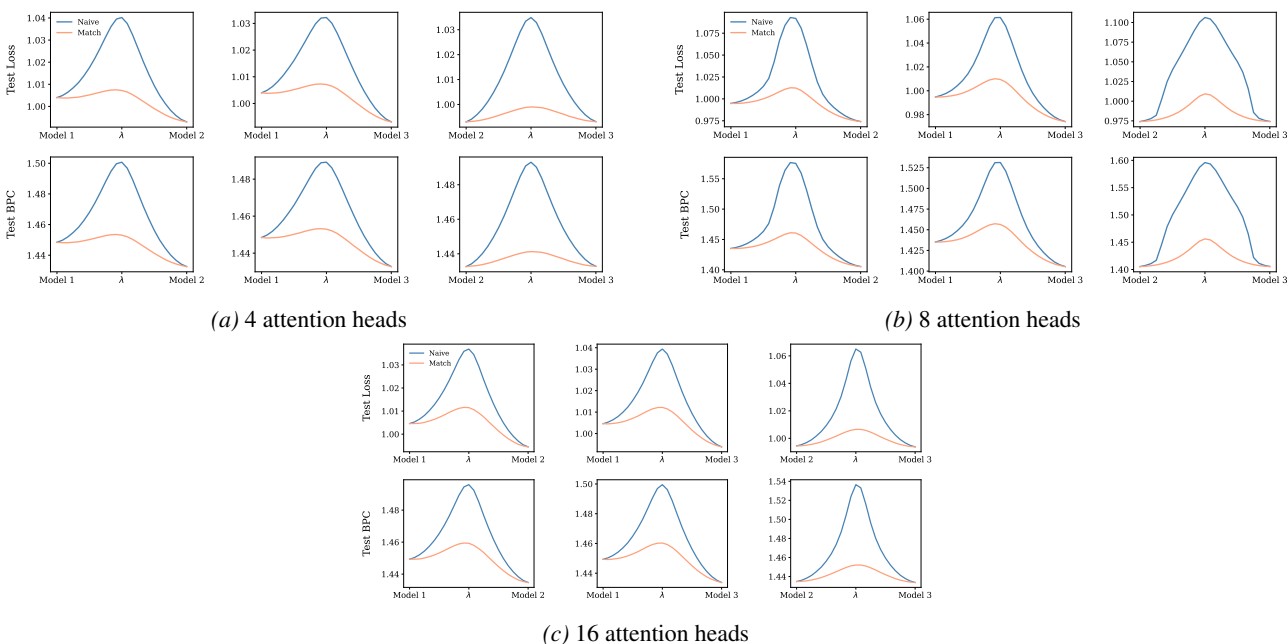

*(a)* 4 attention heads        *(b)* 8 attention heads

*(c)* 16 attention heads

*Figure 19.* Linear Mode Connectivity for GPT2 on Enwik8 with 12 layers.

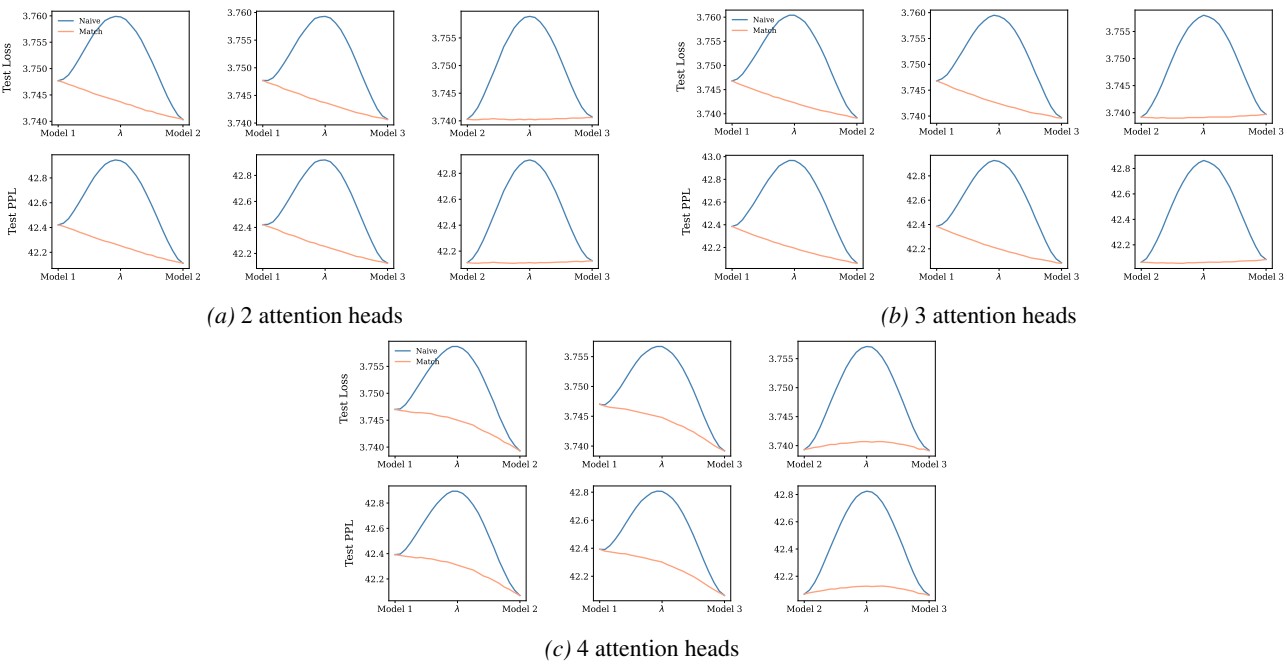

*Figure 20.* Linear Mode Connectivity for GPT2 on Wikitext103 with 12 layers.

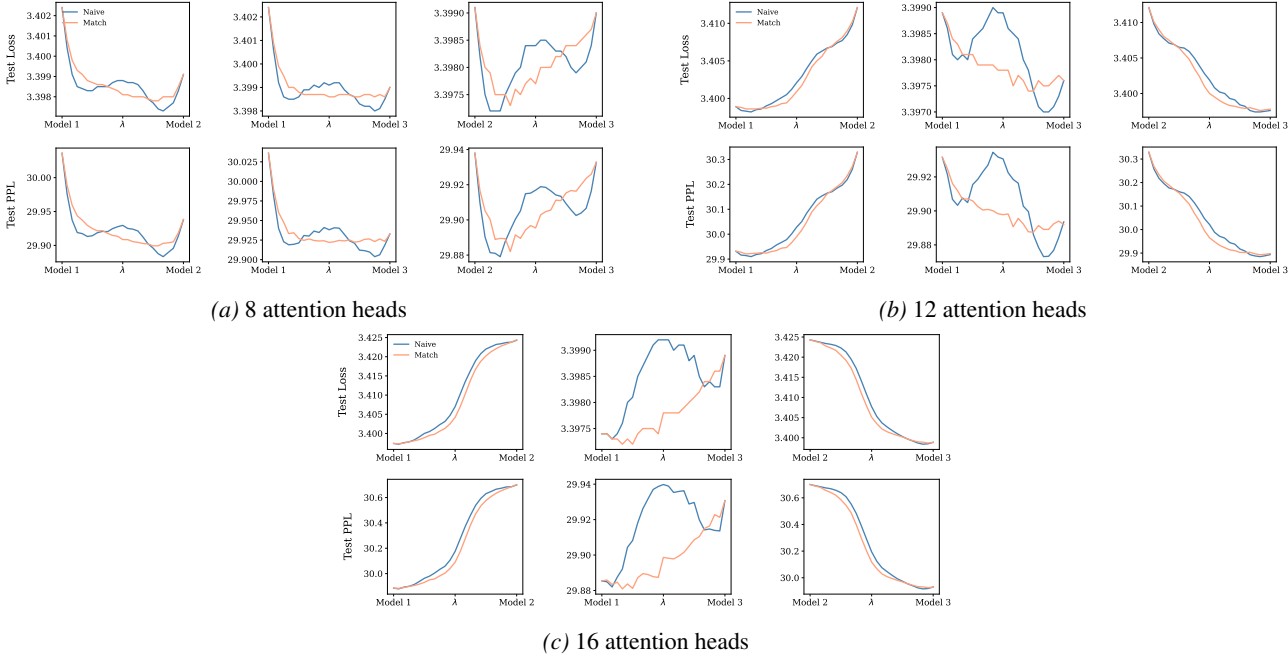

*Figure 21.* Linear Mode Connectivity for GPT2 on One Billion Words with 12 layers.

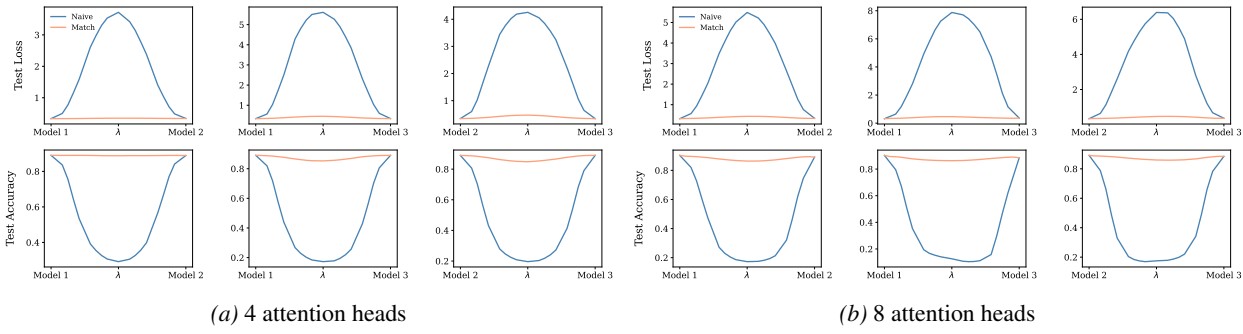

*(a)* 4 attention heads                    *(b)* 8 attention heads

*Figure 22.* Linear Mode Connectivity for ViT-RoPE on MNIST with 1 layer

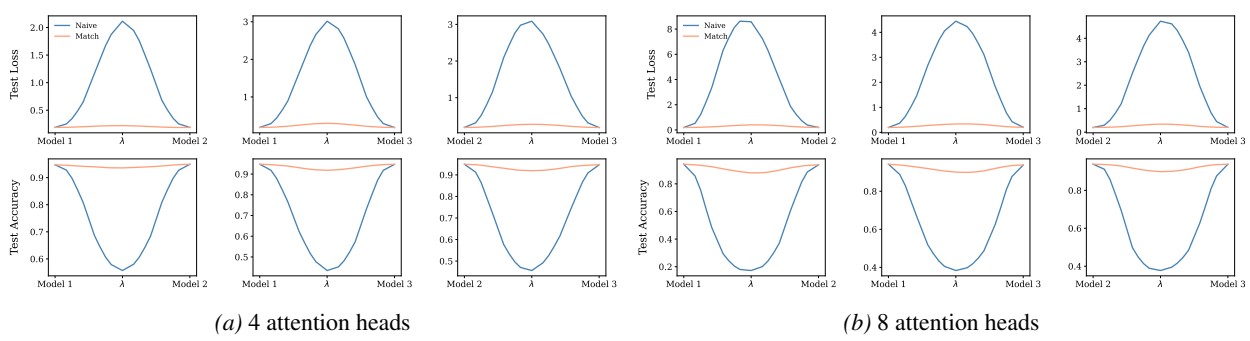

*(a)* 4 attention heads                    *(b)* 8 attention heads

*Figure 23.* Linear Mode Connectivity for ViT-RoPE on MNIST with 2 layers

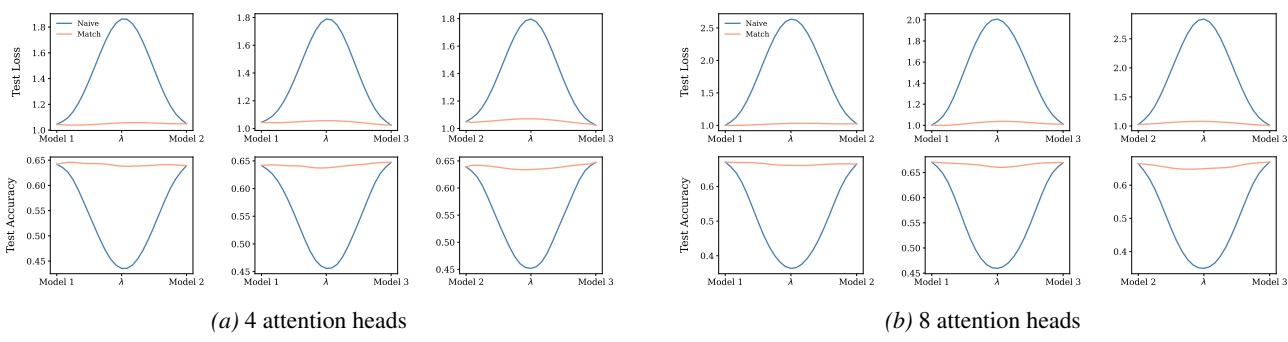

*(a)* 4 attention heads                    *(b)* 8 attention heads

*Figure 24.* Linear Mode Connectivity for ViT-RoPE on CIFAR-10 with 2 layers

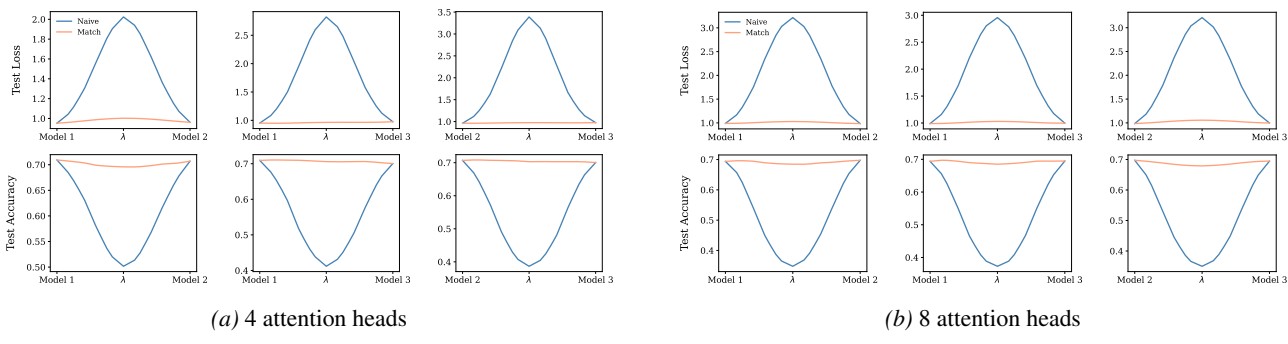

*(a)* 4 attention heads                    *(b)* 8 attention heads

*Figure 25.* Linear Mode Connectivity for ViT-RoPE on CIFAR-10 with 4 layers

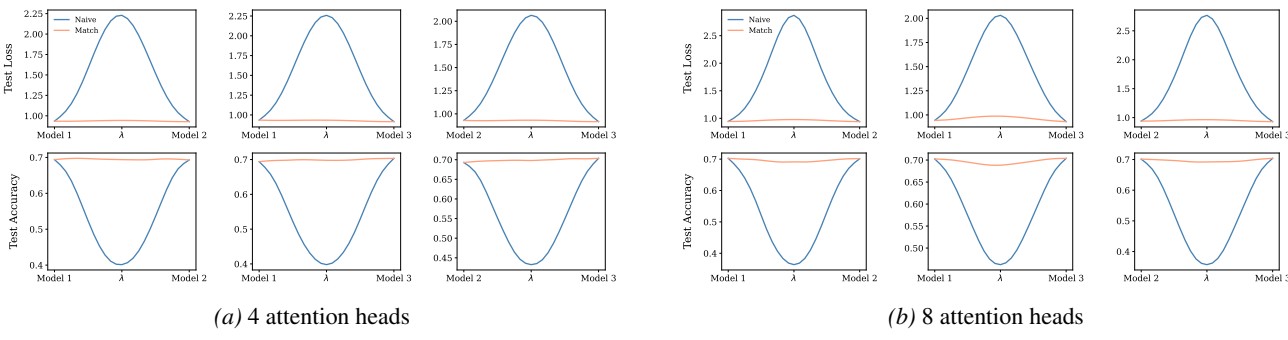

*(a)* 4 attention heads                    *(b)* 8 attention heads

*Figure 26.* Linear Mode Connectivity for ViT-RoPE on CIFAR-10 with 6 layers

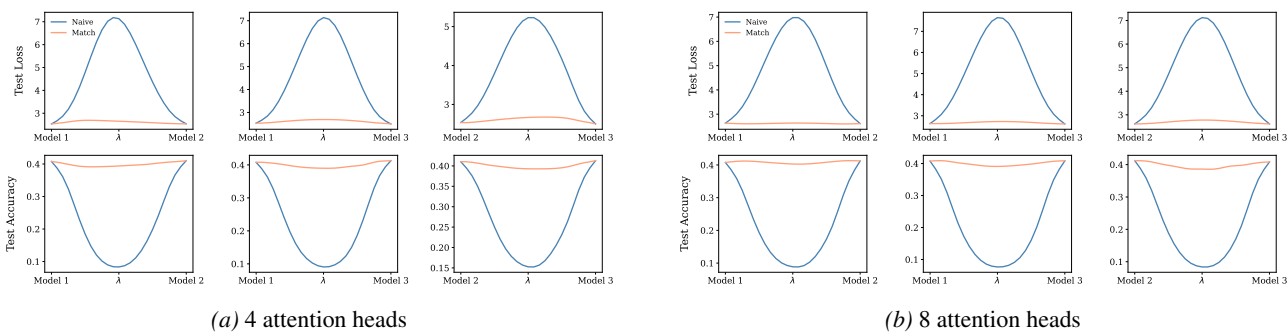

*(a)* 4 attention heads                    *(b)* 8 attention heads

*Figure 27.* Linear Mode Connectivity for ViT-RoPE on CIFAR-100 with 6 layers

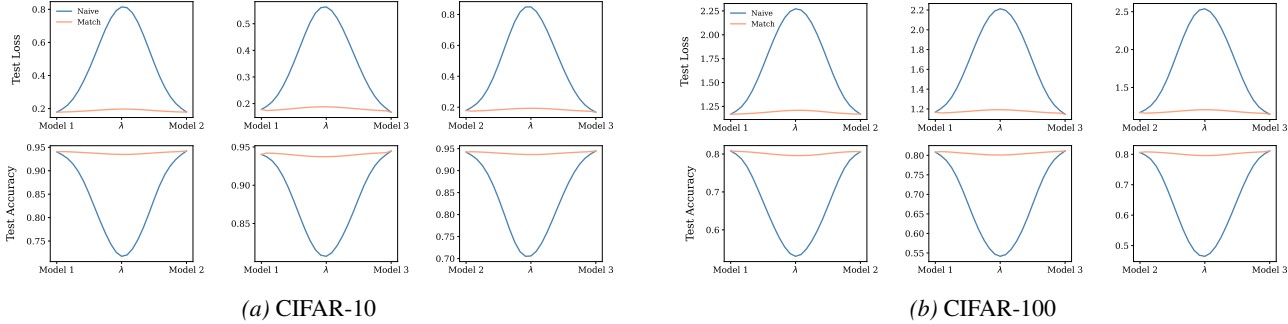

*(a)* CIFAR-10                    *(b)* CIFAR-100

*Figure 28.* Linear Mode Connectivity for ViT-RoPE on ImageNet21k→CIFAR-10/100 with 12 layers and 6 heads

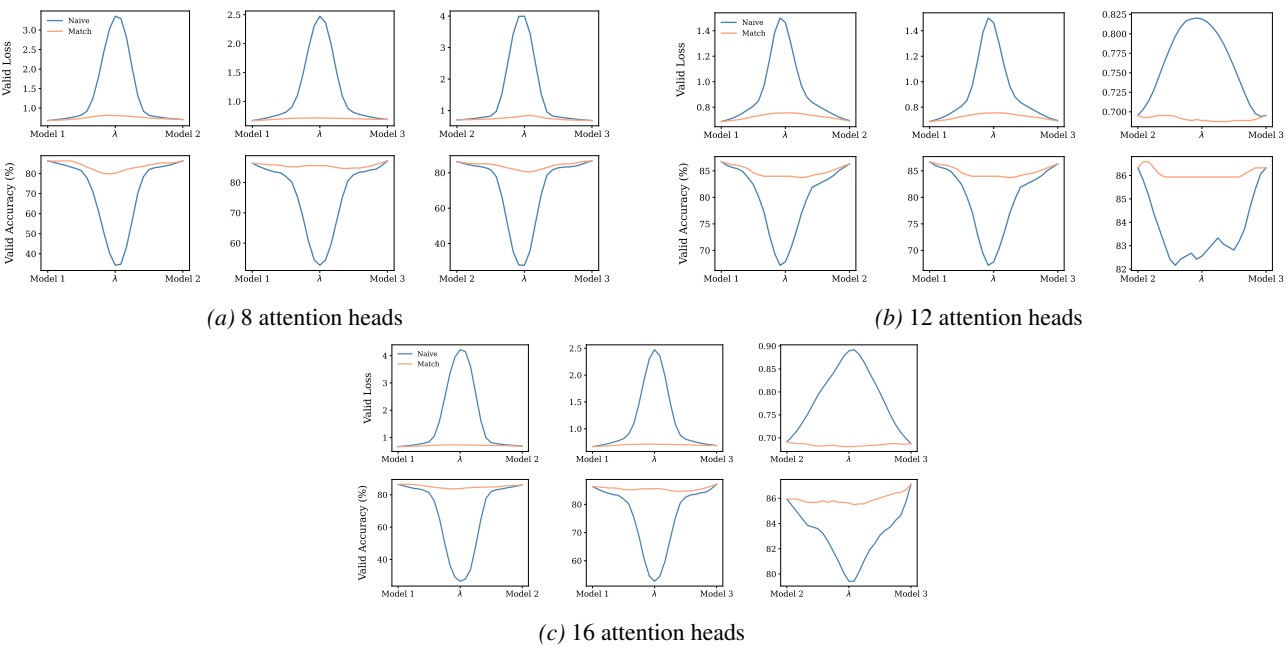

*(a)* 8 attention heads  *(b)* 12 attention heads

*(c)* 16 attention heads

*Figure 29.* Linear Mode Connectivity for ViT-RoPE on ImageNet with 12 layers.

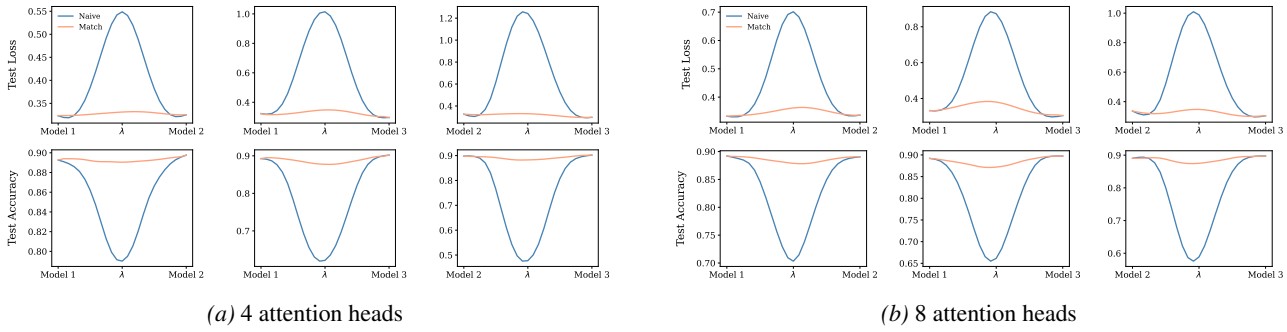

*(a)* 4 attention heads  *(b)* 8 attention heads

*Figure 30.* Linear Mode Connectivity for BERT-RoPE on AGnews with 2 layers

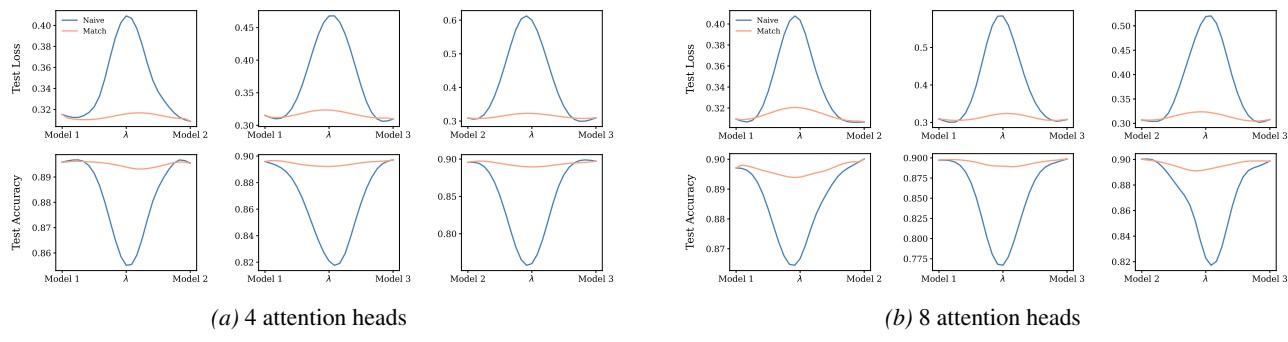

*(a)* 4 attention heads  *(b)* 8 attention heads

*Figure 31.* Linear Mode Connectivity for BERT-RoPE on AGnews with 6 layers

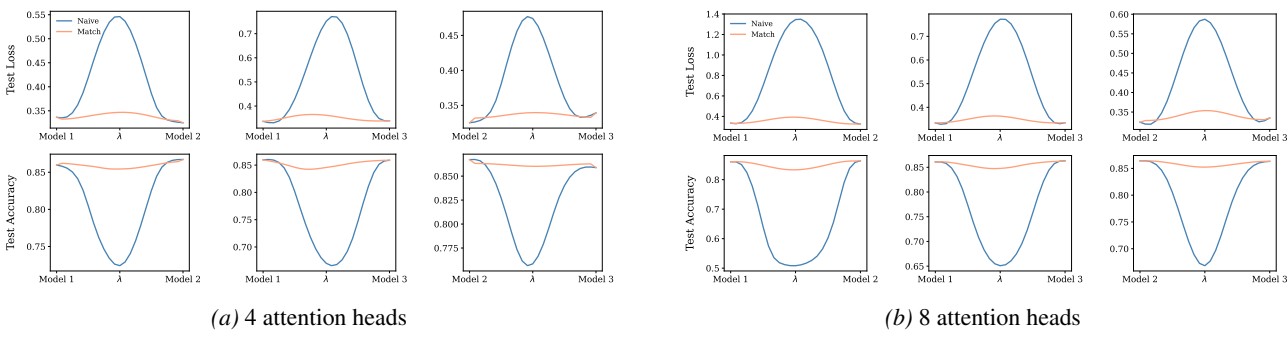

*(a)* 4 attention heads     *(b)* 8 attention heads

*Figure 32.* Linear Mode Connectivity for BERT-RoPE on IMDBreview with 2 layers

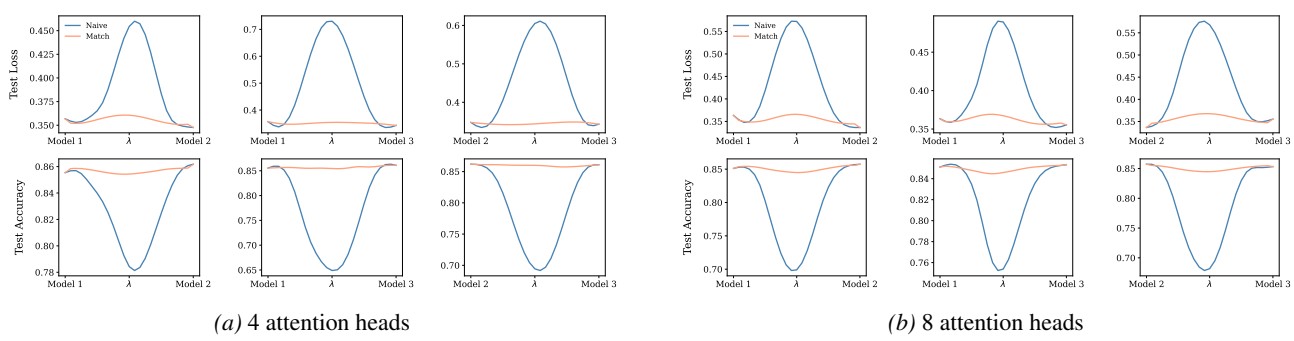

*(a)* 4 attention heads     *(b)* 8 attention heads

*Figure 33.* Linear Mode Connectivity for BERT-RoPE on IMDBreview with 6 layers

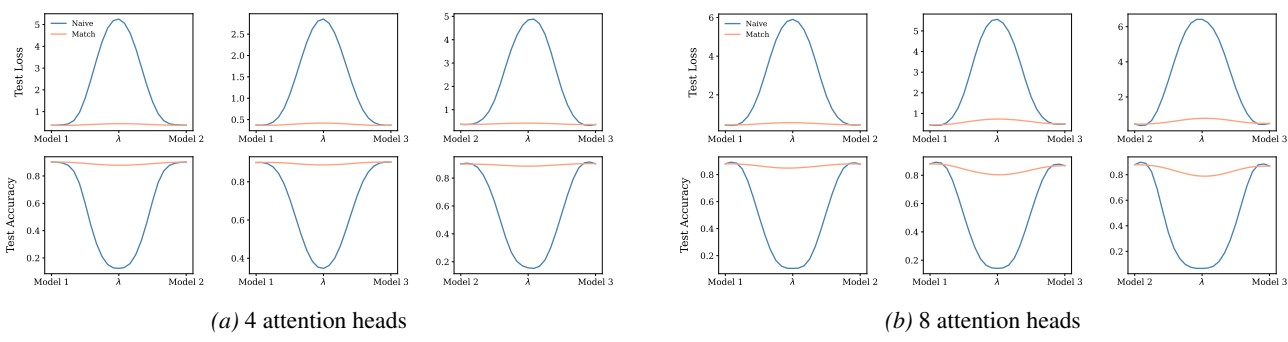

*(a)* 4 attention heads     *(b)* 8 attention heads

*Figure 34.* Linear Mode Connectivity for BERT-RoPE on DBPedia with 2 layers

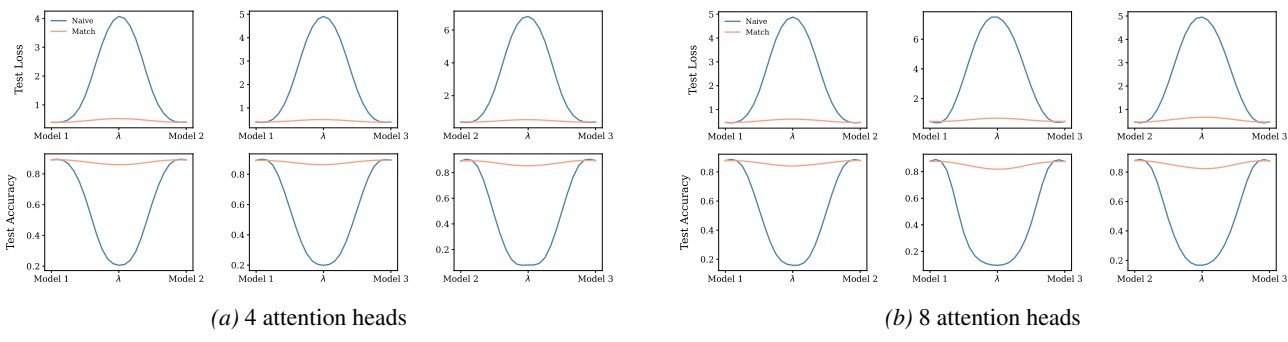

*(a)* 4 attention heads     *(b)* 8 attention heads

*Figure 35.* Linear Mode Connectivity for BERT-RoPE on DBPedia with 6 layers

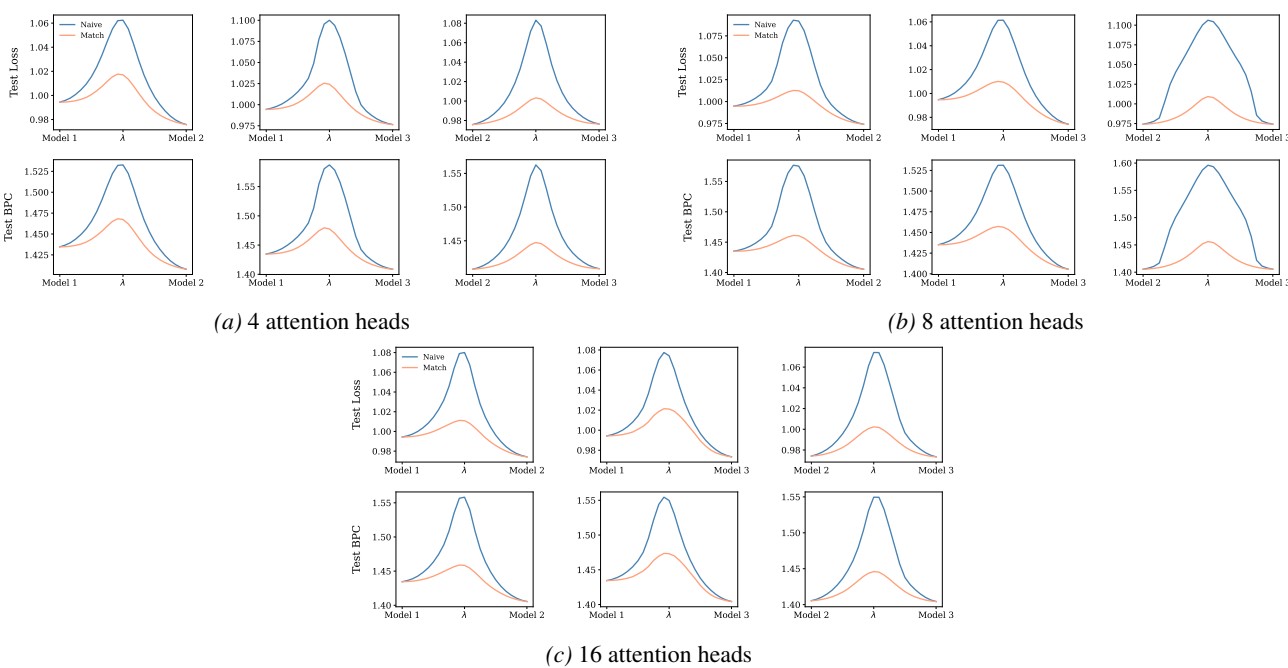

*(a)* 4 attention heads

*(b)* 8 attention heads

*(c)* 16 attention heads

*Figure 36.* Linear Mode Connectivity for GPT2-RoPE on Enwik8 with 12 layers.

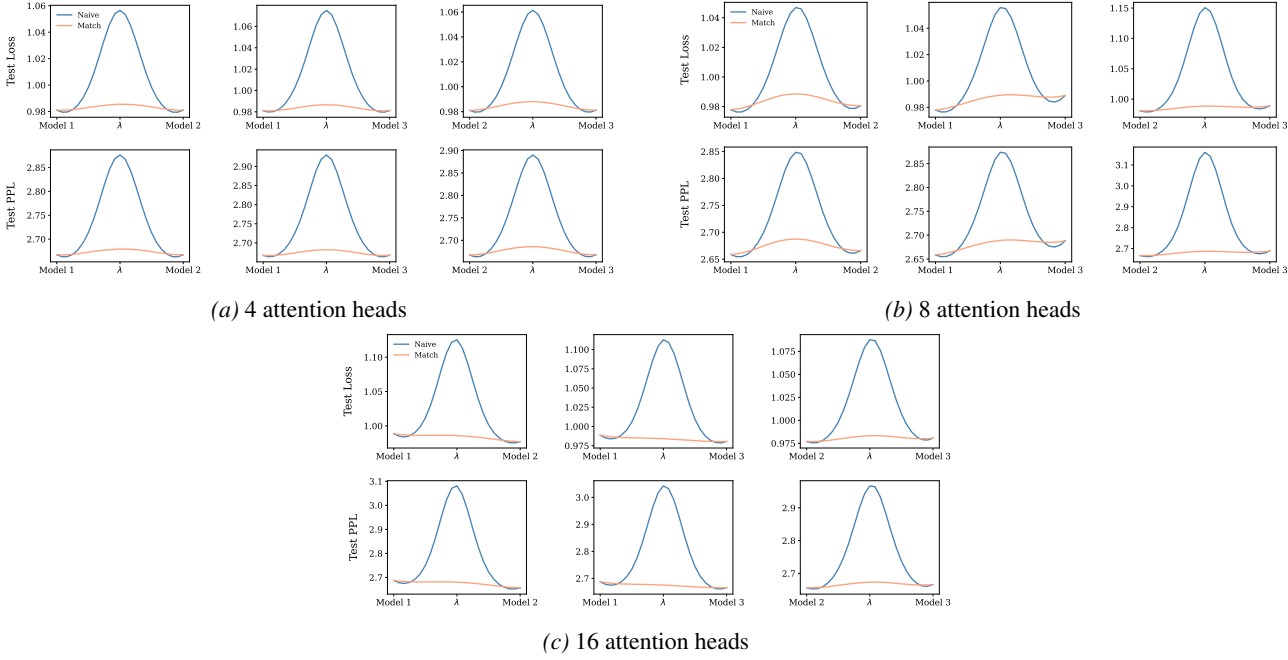

*(a)* 4 attention heads

*(b)* 8 attention heads

*(c)* 16 attention heads

*Figure 37.* Linear Mode Connectivity for Llama on Enwik8 with 12 layers.

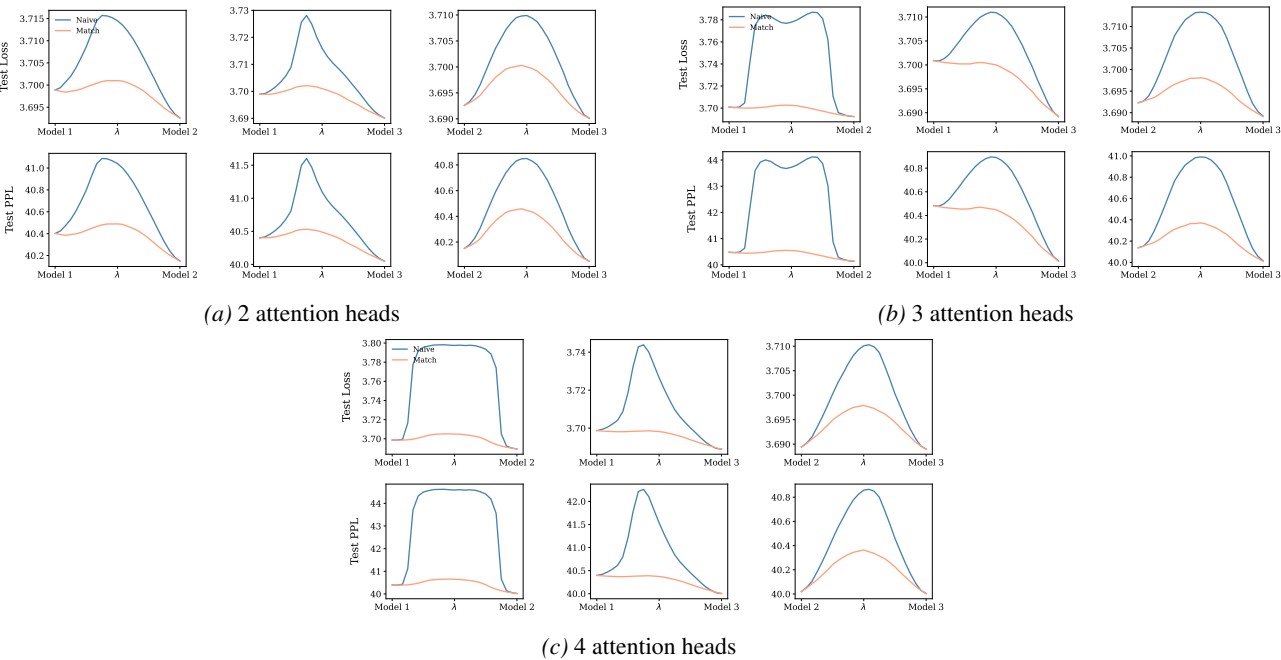

*Figure 38.* Linear Mode Connectivity for GPT2-RoPE on Wikitext103 with 12 layers.

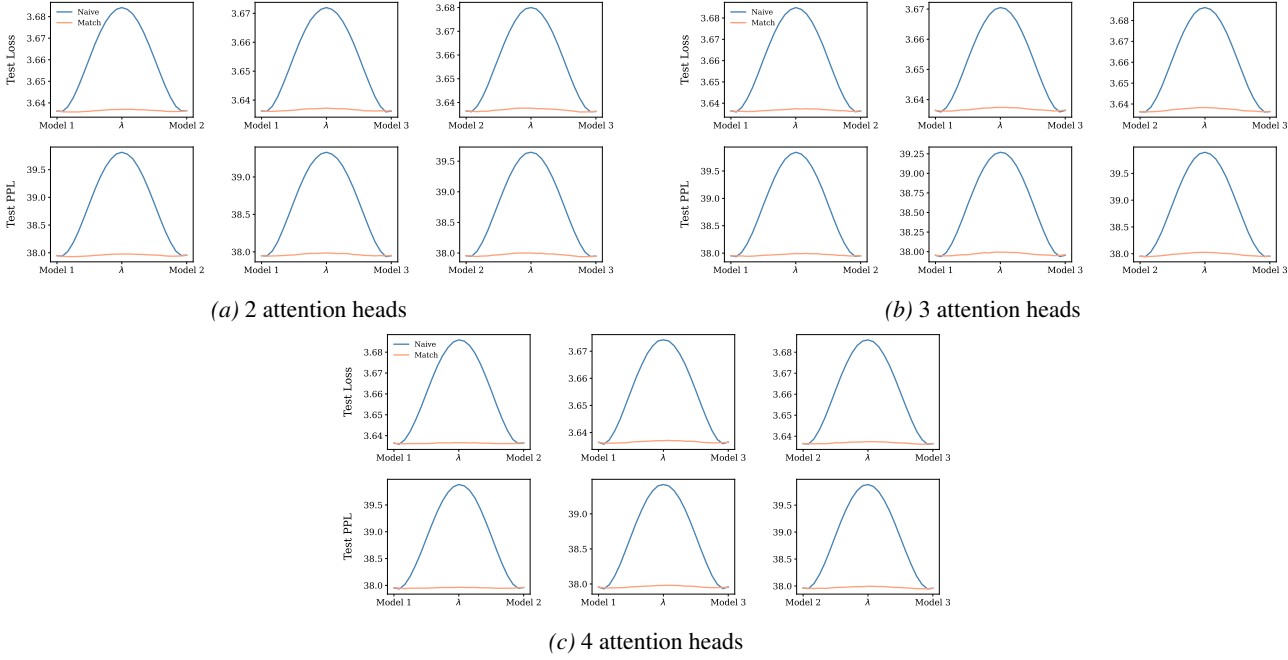

*Figure 39.* Linear Mode Connectivity for LLama on Wikitext103 with 12 layers.

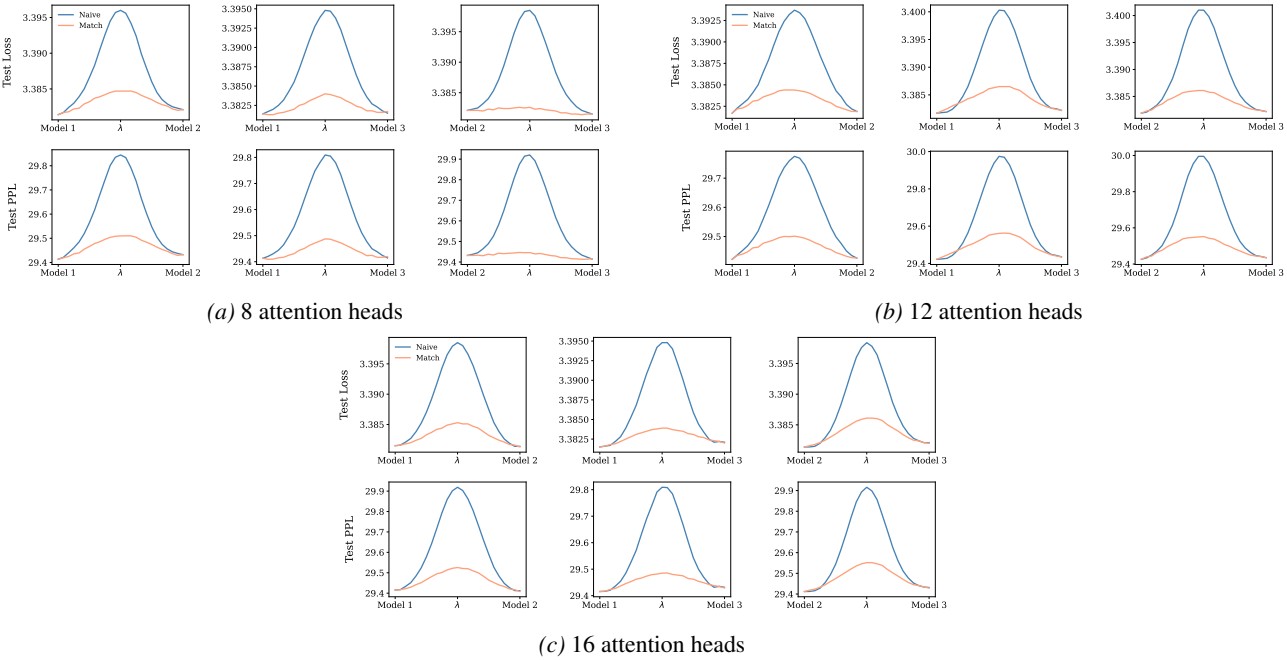

*(a)* 8 attention heads

*(b)* 12 attention heads

*(c)* 16 attention heads

*Figure 40.* Linear Mode Connectivity for GPT2-RoPE on OneBillionWord with 12 layers.

## J.2. Linear Mode Connectivity for Attention at All Layers

*Table 3.* Experimental configurations for LMC evaluation under re-initialization of *all attention layers*. The table reports datasets, model depth, and attention head counts, with figure references showing interpolation curves for APE and RoPE variants. Entries of the form $A \rightarrow B$ denote models pretrained on $A$, fine-tuned on $B$, and assessed on $B$.

| Dataset | Layers | Heads | APE | RoPE | Dataset | Layers | Heads | APE | RoPE |
|---|---|---|---|---|---|---|---|---|---|
| MNIST | 2 | [4, 8] | [41a, 41b] | [56a, 56b] | AGNews | 2 | [4, 8] | [48a, 48b] | [62a, 62b] |
| CIFAR-10 | 2 | [4, 8] | [42a, 42b] | [57a, 57b] | | 6 | [4, 8] | [49a, 49b] | [63a, 63b] |
| | 4 | [4, 8] | [43a, 43b] | [58a, 58b] | IMDB | 2 | [4, 8] | [50a, 50b] | [64a, 64b] |
| | 6 | [4, 8] | [44a, 44b] | [59a, 59b] | | 6 | [4, 8] | [51a, 51b] | [65a, 65b] |
| CIFAR-100 | 6 | [4, 8] | [45a, 45b] | [60a, 60b] | DBPedia | 2 | [4, 8] | [52a, 52b] | [66a, 66b] |
| ImageNet-21k→CIFAR-10 | 12 | [6] | [46a] | [61a] | | 6 | [4, 8] | [53a, 53b] | [67a, 67b] |
| ImageNet-21k→CIFAR-100 | 12 | [6] | [46b] | [61b] | Enwik8 (GPT2) | 12 | [8] | [54a] | [54b] |
| ImageNet-1k | 12 | [12] | [47a] | [47b] | WikiText103 (GPT2) | 12 | [3] | [55a] | [55b] |
| OneBillionWord (GPT2) | 12 | [12] | [68a] | [68b] | WikiText103 (Llama) | 12 | [3] | [-] | [69] |

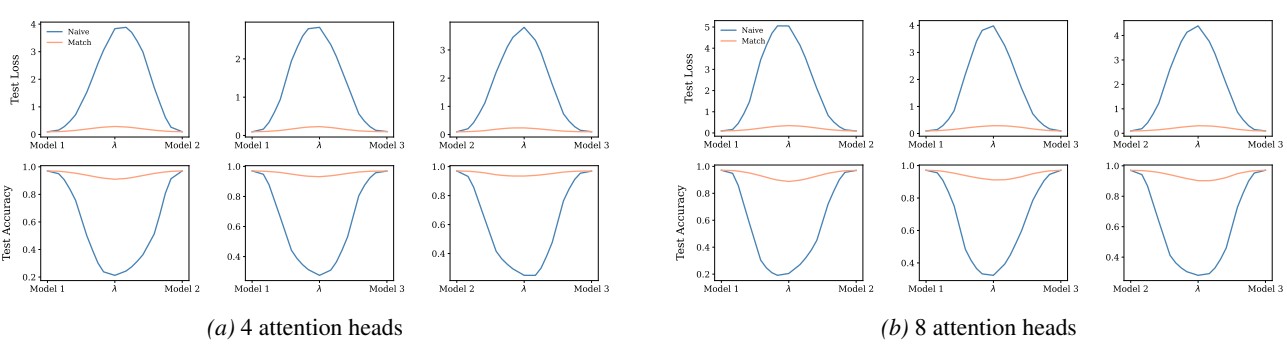

*(a)* 4 attention heads

*(b)* 8 attention heads

*Figure 41.* Linear Mode Connectivity for ViT on MNIST with 2 layers

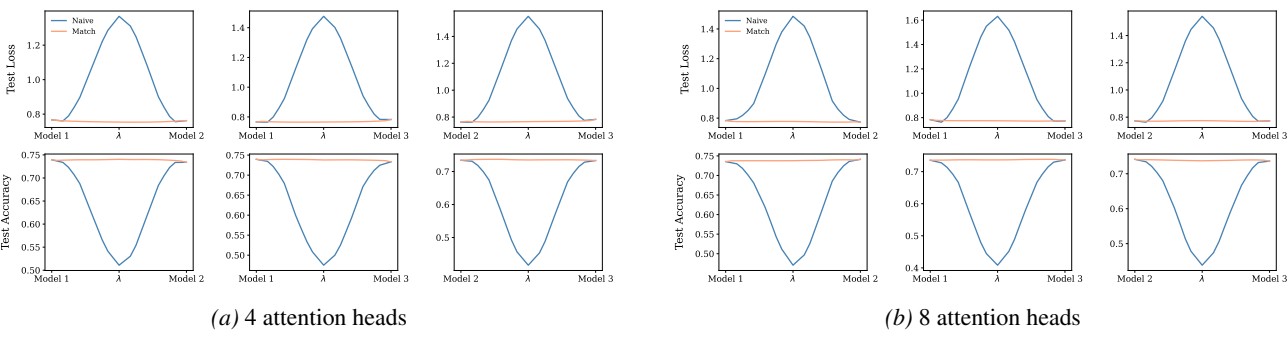

*(a)* 4 attention heads          *(b)* 8 attention heads

*Figure 42.* Linear Mode Connectivity for ViT on CIFAR-10 with 2 layers

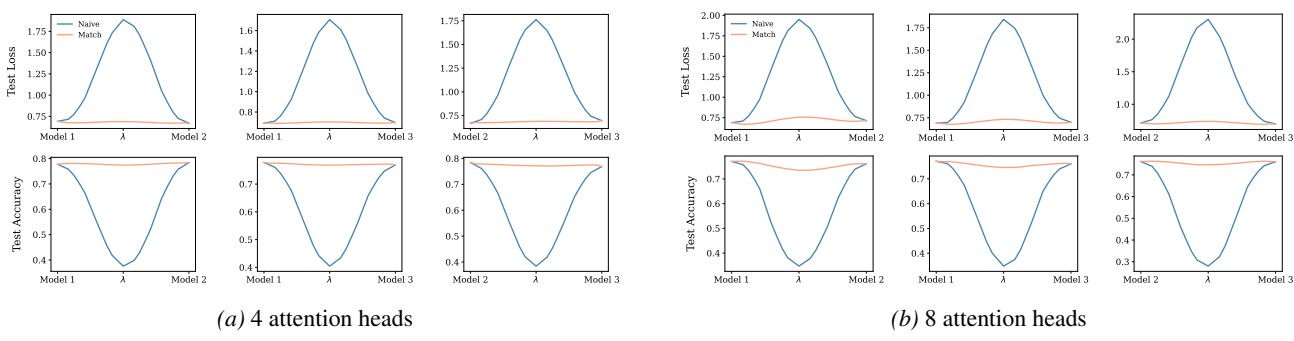

*(a)* 4 attention heads          *(b)* 8 attention heads

*Figure 43.* Linear Mode Connectivity for ViT on CIFAR-10 with 4 layers

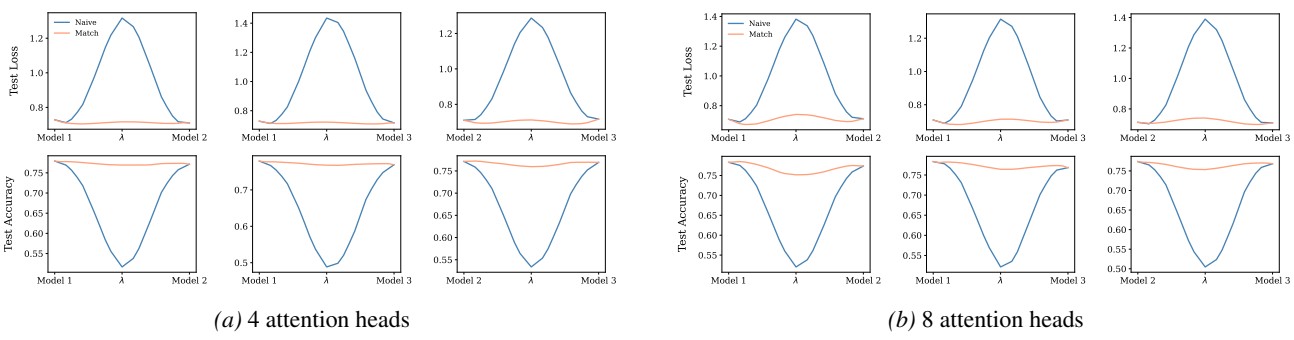

*(a)* 4 attention heads          *(b)* 8 attention heads

*Figure 44.* Linear Mode Connectivity for ViT on CIFAR-10 with 6 layers

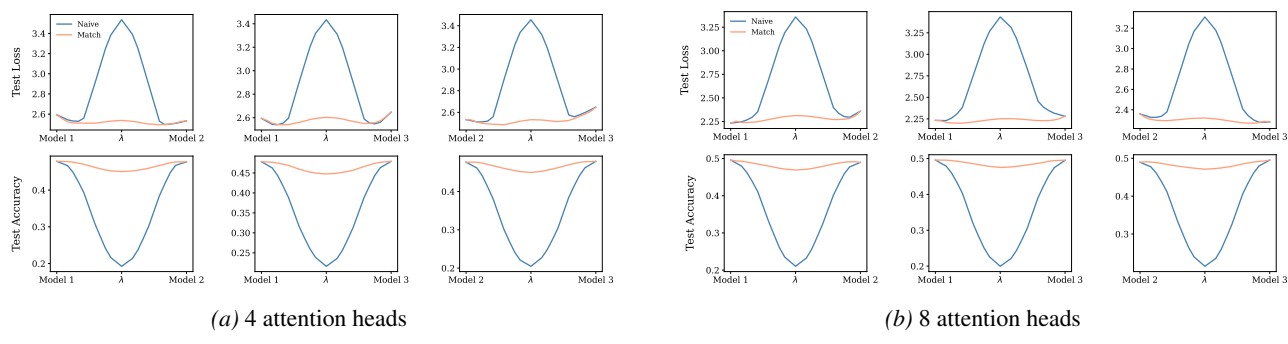

*(a)* 4 attention heads          *(b)* 8 attention heads

*Figure 45.* Linear Mode Connectivity for ViT on CIFAR-100 with 6 layers

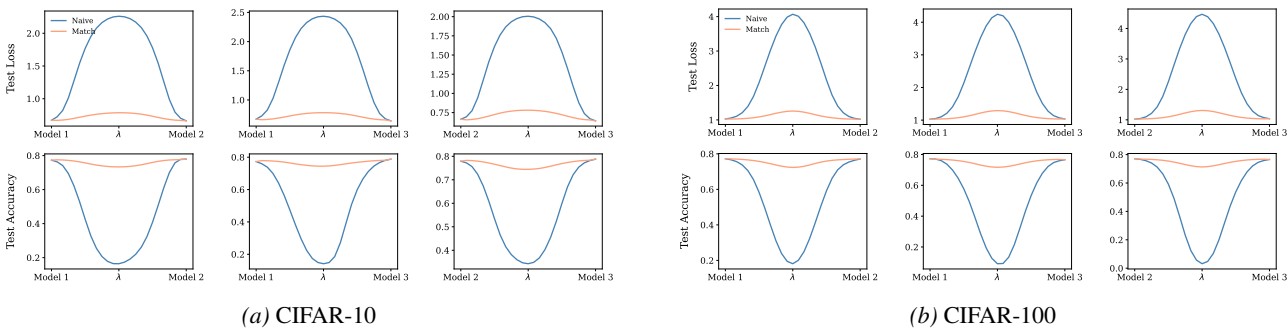

*(a)* CIFAR-10          *(b)* CIFAR-100

*Figure 46.* Linear Mode Connectivity for ViT on ImageNet21k→CIFAR-10/100 with 12 layers and 6 heads

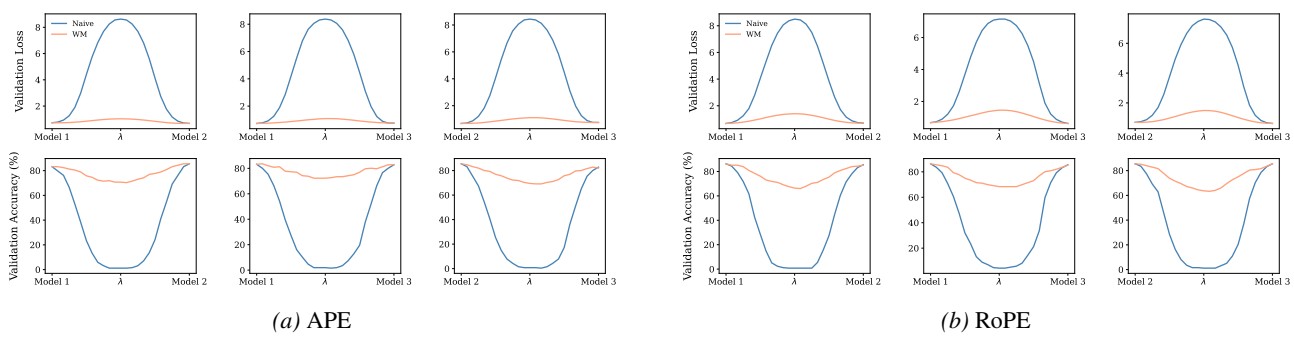

*(a)* APE          *(b)* RoPE

*Figure 47.* Linear Mode Connectivity for ViT with APE and RoPE on ImageNet-1k with 12 layers

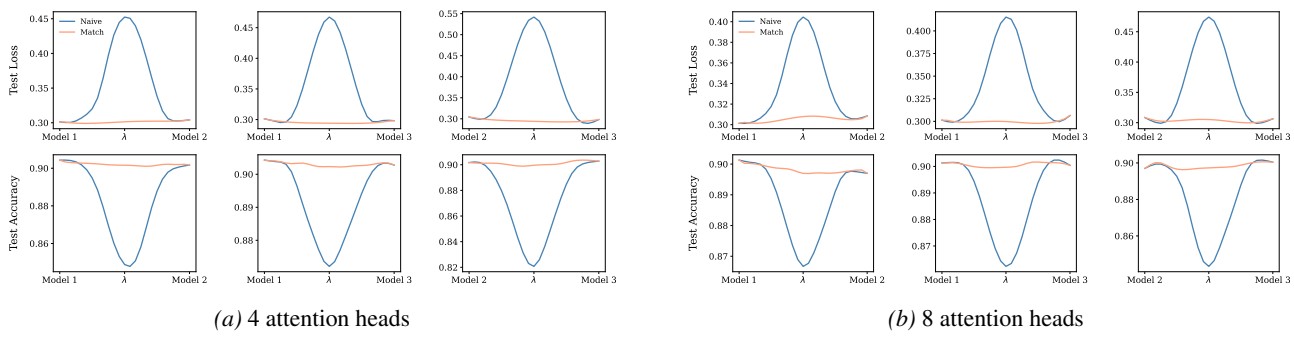

*(a)* 4 attention heads          *(b)* 8 attention heads

*Figure 48.* Linear Mode Connectivity for BERT on AGnews with 2 layers

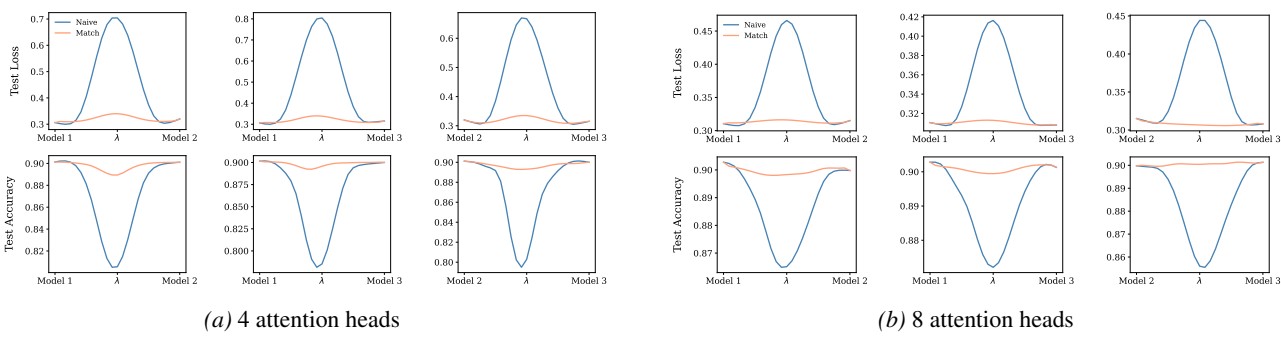

*(a)* 4 attention heads          *(b)* 8 attention heads

*Figure 49.* Linear Mode Connectivity for BERT on AGnews with 6 layers

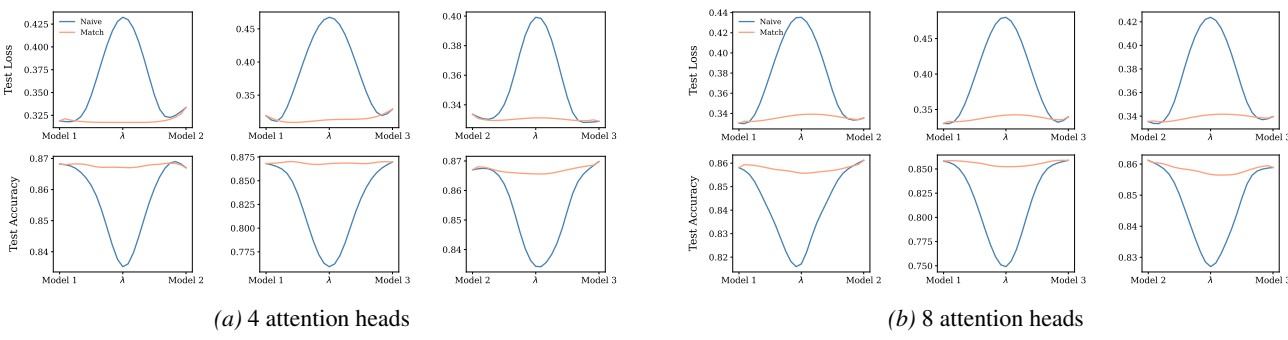

*(a)* 4 attention heads    *(b)* 8 attention heads

*Figure 50.* Linear Mode Connectivity for BERT on IMDBreview with 2 layers

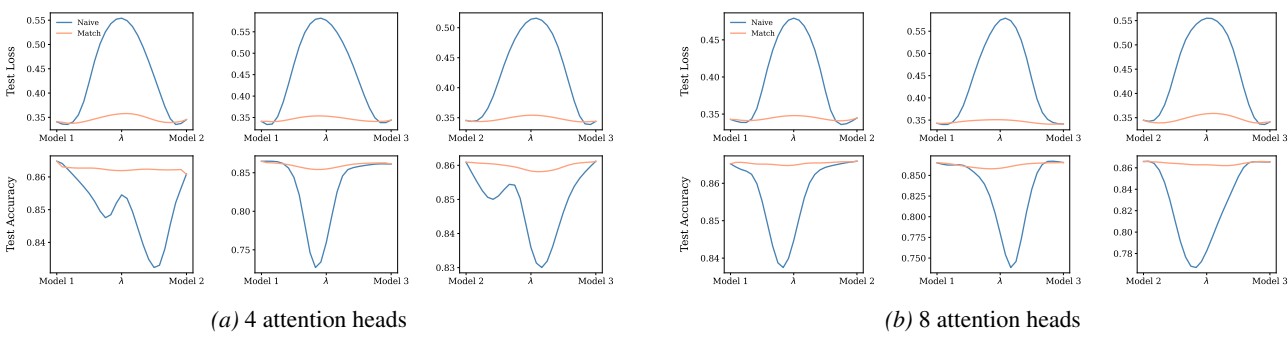

*(a)* 4 attention heads    *(b)* 8 attention heads

*Figure 51.* Linear Mode Connectivity for BERT on IMDBreview with 6 layers

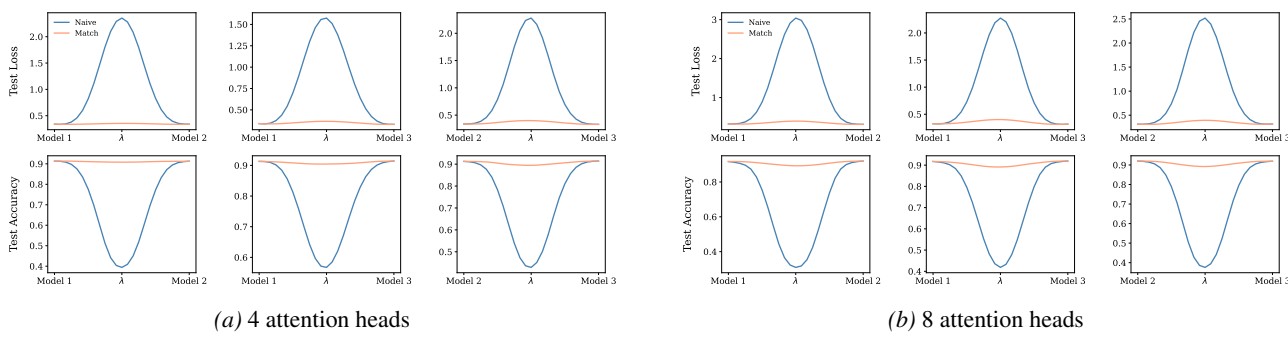

*(a)* 4 attention heads    *(b)* 8 attention heads

*Figure 52.* Linear Mode Connectivity for BERT on DBPedia with 2 layers

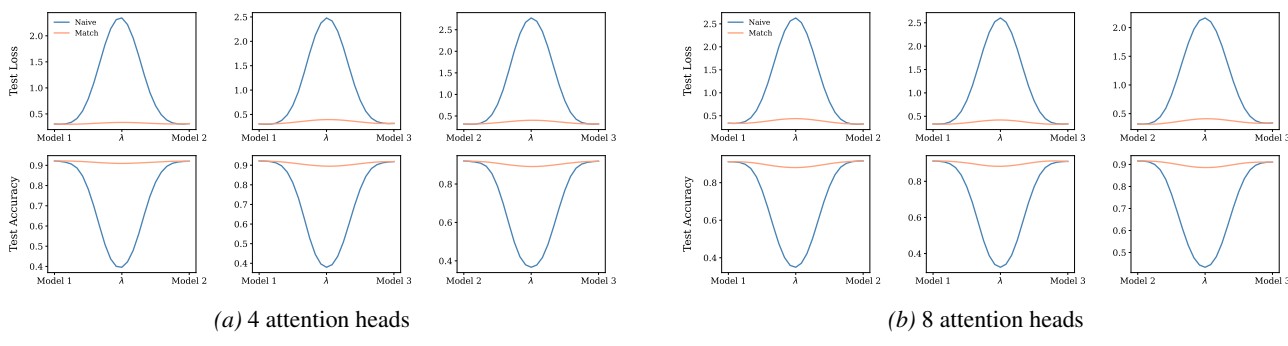

*(a)* 4 attention heads    *(b)* 8 attention heads

*Figure 53.* Linear Mode Connectivity for BERT on DBPedia with 6 layers

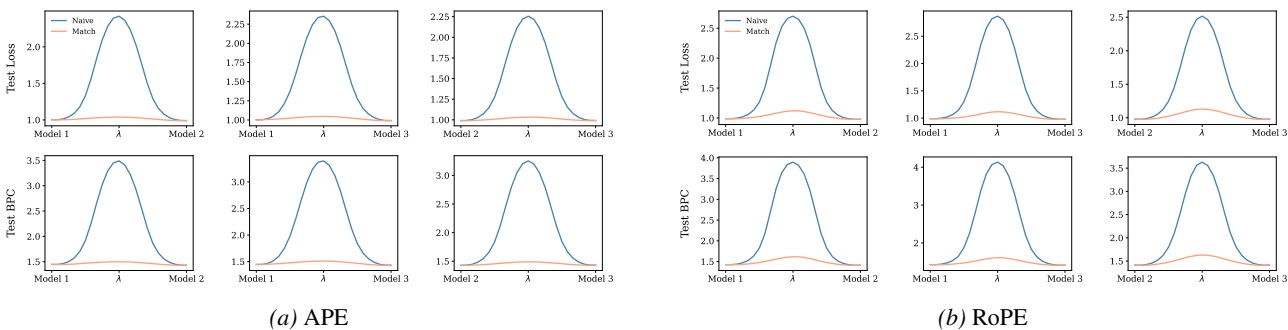

*(a)* APE  *(b)* RoPE

*Figure 54.* Linear Mode Connectivity for GPT2 with APE and RoPE on Enwik8 with 12 layers and 8 heads

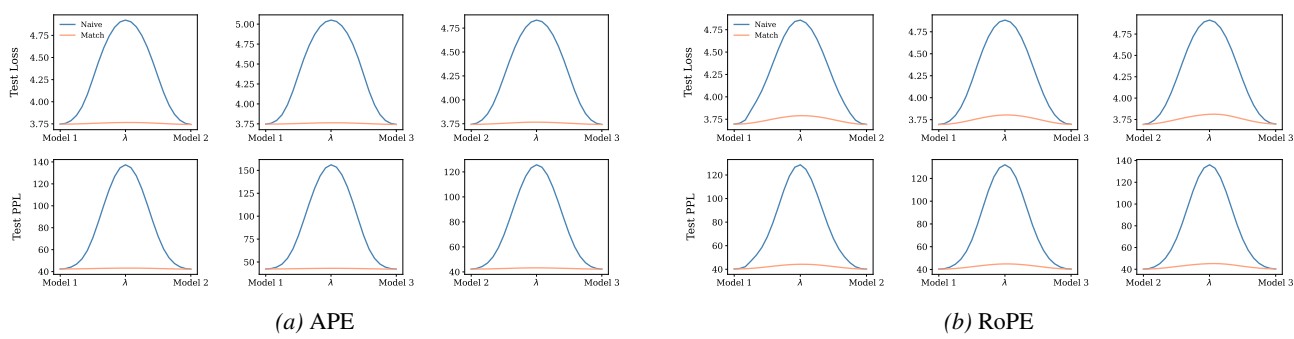

*(a)* APE  *(b)* RoPE

*Figure 55.* Linear Mode Connectivity for GPT2 with APE and RoPE on Wikitext103 with 12 layers and 3 heads

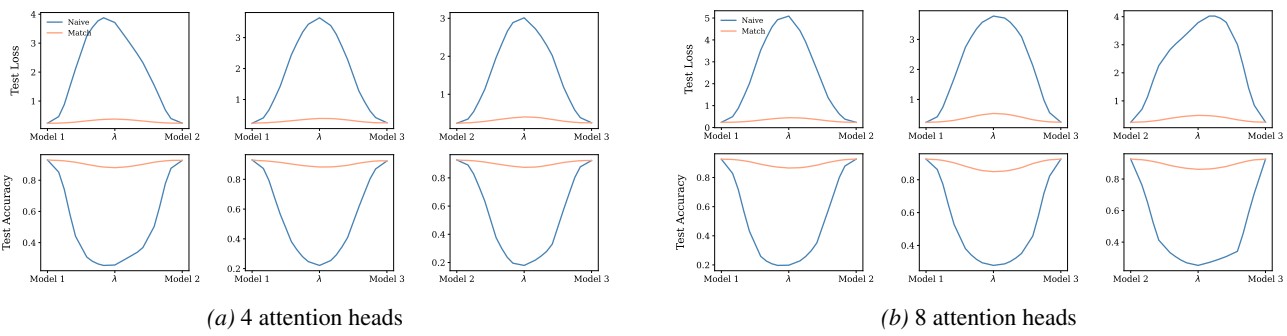

*(a)* 4 attention heads  *(b)* 8 attention heads

*Figure 56.* Linear Mode Connectivity for ViT-RoPE on MNIST with 2 layers

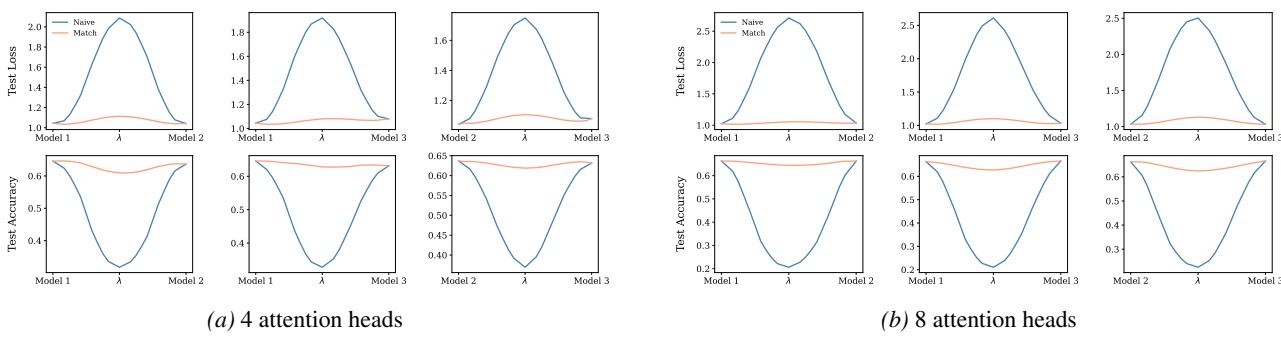

*(a)* 4 attention heads  *(b)* 8 attention heads

*Figure 57.* Linear Mode Connectivity for ViT-RoPE on CIFAR-10 with 2 layers

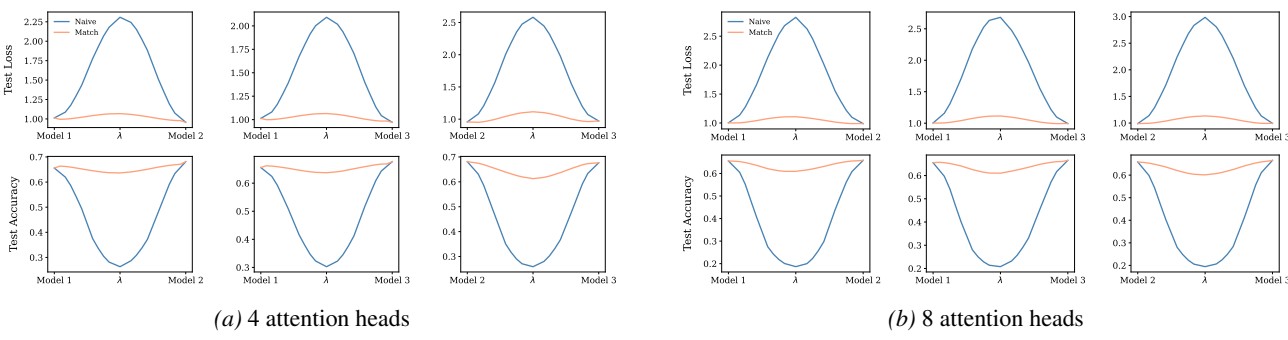

*(a)* 4 attention heads                    *(b)* 8 attention heads

*Figure 58.* Linear Mode Connectivity for ViT-RoPE on CIFAR-10 with 4 layers

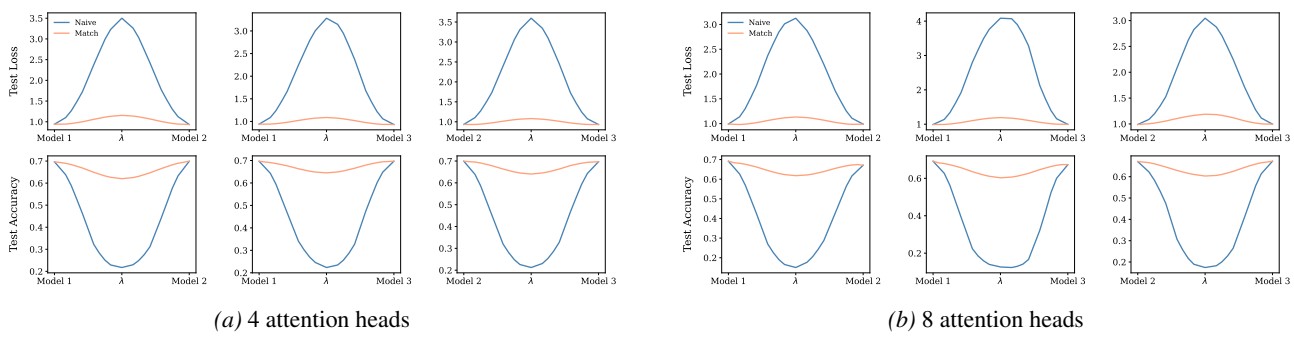

*(a)* 4 attention heads                    *(b)* 8 attention heads

*Figure 59.* Linear Mode Connectivity for ViT-RoPE on CIFAR-10 with 6 layers

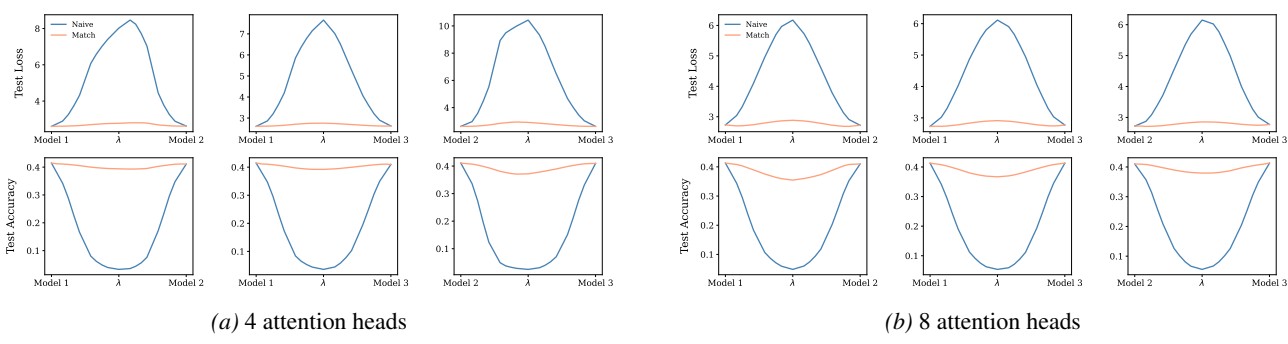

*(a)* 4 attention heads                    *(b)* 8 attention heads

*Figure 60.* Linear Mode Connectivity for ViT-RoPE on CIFAR-100 with 6 layers

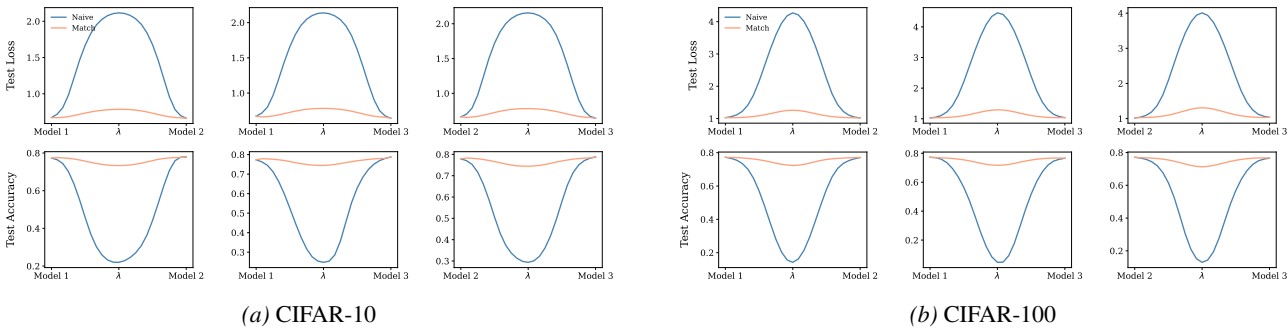

*(a)* CIFAR-10                    *(b)* CIFAR-100

*Figure 61.* Linear Mode Connectivity for ViT-RoPE on ImageNet21k→CIFAR-10/100 with 12 layers and 6 heads

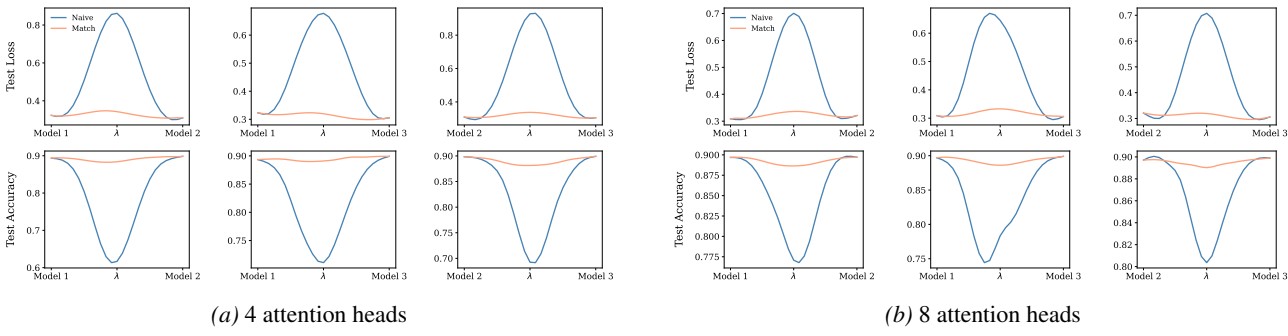

*(a)* 4 attention heads                    *(b)* 8 attention heads

*Figure 62.* Linear Mode Connectivity for BERT-RoPE on AGnews with 2 layers

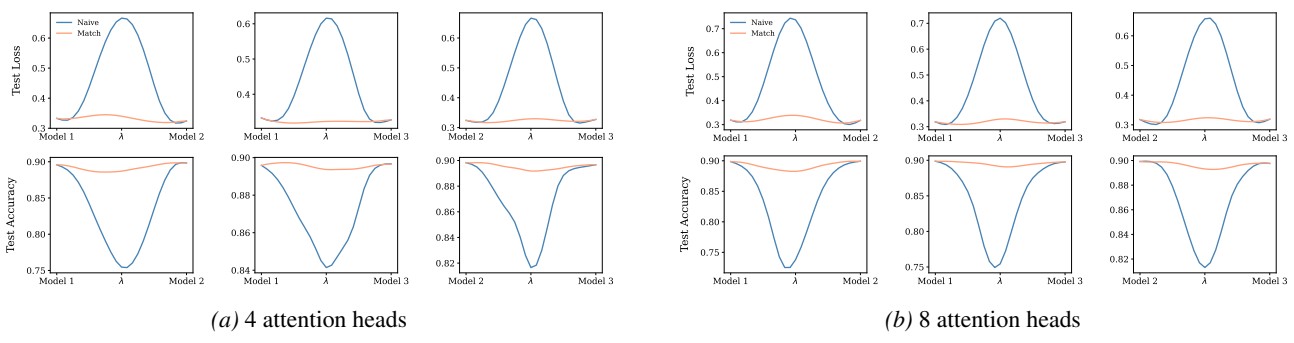

*(a)* 4 attention heads                    *(b)* 8 attention heads

*Figure 63.* Linear Mode Connectivity for BERT-RoPE on AGnews with 6 layers

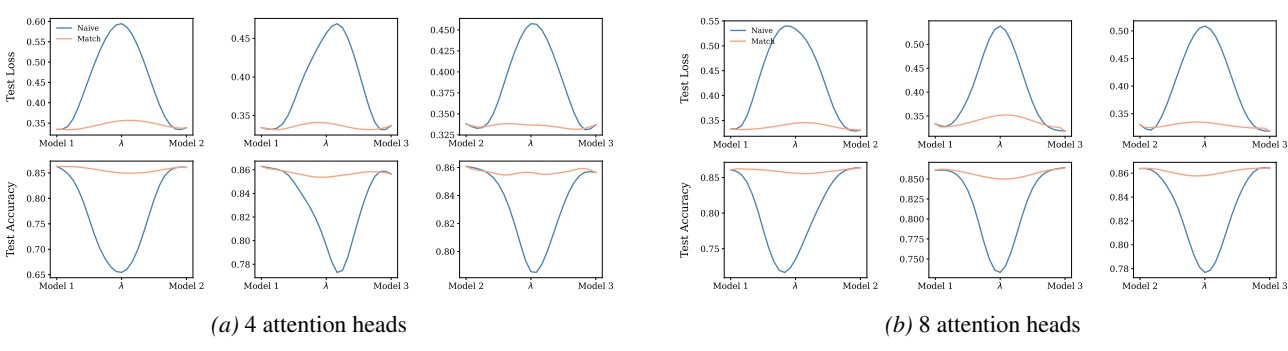

*(a)* 4 attention heads                    *(b)* 8 attention heads

*Figure 64.* Linear Mode Connectivity for BERT-RoPE on IMDBreview with 2 layers

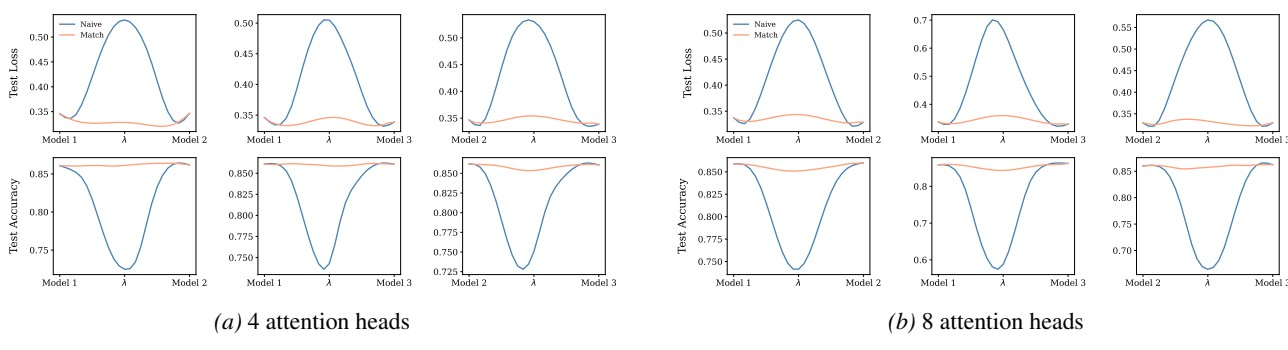

*(a)* 4 attention heads                    *(b)* 8 attention heads

*Figure 65.* Linear Mode Connectivity for BERT-RoPE on IMDBreview with 6 layers

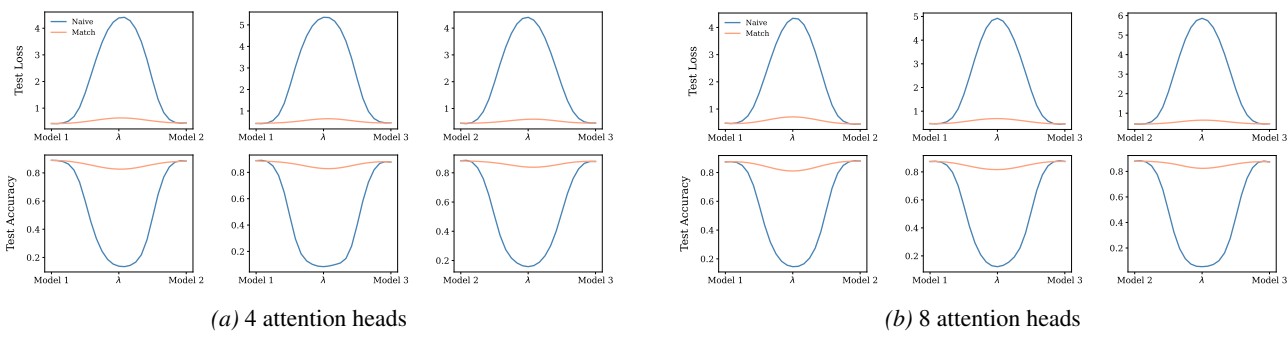

*(a)* 4 attention heads          *(b)* 8 attention heads

*Figure 66.* Linear Mode Connectivity for BERT-RoPE on DBPedia with 2 layers

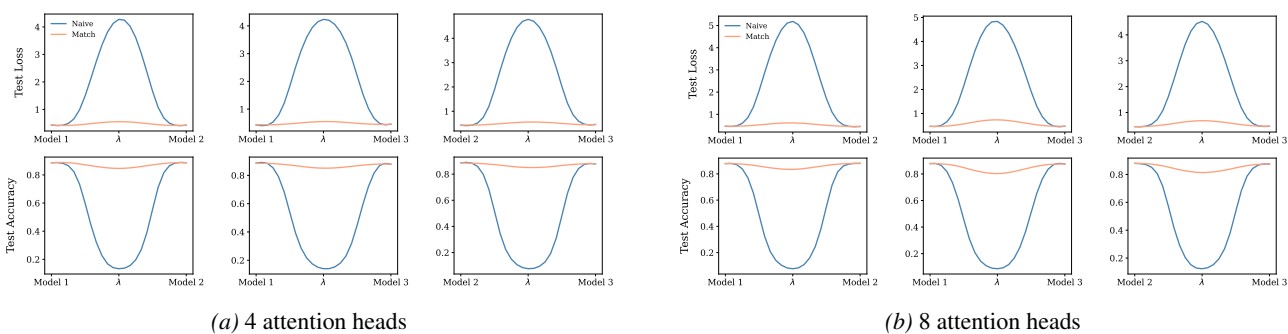

*(a)* 4 attention heads          *(b)* 8 attention heads

*Figure 67.* Linear Mode Connectivity for BERT-RoPE on DBPedia with 6 layers

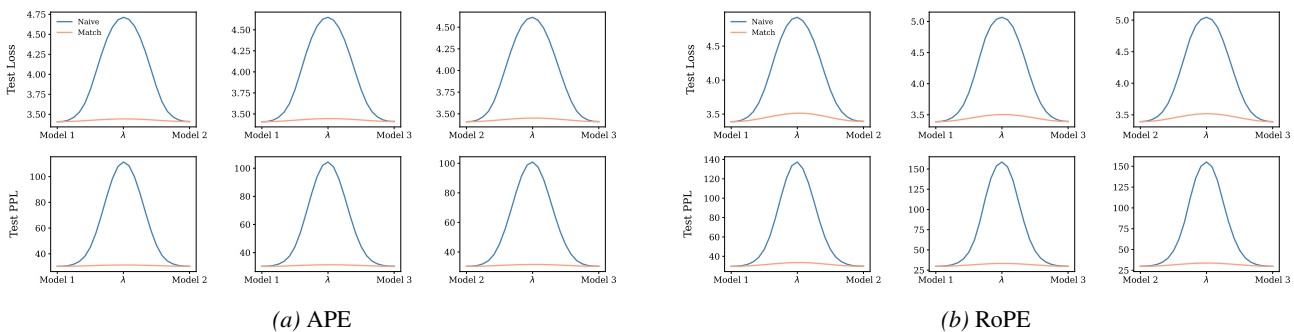

*(a)* APE          *(b)* RoPE

*Figure 68.* Linear Mode Connectivity for GPT2 on OneBillionWord with 12 layers

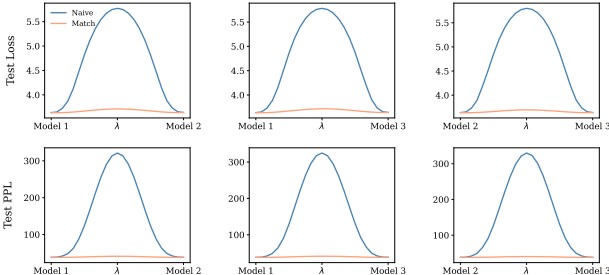

*Figure 69.* Linear Mode Connectivity for Llama on Wikitext103 with 12 layers

## J.3. Linear Mode Connectivity for Transformer First Layer

*Table 4.* Experimental setups for LMC under first Transformer layer re-initialization. The table lists datasets, model depths, and attention head counts, PE type, along with references to figures.

| Dataset | Layers | Heads | APE | RoPE | Dataset | Layers | Heads | APE | RoPE |
|---|---|---|---|---|---|---|---|---|---|
| CIFAR-10 | 6 | [8] | [70a] | [70b] | AGNews | 6 | [8] | [73a] | [73b] |
| CIFAR-100 | 6 | [8] | [71a] | [71b] | DBPedia | 6 | [8] | [74a] | [74b] |
| ImageNet-1k | 12 | [12] | [72a] | [72b] | Wikitext103 (GPT2) | 12 | [12] | [76a] | [76b] |
| Enwik8 (GPT2) | 12 | [12] | [75a] | [75b] | OneBillionWord (GPT2) | 12 | [12] | [77a] | [77b] |

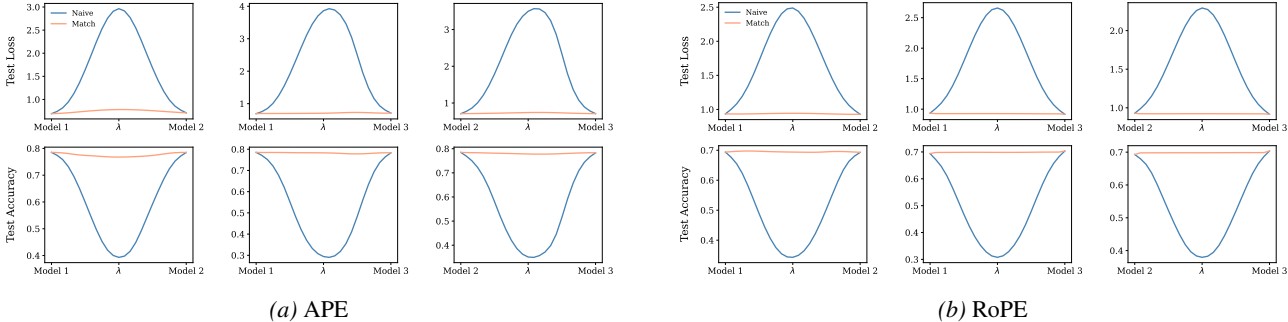

*(a)* APE        *(b)* RoPE

*Figure 70.* Linear Mode Connectivity for ViT with APE and RoPE on CIFAR-10 with 6 layers and 8 heads

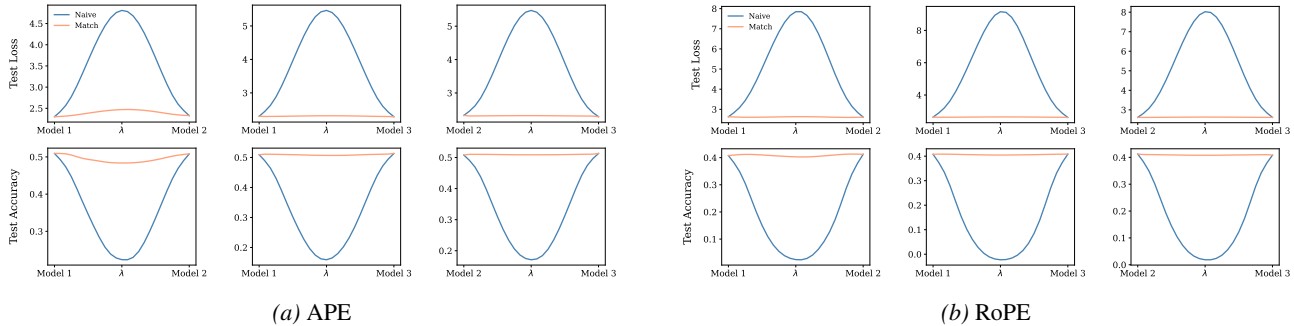

*(a)* APE        *(b)* RoPE

*Figure 71.* Linear Mode Connectivity for ViT with APE and RoPE on CIFAR-100 with 6 layers and 8 heads

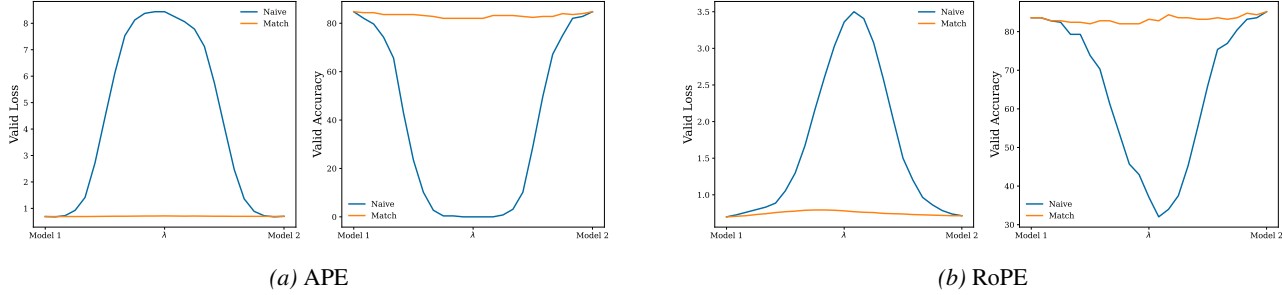

*(a)* APE        *(b)* RoPE

*Figure 72.* Linear Mode Connectivity for ViT with APE and RoPE on ImageNet-1k with 12 layers

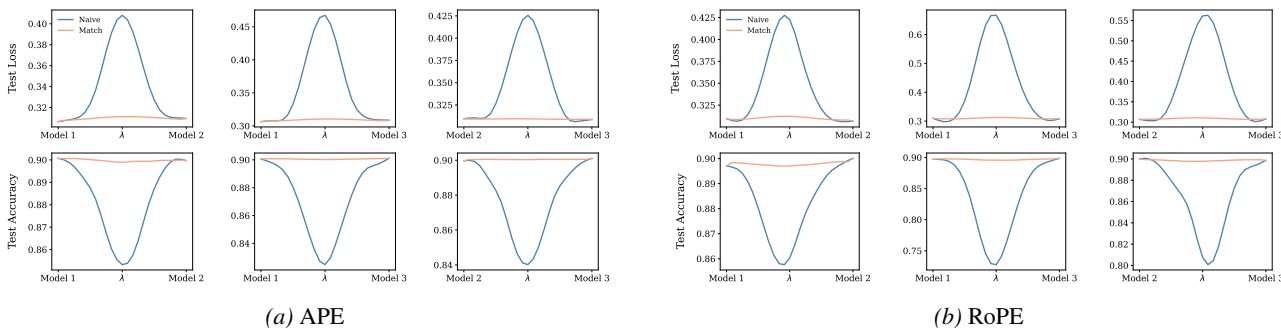

*(a)* APE        *(b)* RoPE

*Figure 73.* Linear Mode Connectivity for BERT with APE and RoPE on AGNews with 6 layers and 8 heads

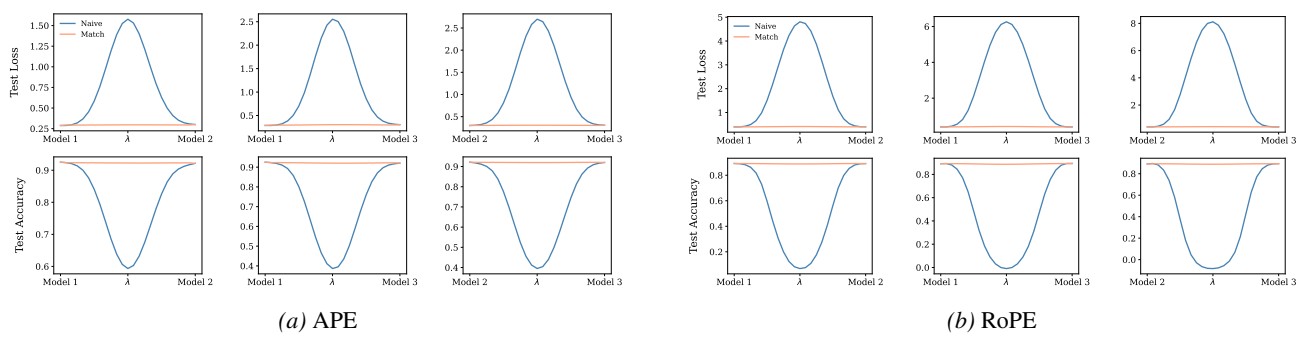

*(a)* APE        *(b)* RoPE

*Figure 74.* Linear Mode Connectivity for BERT with APE and RoPE on DBPedia with 6 layers and 8 heads

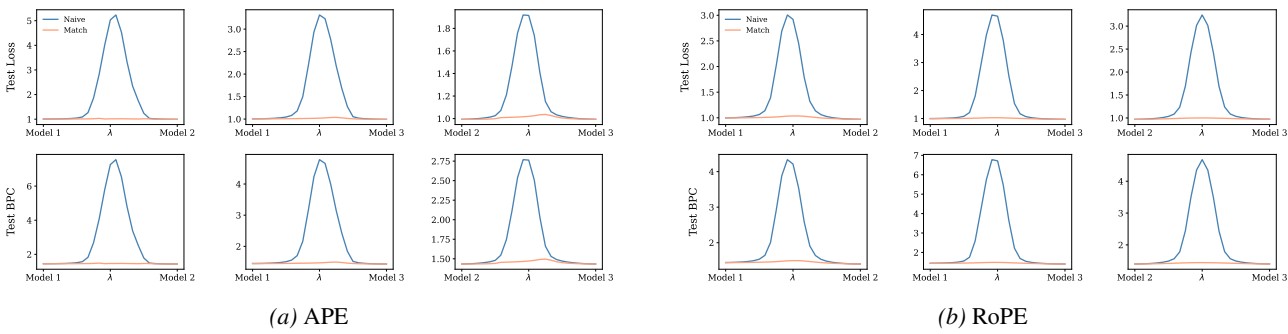

*(a)* APE        *(b)* RoPE

*Figure 75.* Linear Mode Connectivity for GPT2 with APE and RoPE on Enwik8 with 12 layers

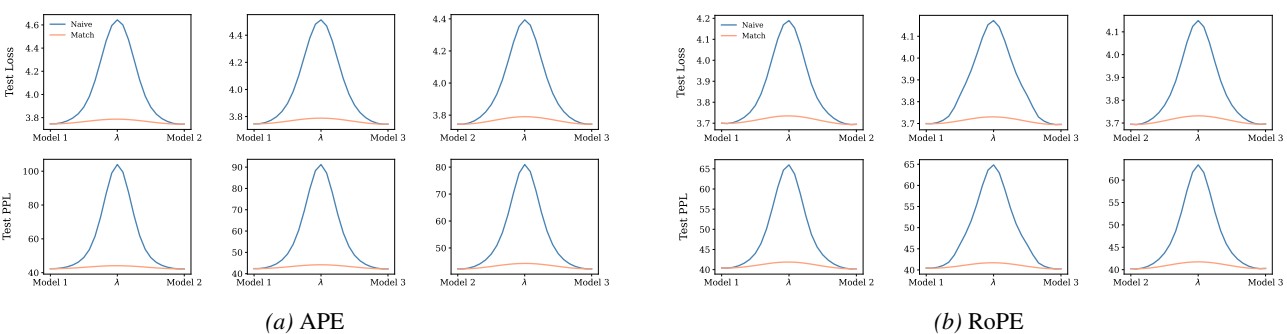

*(a)* APE        *(b)* RoPE

*Figure 76.* Linear Mode Connectivity for GPT2 with APE and RoPE on Wikitext103 with 12 layers

*Table 5.* Experimental setups for LMC under full Transformer re-initialization. The table lists datasets, model depths, and attention head counts, along with references to figures comparing APE and RoPE. This configuration represents the most disruptive reset scenario considered in our study.

| Dataset | Layers | Heads | APE | RoPE |
|---|---|---|---|---|
| CIFAR-10 | 6 | [8] | [79a] | [79b] |
| CIFAR-100 | 6 | [8] | [80a] | [80b] |
| AGNews | 6 | [8] | [81a] | [81b] |
| DBPedia | 6 | [8] | [82a] | [82b] |
| ImageNet-1k | [12] | [12] | [83a] | [83b] |
| Wikitext103 | [12] | [3] | [84a] | [84b] |

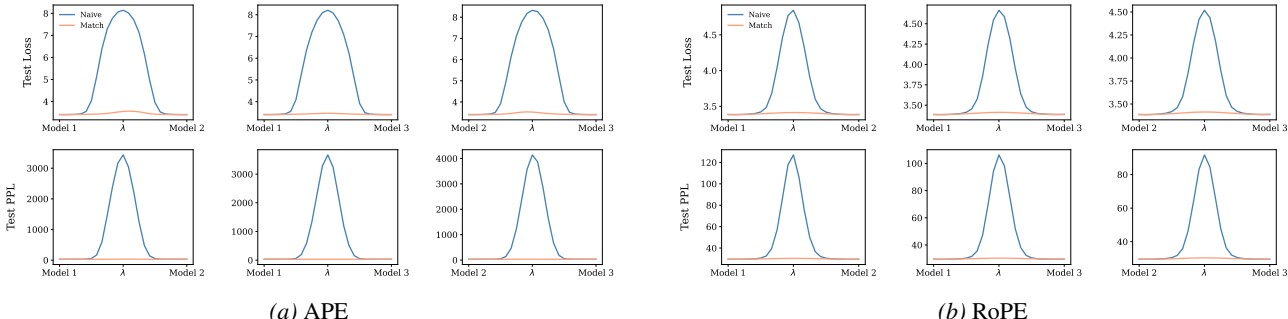

*(a)* APE   *(b)* RoPE

*Figure 77.* Linear Mode Connectivity for GPT2 with APE and RoPE on OneBillionWord with 12 layers

## J.4. Linear Mode Connectivity for Full Model

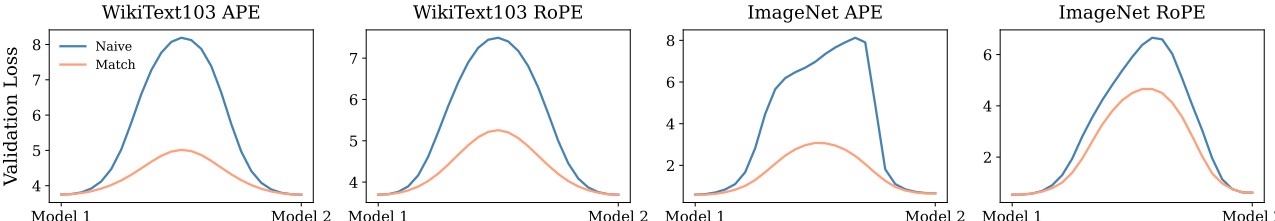

*Figure 78.* LMC interpolation plots for ViT on ImageNet-1K (subplots 3 and 4) and GPT-2 on WikiText103 (subplots 1 and 2), with APE and RoPE under full Transformer re-initialization.

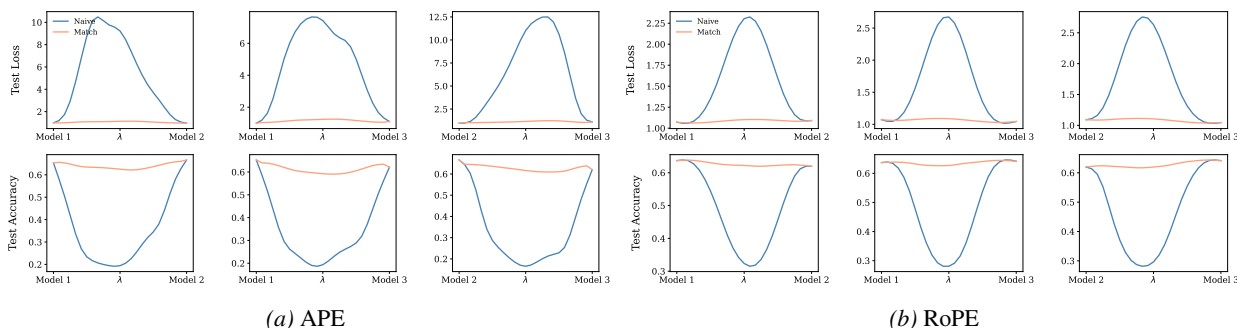

*(a)* APE   *(b)* RoPE

*Figure 79.* Linear Mode Connectivity for ViT with APE and RoPE on CIFAR-10 with 6 layers and 8 heads

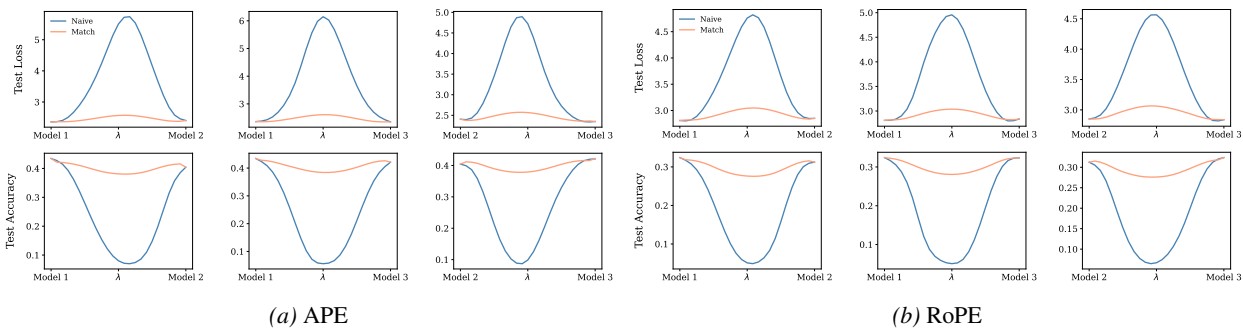

*(a)* APE        *(b)* RoPE

*Figure 80.* Linear Mode Connectivity for ViT with APE and RoPE on CIFAR-100 with 6 layers and 8 heads

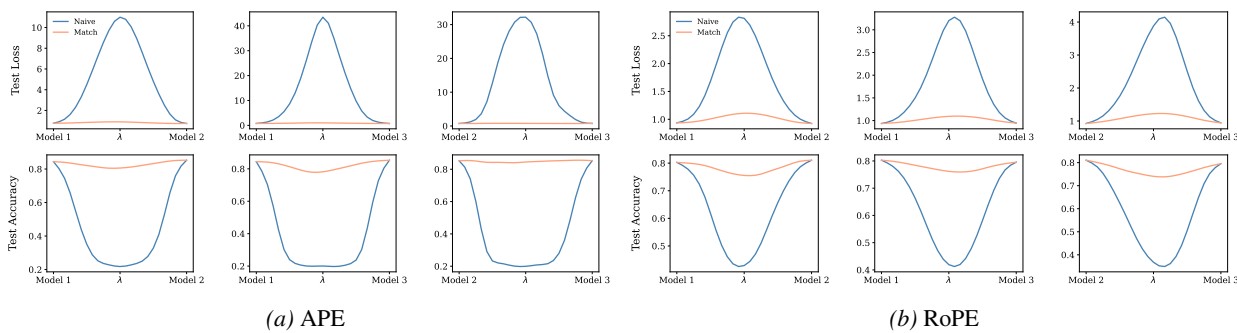

*(a)* APE        *(b)* RoPE

*Figure 81.* Linear Mode Connectivity for BERT with APE and RoPE on AGNews with 6 layers and 8 heads

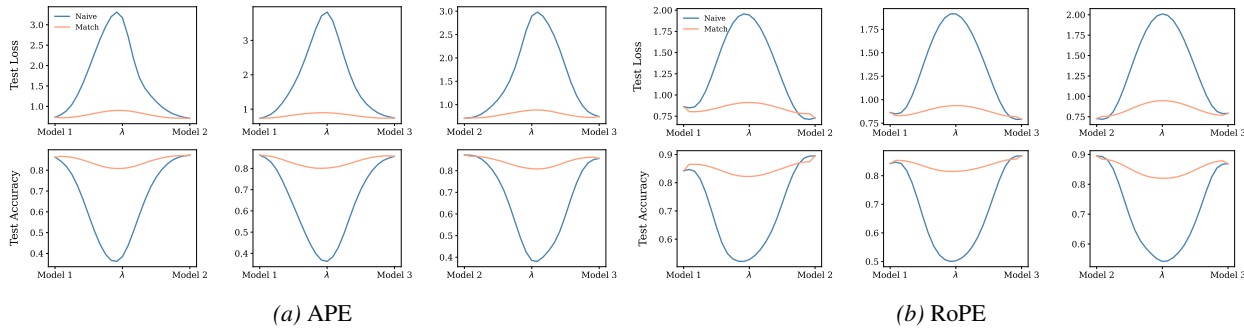

*(a)* APE        *(b)* RoPE

*Figure 82.* Linear Mode Connectivity for BERT with APE and RoPE on DBPedia with 6 layers and 8 heads

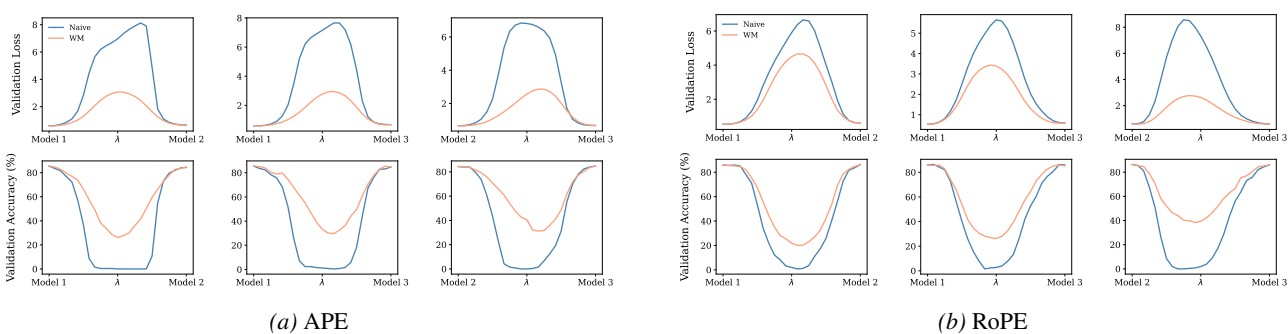

*(a)* APE        *(b)* RoPE

*Figure 83.* Linear Mode Connectivity for ViT with APE and RoPE on ImageNet-1k with 12 layers

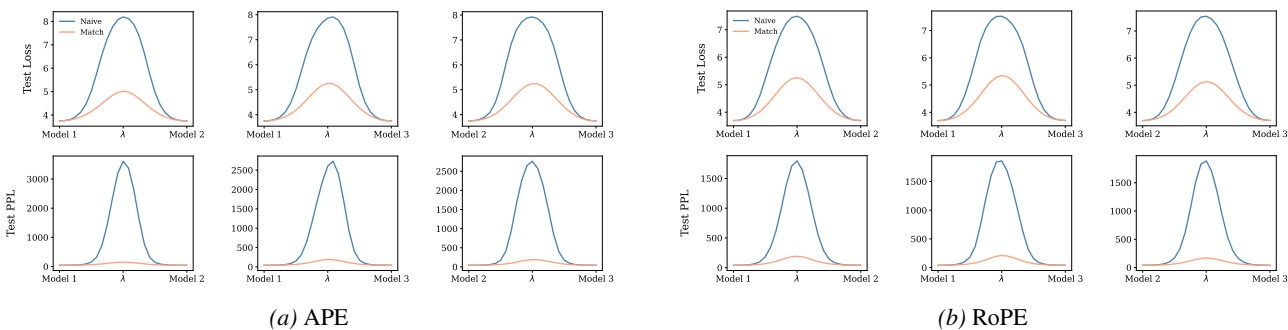

*(a)* APE

*(b)* RoPE

*Figure 84.* Linear Mode Connectivity for GPT2 with APE and RoPE on Wikitext103 with 12 layers

## J.5. Ablation study on Head Permutation

We plot 24 head permutations, including the one selected by Stage 1 our method, with Stage 2 applied post-reordering for all permutation. For the 4-head case, this encompasses all possible permutations (4! = 24). For the 8-head case, it includes 23 randomly sampled permutations along with the one chosen by our method.

*Table 6.* Ablation study on head permutation

| Dataset | No. layers | No. heads | APE Figure | RoPE Figure |
|---------|------------|-----------|------------|-------------|
| CIFAR-10 | 2 | [4, 8] | [85a, 85b] | [92a, 92b] |
|          | 6 | [4, 8] | [86a, 86b] | [93a, 93b] |
| CIFAR-100 | 6 | [4, 8] | [87a, 87b] | [94a, 94b] |
| IMDBreview | 2 | [4, 8] | [88a, 88b] | [95a, 95b] |
|            | 6 | [4, 8] | [89a, 89b] | [96a, 96b] |
| DBPedia | 2 | [4, 8] | [90a, 90b] | [97a, 97b] |
|         | 6 | [4, 8] | [91a, 91b] | [98a, 98b] |

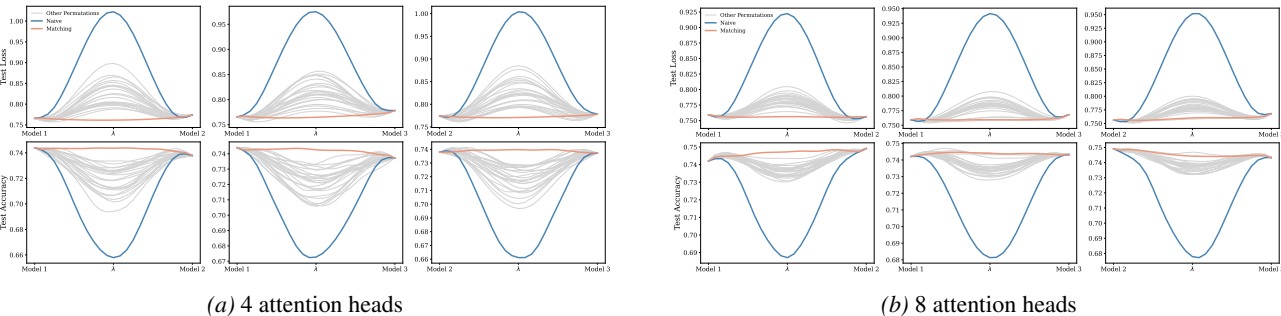

*(a)* 4 attention heads

*(b)* 8 attention heads

*Figure 85.* Linear Mode Connectivity for ViT on CIFAR-10 with 2 layers (all head permutations)

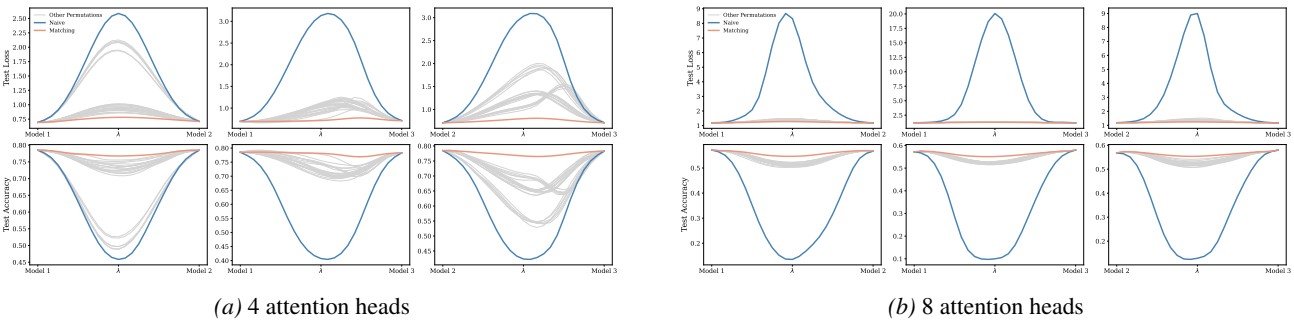

*(a)* 4 attention heads        *(b)* 8 attention heads

*Figure 86.* Linear Mode Connectivity for ViT on CIFAR-10 with 6 layers (all head permutations)

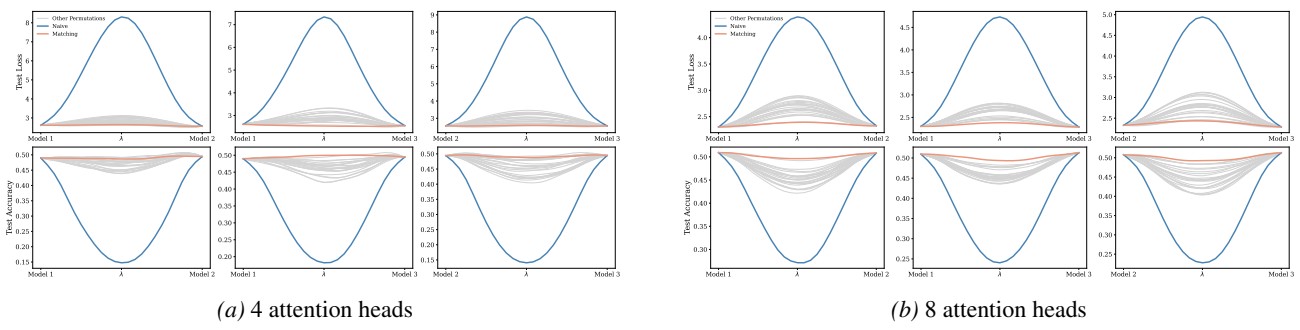

*(a)* 4 attention heads        *(b)* 8 attention heads

*Figure 87.* Linear Mode Connectivity for ViT on CIFAR-100 with 6 layers (all head permutations)

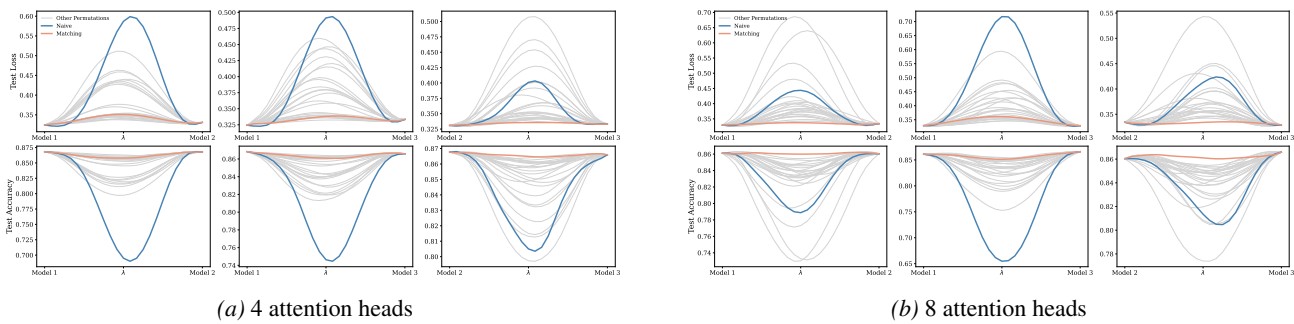

*(a)* 4 attention heads        *(b)* 8 attention heads

*Figure 88.* Linear Mode Connectivity for BERT on IMDBreview with 2 layers (all head permutations)

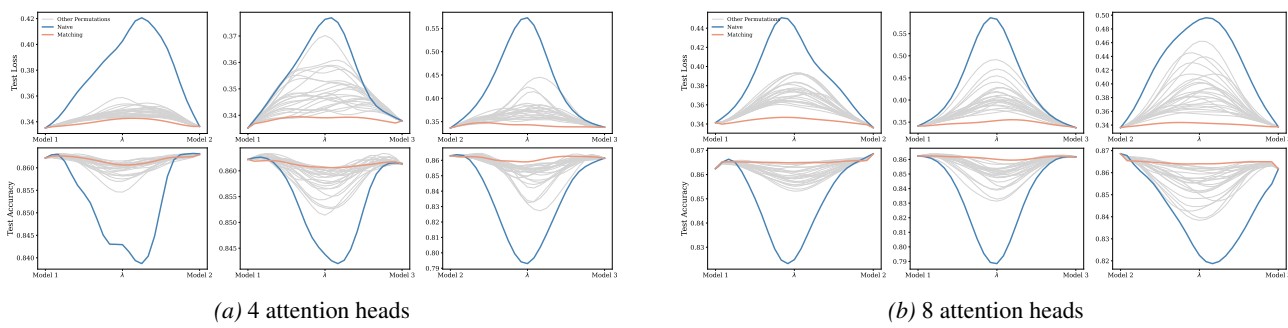

*(a)* 4 attention heads        *(b)* 8 attention heads

*Figure 89.* Linear Mode Connectivity for BERT on IMDBreview with 6 layers (all head permutations)

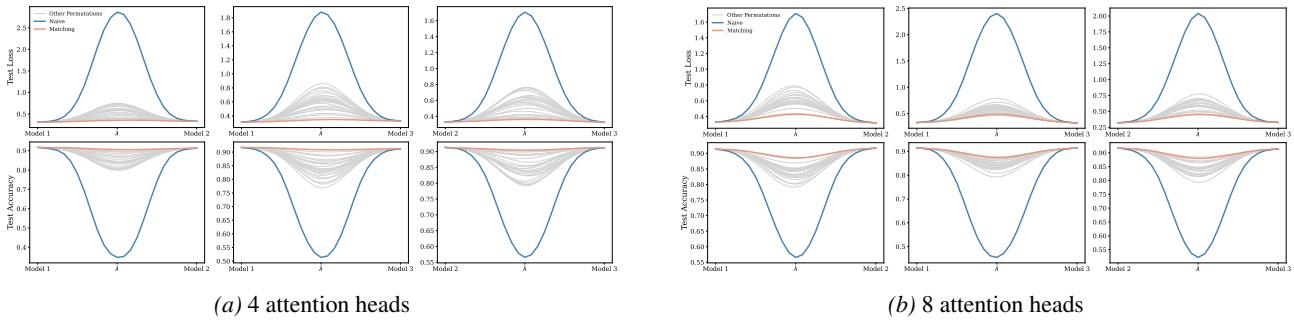

*(a)* 4 attention heads

*(b)* 8 attention heads

*Figure 90.* Linear Mode Connectivity for BERT on DBPedia with 2 layers (all head permutations)

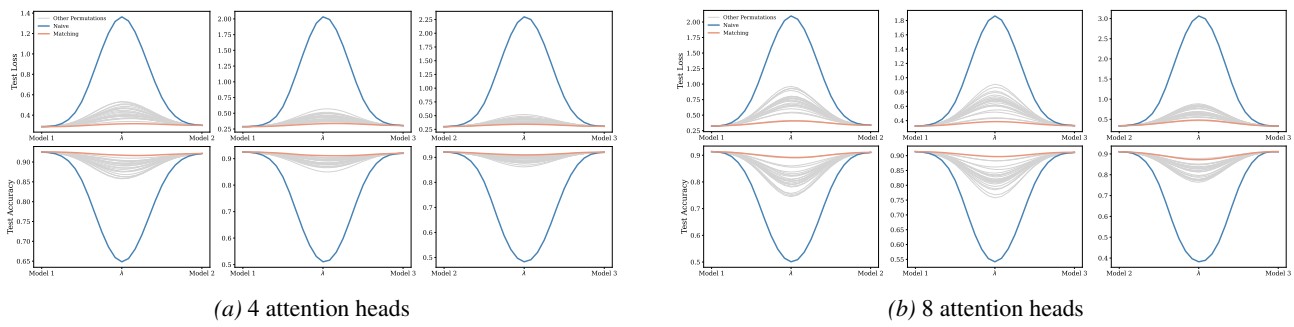

*(a)* 4 attention heads

*(b)* 8 attention heads

*Figure 91.* Linear Mode Connectivity for BERT on DBPedia with 6 layers (all head permutations)

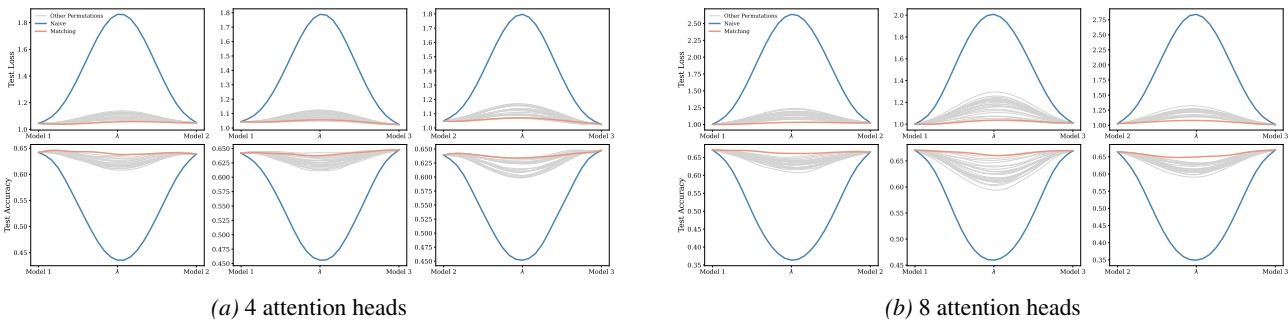

*(a)* 4 attention heads

*(b)* 8 attention heads

*Figure 92.* Linear Mode Connectivity for ViT-RoPE on CIFAR-10 with 2 layers (all head permutations)

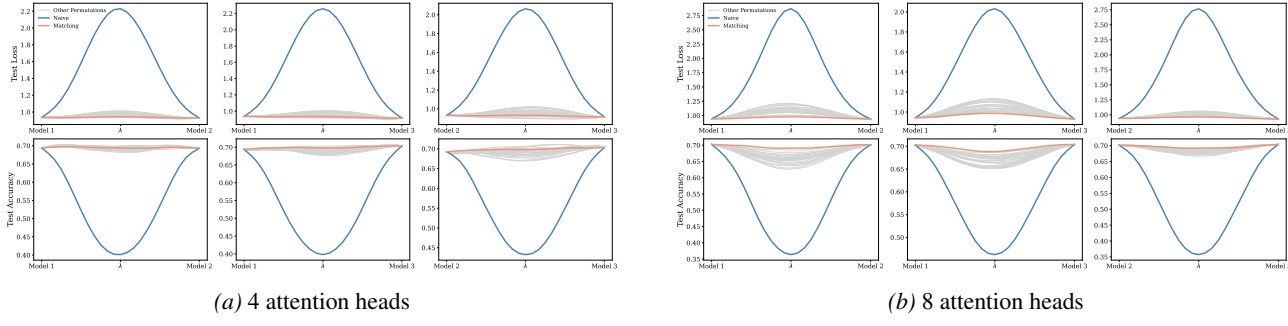

*(a)* 4 attention heads

*(b)* 8 attention heads

*Figure 93.* Linear Mode Connectivity for ViT-RoPE on CIFAR-10 with 6 layers (all head permutations)

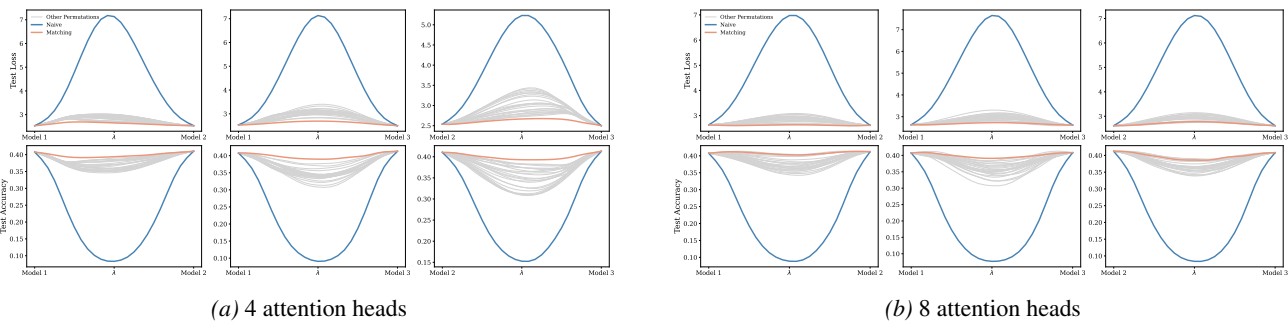

*(a)* 4 attention heads                    *(b)* 8 attention heads

*Figure 94.* Linear Mode Connectivity for ViT-RoPE on CIFAR-100 with 6 layers (all head permutations)

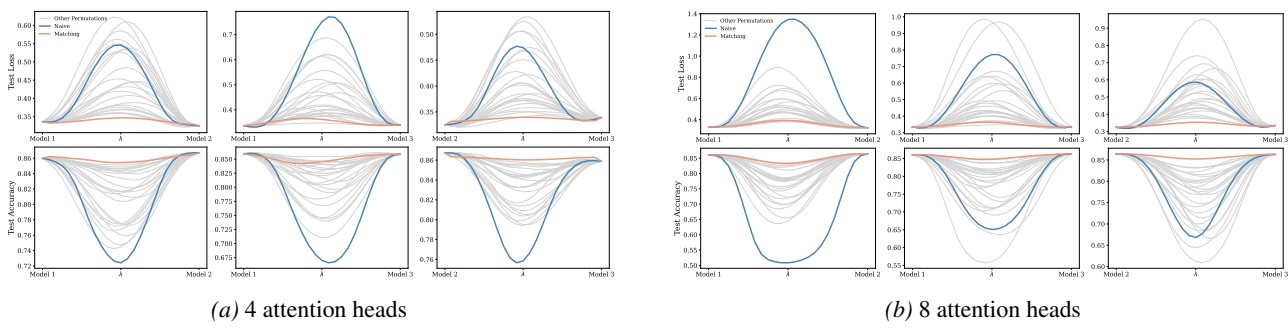

*(a)* 4 attention heads                    *(b)* 8 attention heads

*Figure 95.* Linear Mode Connectivity for BERT-RoPE on IMDBreview with 2 layers (all head permutations)

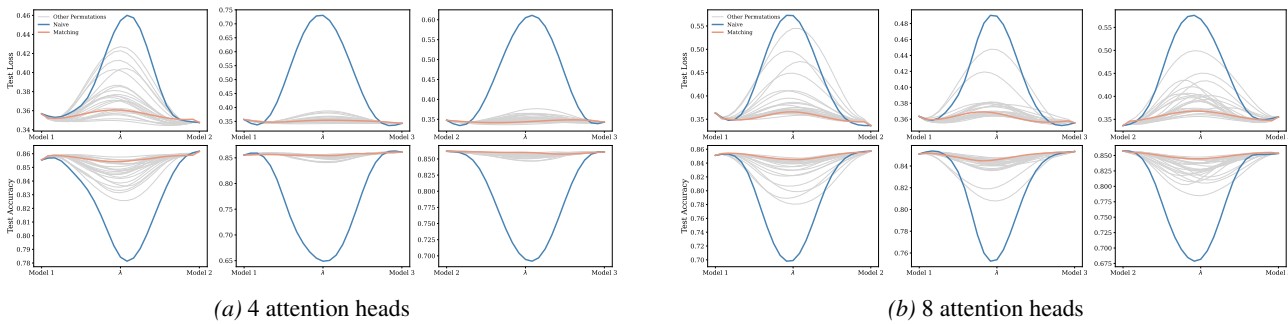

*(a)* 4 attention heads                    *(b)* 8 attention heads

*Figure 96.* Linear Mode Connectivity for BERT-RoPE on IMDBreview with 6 layers (all head permutations)

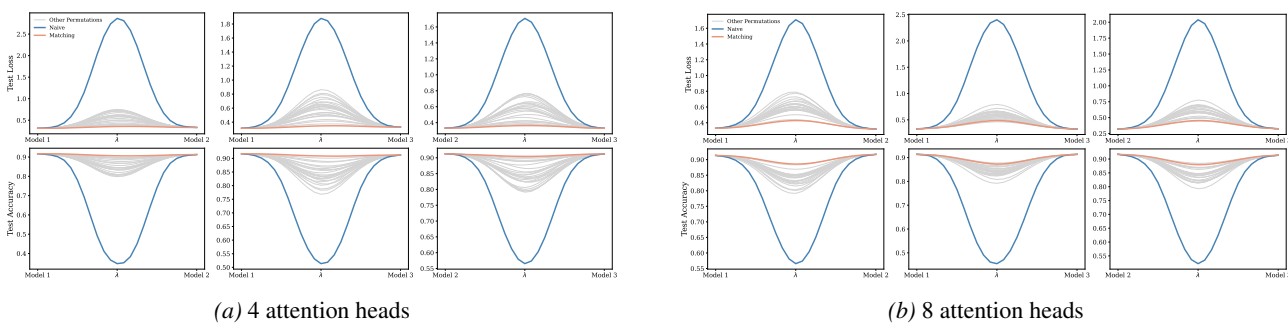

*(a)* 4 attention heads                    *(b)* 8 attention heads

*Figure 97.* Linear Mode Connectivity for BERT-RoPE on DBPedia with 2 layers (all head permutations)

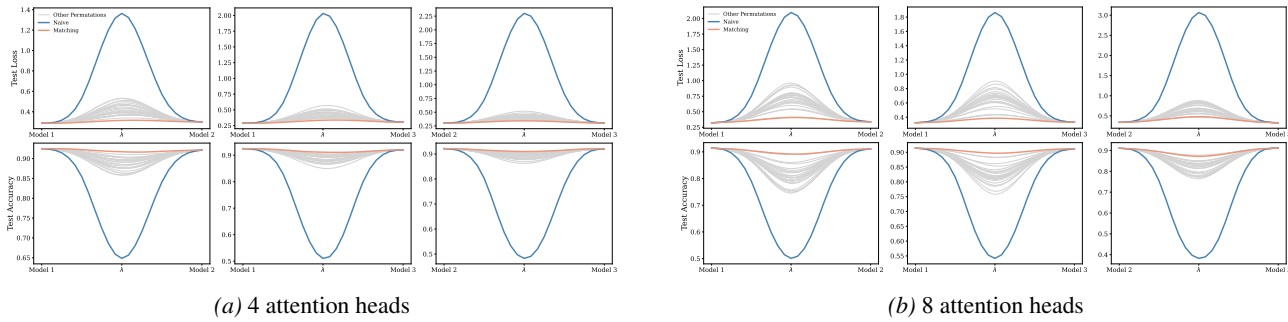

*(a)* 4 attention heads      *(b)* 8 attention heads

*Figure 98.* Linear Mode Connectivity for BERT-RoPE on DBPedia with 6 layers (all head permutations)

## K. Ablation of Distance Metrics for Head Matching

As described in the previous sections, attention-head matching between two transformer models is formulated as a bipartite assignment problem and solved using the Hungarian algorithm. Specifically, a cost matrix is constructed by computing pairwise distances between head $i$ from the first model and head $j$ from the second model. The resulting assignment determines the permutation used for aligning attention heads prior to model matching. This section presents an ablation study on the choice of distance metric used in the cost matrix. Five distance metrics are considered: $\ell_2$, $\ell_1$, cosine distance, correlation distance, and spectral distance. The impact of each metric is evaluated across different matching scopes, including the first attention layer, the first transformer block, the full attention layer, and the full model.

Tables 7, 8, and 9 report results on Wikitext103, OneBillionWord, and ImageNet-1K, respectively. Across all three datasets, $\ell_2$, $\ell_1$, cosine, and correlation distances yield highly similar results across different matching scopes. In contrast, spectral distance consistently leads to degraded performance, particularly when matching larger portions of the model. Among the evaluated metrics, $\ell_2$ distance exhibits the most stable behavior across datasets and matching scopes. Based on these observations, $\ell_2$ distance is used as the default metric for constructing the cost matrix in all remaining experiments.

*Table 7.* Ablation of distance metrics used in the cost matrix for attention-head permutation on Wikitext103. The table reports validation and test loss for different permutation scopes, including the first attention layer, first transformer block, full attention layer, and full model.

| Type | $\ell_2$ | | $\ell_1$ | | cosine | | corr | | spectral | |
| --- | --- | --- | --- | --- | --- | --- | --- | --- | --- | --- |
| | Val Loss | Test Loss | Val Loss | Test Loss | Val Loss | Test Loss | Val Loss | Test Loss | Val Loss | Test Loss |
| First Attention Layer | $3.729 \pm 0.002$ | $3.741 \pm 0.003$ | $3.729 \pm 0.002$ | $3.741 \pm 0.003$ | $3.729 \pm 0.002$ | $3.741 \pm 0.003$ | $3.729 \pm 0.002$ | $3.741 \pm 0.003$ | $3.729 \pm 0.002$ | $3.741 \pm 0.003$ |
| First Transformer Block | $3.773 \pm 0.003$ | $3.789 \pm 0.003$ | $3.773 \pm 0.003$ | $3.789 \pm 0.003$ | $3.773 \pm 0.003$ | $3.789 \pm 0.003$ | $3.773 \pm 0.003$ | $3.789 \pm 0.003$ | $3.773 \pm 0.003$ | $3.789 \pm 0.003$ |
| Full Attention Layer | $3.754 \pm 0.004$ | $3.765 \pm 0.002$ | $3.754 \pm 0.004$ | $3.765 \pm 0.002$ | $3.753 \pm 0.003$ | $3.765 \pm 0.002$ | $3.753 \pm 0.003$ | $3.765 \pm 0.002$ | $3.753 \pm 0.003$ | $3.765 \pm 0.002$ |
| Full Model | $5.178 \pm 0.19$ | $5.172 \pm 0.193$ | $5.202 \pm 0.241$ | $5.197 \pm 0.243$ | $5.203 \pm 0.232$ | $5.196 \pm 0.234$ | $5.203 \pm 0.232$ | $5.196 \pm 0.234$ | $5.203 \pm 0.232$ | $5.196 \pm 0.234$ |

*Table 8.* Ablation of distance metrics used in the cost matrix for attention-head permutation on OneBillionWord. The table reports validation and test loss for different permutation scopes, including the first attention layer, first transformer block, full attention layer, and full model.

| Type | $\ell_2$ | | $\ell_1$ | | cosine | | corr | | spectral | |
| --- | --- | --- | --- | --- | --- | --- | --- | --- | --- | --- |
| | Val Loss | Test Loss | Val Loss | Test Loss | Val Loss | Test Loss | Val Loss | Test Loss | Val Loss | Test Loss |
| First Attention Layer | $1.013 \pm 0.005$ | $1.008 \pm 0.005$ | $1.013 \pm 0.005$ | $1.008 \pm 0.005$ | $1.013 \pm 0.005$ | $1.008 \pm 0.005$ | $1.013 \pm 0.005$ | $1.008 \pm 0.005$ | $1.023 \pm 0.007$ | $1.018 \pm 0.007$ |
| First Transformer Block | $1.054 \pm 0.044$ | $1.046 \pm 0.038$ | $1.054 \pm 0.044$ | $1.046 \pm 0.038$ | $1.057 \pm 0.043$ | $1.049 \pm 0.038$ | $1.057 \pm 0.043$ | $1.049 \pm 0.038$ | $1.05 \pm 0.025$ | $1.042 \pm 0.019$ |
| Full Attention Layer | $1.046 \pm 0.009$ | $1.039 \pm 0.01$ | $1.047 \pm 0.008$ | $1.04 \pm 0.01$ | $1.048 \pm 0.016$ | $1.041 \pm 0.017$ | $1.048 \pm 0.016$ | $1.041 \pm 0.017$ | $1.203 \pm 0.049$ | $1.19 \pm 0.046$ |
| Full Model | $4.18 \pm 0.903$ | $4.174 \pm 0.904$ | $4.088 \pm 0.539$ | $4.089 \pm 0.559$ | $4.137 \pm 0.78$ | $4.13 \pm 0.782$ | $4.141 \pm 0.782$ | $4.133 \pm 0.785$ | $4.303 \pm 0.541$ | $4.297 \pm 0.547$ |

*Table 9.* Ablation of distance metrics for attention-head permutation on ImageNet-1K. The table reports validation loss and top-1 validation accuracy for different permutation scopes, including the first attention layer, full attention layer, and full model.

| Type | $\ell_2$ | | $\ell_1$ | | cosine | | corr | | spectral | |
| --- | --- | --- | --- | --- | --- | --- | --- | --- | --- | --- |
| | Val Loss | Val Accuracy | Val Loss | Val Accuracy | Val Loss | Val Accuracy | Val Loss | Val Accuracy | Val Loss | Val Accuracy |
| First Attention Layer | $0.665 \pm 0.003$ | $84.245 \pm 0.319$ | $0.665 \pm 0.003$ | $84.245 \pm 0.319$ | $0.665 \pm 0.003$ | $84.245 \pm 0.319$ | $0.665 \pm 0.003$ | $84.245 \pm 0.319$ | $0.698 \pm 0.083$ | $82.943 \pm 1.776$ |
| First Transformer Block | $0.691 \pm 0.025$ | $82.161 \pm 0.319$ | $0.697 \pm 0.022$ | $81.641 \pm 0.552$ | $0.689 \pm 0.014$ | $82.292 \pm 1.39$ | $0.689 \pm 0.014$ | $82.292 \pm 1.39$ | $0.686 \pm 0.022$ | $83.203 \pm 0.0$ |
| Full Attention Layer | $1.073 \pm 0.03$ | $70.703 \pm 0.957$ | $1.073 \pm 0.061$ | $70.964 \pm 1.289$ | $1.03 \pm 0.089$ | $71.615 \pm 2.051$ | $1.03 \pm 0.089$ | $71.615 \pm 2.051$ | $2.729 \pm 0.402$ | $39.063 \pm 7.793$ |
| Full Model | $3.063 \pm 0.08$ | $29.818 \pm 6.282$ | $3.214 \pm 0.338$ | $27.604 \pm 6.995$ | $3.167 \pm 0.18$ | $27.865 \pm 5.133$ | $3.167 \pm 0.18$ | $27.865 \pm 5.133$ | $3.99 \pm 0.49$ | $17.448 \pm 6.16$ |

# L. Generalization under Distribution Shifts

To evaluate the generalization ability of matched models under distribution shifts, experiments are conducted on the ImageNet-C benchmark (Hendrycks & Dietterich, 2019). ImageNet-C consists of 15 corruption types applied to the ImageNet validation set, grouped into four categories: noise, blur, weather, and digital corruptions. Each corruption is evaluated at five increasing severity levels, indexed from 1 to 5. In addition, we define level 0 to correspond to the clean ImageNet-1K validation set without any corruption.

This work focuses on the noise category, including Gaussian noise, Shot noise, and Impulse noise, to assess robustness after model matching. Two matching configurations are considered: (i) matching at the first transformer layer, where linear mode connectivity (LMC) exists between the models, and (ii) matching the full transformer model, where linear mode connectivity does not exist between the models. The matched model is obtained by averaging the parameters of two aligned models. Robustness is evaluated across increasing corruption severity levels. All reported results are averaged over three random seeds, with standard deviations reported.

*Table 10.* Robustness evaluation on ImageNet-C (noise corruptions) after matching at the first transformer layer.

| Level | Gaussian Noise | | | | Shot Noise | | | | Impulse noise | | | |
| | Loss | | Acc | | Loss | | Acc | | Loss | | Acc | |
| | Original | Match | Original | Match | Original | Match | Original | Match | Original | Match | Original | Match |
|---|---|---|---|---|---|---|---|---|---|---|---|---|
| 0 | 0.689 ± 0.003 | 0.693 ± 0.018 | 84.668 ± 0.169 | 82.227 ± 0.195 | 0.689 ± 0.003 | 0.693 ± 0.018 | 84.668 ± 0.169 | 82.227 ± 0.195 | 0.689 ± 0.003 | 0.693 ± 0.018 | 84.668 ± 0.169 | 82.227 ± 0.195 |
| 1 | 1.195 ± 0.01 | 1.172 ± 0.008 | 68.75 ± 1.1398 | 69.922 ± 0.391 | 1.254 ± 0.016 | 1.215 ± 0.004 | 68.848 ± 0.697 | 68.555 ± 0.977 | 1.326 ± 0.006 | 1.297 ± 0.016 | 66.016 ± 0.996 | 67.578 ± 0.171 |
| 2 | 1.494 ± 0.01 | 1.465 ± 0.012 | 61.816 ± 0.89 | 62.500 ±0.000 | 1.672 ± 0.051 | 1.555 ± 0.01 | 59.082 ± 0.972 | 59.766 ± 1.172 | 1.705 ± 0.012 | 1.574 ± 0.02 | 56.543 ± 0.697 | 60.938 ± 0.391 |
| 3 | 2.000 ± 0.017 | 1.902 ± 0.035 | 49.609 ± 1.172 | 50.195 ± 1.758 | 2.332 ± 0.007 | 2.211 ± 0.008 | 45.313 ± 0.731 | 48.242 ± 0.977 | 1.934 ± 0.012 | 1.813 ± 0.023 | 51.465 ± 0.697 | 53.711 ± 2.539 |
| 4 | 2.703 ± 0.047 | 2.484 ± 0.031 | 34.766 ± 1.172 | 38.867 ± 2.93 | 3.586 ± 0.055 | 3.367 ± 0.023 | 25.879 ± 1.307 | 29.297 ± 0.391 | 2.734 ± 0.029 | 2.477 ± 0.008 | 35.742 ± 0.977 | 40.039 ± 0.586 |
| 5 | 3.902 ± 0.051 | 3.563 ± 0.001 | 17.969 ± 0.829 | 22.461 ± 0.977 | 4.313 ± 0.038 | 3.992 ± 0.008 | 15.234 ± 0.996 | 21.484 ± 0.781 | 3.723 ± 0.03 | 3.406 ± 0.047 | 21.094 ± 0.829 | 26.172 ± 1.172 |

*Table 11.* Robustness evaluation on ImageNet-C (noise corruptions) after matching the full transformer model.

| Level | Gaussian Noise | | | | Shot Noise | | | | Impulse noise | | | |
| | Loss | | Acc | | Loss | | Acc | | Loss | | Acc | |
| | Original | Match | Original | Match | Original | Match | Original | Match | Original | Match | Original | Match |
|---|---|---|---|---|---|---|---|---|---|---|---|---|
| 0 | 0.656 ± 0.051 | 3.063 ± 0.046 | 85.221 ± 0.474 | 29.688 ± 3.678 | 0.656 ± 0.051 | 3.063 ± 0.046 | 85.221 ± 0.474 | 29.688 ± 3.678 | 0.656 ± 0.051 | 3.063 ± 0.046 | 85.221 ± 0.474 | 29.688 ± 3.678 |
| 1 | 1.421 ± 0.06 | 4.146 ± 0.097 | 63.737 ± 2.214 | 10.026 ±1.605 | 1.522 ± 0.026 | 4.26 ± 0.106 | 64.193 ± 0.487 | 10.286 ± 2.051 | 1.638 ± 0.026 | 4.5 ± 0.088 | 61.654 ± 0.762 | 6.901 ± 1.289 |
| 2 | 1.868 ± 0.054 | 4.865 ± 0.191 | 56.055 ± 2.569 | 3.776 ± 1.025 | 2.065 ± 0.095 | 0.095 ± 0.345 | 53.711 ± 2.962 | 4.036 ± 2.3 | 2.167 ± 0.052 | 5.198 ± 0.232 | 50.846 ± 1.068 | 2.604 ± 1.12 |
| 3 | 2.555 ± 0.083 | 5.635 ± 0.304 | 43.034 ± 1.284 | 2.083 ± 1.328 | 2.901 ± 0.141 | 5.802 ± 0.324 | 36.263 ± 1.52 | 3.906 ± 2.21 | 2.539 ± 0.12 | 5.625 ± 0.301 | 42.122 ± 4.005 | 1.823 ± 1.12 |
| 4 | 3.685 ± 0.185 | 6.396 ± 0.374 | 23.698 ± 0.664 | 0.911 ± 0.737 | 4.406 ± 0.266 | 6.625 ± 0.345 | 13.281 ± 4.442 | 0.521 ± 0.184 | 3.766 ± 0.273 | 6.448 ± 0.379 | 23.047 ± 1.94 | 1.042 ± 0.487 |
| 5 | 5.281 ± 0.199 | 7.167 ± 0.337 | 8.789 ± 3.614 | 0.26 ± 0.184 | 5.354 ± 0.237 | 7.135 ± 0.307 | 8.529 ± 1.648 | 0.911 ± 0.368 | 5.146 ± 0.325 | 7.104 ± 0.329 | 9.375 ± 2.091 | 0.26 ± 0.184 |

Based on Table 10, when linear mode connectivity (LMC) exists between the two models, the matched model exhibits consistently improved robustness compared to the original model under noise corruptions. In particular, for moderate to high corruption severity levels (levels 2–5), the matched model achieves lower validation loss and higher top-1 accuracy across Gaussian, Shot, and Impulse noise. For example, under Gaussian noise at severity level 5, the matched model improves accuracy from 17.97% to 22.46%, while similar trends are observed for Shot noise (15.23% to 21.48%) and Impulse noise (21.09% to 26.17%). These results indicate that model matching under LMC preserves and enhances robustness to distribution shifts. In contrast, Table 11 shows that when linear mode connectivity does not exist between the two models, matching the full transformer model leads to severe degradation in performance even at the clean setting (level 0). The matched model exhibits substantially higher loss and significantly lower accuracy across all corruption types and severity levels, indicating a failure to generalize. As a result, robustness under noise corruptions is not improved in this setting, highlighting the importance of linear mode connectivity for effective model matching.

