# OpenReview forum: "Functional Equivalence in Attention: A Comprehensive Study with Applications to Linear Mode Connectivity"
_ICML.cc/2026/Conference — ICML 2026 regular_

### Official Review · Reviewer_cJKU · 2026-02-23

**Soundness:** 3
**Presentation:** 2
**Significance:** 3
**Originality:** 2
**Overall Recommendation:** 4
**Confidence:** 3

**Summary:**

The paper extends previous results on functional equivalence in attention for attention with positional encodings. The authors study two variants of positional encoding:  Sinusoidal positional encodings, which are shown to have the same symmetry group than the "vanilla" attention model, and rotary positional encodings, which are shown to have a different symmetry group. The latter is the main contribution of the paper. The focus is then changed to study linear mode connectivity. In this domain, the authors study how positional encodings affect linear mode connectivity,

**Compliance With Llm Reviewing Policy:**

Affirmed.

**Key Questions For Authors:**

1. What are the technical differences between the proof of the main result (Theorem 4.2 ) and previous work Theorem 2.1? Since this is a theory paper, I think that even though this proof is long, the authors should try to discuss in the manuscript why this is technically different than the current results.
2. In section 6.1,  the authors write:  "We find that LMC reliably emerges when re-initializing the first attention layer, the first Transformer layer, and all attention layers, with the exception of ImageNet under full attention layer re-initialization." What are the exact plots where one can see this ? How do the authors observe this in the plots ? What experiment should the reader see in order to understand this claim ?

**Limitations:**

Yes.

**Strengths And Weaknesses:**

Strengths:
1. The paper is technically solid, and the theoretical contributions are well written. I didn't went carefully through the proofs, but they seem to be well presented.
2. Extensions and differences from previous work are carefully explained, and the setting for introducing functional equivalence is very well described.
3. On the significance part, the paper definitely provides interesting and novel results that are important in the theoretical understanding of attention models and positional encoding, which are widely used in the present.

Weaknesses:
1. Section 1, the introduction, is not well-connected in general.  The readability seems to be affected by the fact that the authors jump through different topics between different paragraphs, without motivating the reason for this.
2. Even though the results about functional equivalence are very well written, the study (and also algorithm) of LMC is not well motivated from the theoretical results. At the same time, the results from experiments are provided, but almost all of them are left for the appendix of the paper, and the ones in the main (like Figure 2) are difficult to understand with the shown information. I think the tables explaining the experimental set-up could definitely be shown in the appendix, leaving plenty of room for more motivation and explanation.
3.

---

> ### Author Rebuttal · Authors · 2026-03-30
>
> We thank the Reviewer for the response and address the concerns below.
>
> ---
>
> **W1.** We thank the reviewer for this suggestion. The topics raised in Section 1 are all closely related to our study of linear mode connectivity (up to symmetry) in multihead attention. We will revise the manuscript accordingly to improve clarity and readability.
>
> **W2.** The idea of LMC (up to symmetry) can be formulated as follows: can we find a transformation $g$ such that the loss barrier between $\theta_A$ and $g\theta_B$ becomes approximately zero?
>
> At its core, the problem reduces to identifying a suitable $g$. This raises two key issues:
>
> - **Characterizing the search space.** What is the set of transformations $g$ we should consider? Equivalently, what is the full symmetry group of the model $f$?
> - **Designing the matching algorithm.** If the symmetry of $f$ is fully characterized, one can then design an algorithm to search over this set to find a suitable $g$. In this case, even if the algorithm does not find the optimal $g$, it does not overlook any valid symmetry. In contrast, if the assumed symmetry class is incomplete, the correct $g$ may lie outside the search space, rendering the method fundamentally incapable of success.
>
> Our theoretical results address the first issue by providing a complete characterization of the symmetry of general multihead attention.
>
> Regarding the experiments, the purpose of conducting a wide range of evaluations on LMC is to determine in which settings LMC exists. By exploring diverse scenarios, we aim to assess how consistently LMC appears or fails to appear across different conditions. We thank the reviewer for the suggestion and will revise the manuscript accordingly.
>
> **Q1.** The main difference between vanilla attention and RoPE attention lies in the similarity score.
>
> - In vanilla attention, for the $i$-th head, the attention score between the $m$-th and $n$-th tokens is $x_m Q_i K_i^\top x_n^\top$.
> Here, the matrix in the middle is always $Q_i K_i^\top$, which remains unchanged as the indices $m,n$ vary.
>
> - In RoPE attention, for the $i$-th head, the attention score between the $m$-th and $n$-th tokens is $x_m Q_i R_{m-n} K_i^\top x_n^\top$.
> In this case, the middle matrix becomes $Q_i R_{m-n} K_i^\top$, which varies with the relative position $m-n$.
>
> Thus, RoPE attention can be viewed as a nontrivial generalization of vanilla attention (recovering the latter when the rotary matrix $R_{m-n}$ is the identity). As a consequence, previous analyses of functional equivalence for attention, such as [1], no longer directly apply, since they rely on the invariance of the middle matrix across token pairs.
>
> In our work, we further generalize this setting by allowing the middle matrix to depend arbitrarily on the indices $m,n$. While this may appear to complicate the formulation, it is mathematically natural: RoPE attention already introduces index dependence, and our framework captures this phenomenon in full generality. This leads to a cleaner and more unified statement of the main theorem, which is not restricted to the specific structure of RoPE. The functional equivalence results for RoPE attention then follow directly as a special case of our general theory.
>
> We sincerely appreciate this question from the reviewer. In the revision, we will include a high-level summary of the proof in the main text, along with a discussion comparing our results with prior work.
>
> [1] Viet-Hoang Tran et al., Equivariant Neural Functional Networks for Transformers
>
> **Q2.** We thank the reviewer for pointing this out and clarify a typo: the correct statement is *“with the exception of large-scale datasets under full *transformer layer re-initialization,”** rather than *"full attention layer re-initialization"*.. The claim can be observed in Figures 5–77 (Subsections J1–J3), where linear mode connectivity (LMC) is evidenced by the orange curve (Ours matching) remaining close to the endpoints and forming an almost straight interpolation path, consistently below the blue curve (Naive matching). This flat or near-flat interpolation indicates the emergence of LMC after re-initializing the first attention layer, the first transformer layer, or all attention layers. In contrast, for large-scale datasets under full transformer layer re-initialization (Figures 78, 83, and 84), the orange curve shows a pronounced loss increase at intermediate points, breaking linearity despite still reducing the loss barrier.
>
> ---
>
> We thank the Reviewer for the constructive feedback and thoughtful suggestions. If our responses adequately address the concerns, we kindly hope that the evaluation may be adjusted to reflect this. We remain open to further discussion during the next stage of discussion.

---

> > ### Author Rebuttal · Reviewer_cJKU · 2026-04-03
> >
> > My questions have been solved.

---

> > > ### Author Response · Authors · 2026-04-07
> > >
> > > We thank the Reviewer for their feedback and are pleased that our responses have addressed their concerns. If so, we kindly ask that the evaluation be updated accordingly to reflect this.
> > >
> > > In accordance with the policy, while we would be happy to continue the discussion if further questions remain, this marks our final response.
> > >
> > > Best regards,
> > >
> > > Authors

---

### Official Review · Reviewer_6aks · 2026-03-11

**Soundness:** 2
**Presentation:** 1
**Significance:** 2
**Originality:** 2
**Overall Recommendation:** 3
**Confidence:** 5

**Summary:**

The paper studies functional equivalence and linear mode connectivity in Transformers, with a focus on how positional encodings, especially RoPE, change the symmetry structure of attention. Its main claim is that RoPE restricts some of the permutation symmetries present in standard attention, which in turn affects how independently trained Transformer models can be aligned and connected in weight space. Empirically, it combines symmetry-aware matching with extensive LMC experiments across several architectures and settings.

**Compliance With Llm Reviewing Policy:**

Affirmed.

**Final Justification:**

The paper studies an interesting and relevant question about Transformer symmetries, positional encodings, and linear mode connectivity. I see two clear strengths: the theoretical perspective on how RoPE changes the symmetry structure of attention, and the breadth of the experimental effort across architectures and settings.

After considering both the paper and the rebuttal, however, my overall recommendation remains Reject. The rebuttal clarified that the main novelty lies in the symmetry characterization for MHA with RoPE, and I agree this is the paper’s strongest contribution. Still, in my view, the overall contribution remains somewhat incremental relative to prior work on symmetry, matching, and LMC, with the main new element being the RoPE-specific extension.

My main remaining concerns are about scope and presentation. The authors acknowledged that broader conclusions about full-model LMC should be more carefully qualified, which partially addresses my concern, but it also reinforces that the current presentation can overstate what is supported by the theory. Likewise, I appreciate that the authors agreed aggregated summaries would improve the paper, but in the submitted version the empirical section is still difficult to interpret because the evidence is spread across many per-seed figures with limited synthesis.

**Key Questions For Authors:**

- **Q1**: Could the authors provide aggregated statistics (e.g., mean ± std across seeds) summarizing the key experiments?
A summary table comparing LMC success rates across positional encodings, datasets and architectures would make the results significantly easier to interpret.
- **Q2**: How do additional Transformer symmetries (e.g., residual stream transformations) affect the interpretation of the experimental results when the entire network is reinitialized?
- **Q3**: Can the authors clarify the theoretical relationship between the symmetry structure they analyze and the observed LMC behavior?
Is there any formal or empirical evidence that the reduced symmetry induced by RoPE directly affects LMC?
- **Q4**: The paper suggests that causal attention inherently leads to higher loss barriers. However, the compared models (e.g., GPT-style vs. BERT-style) differ not only in the attention mask but also in training objective and dataset complexity. Could the authors clarify why the observed effect can be attributed specifically to causal attention rather than these other factors? It would be particularly interesting if this effect could be demonstrated in a controlled ablation where only the attention mask differs while the dataset, architecture, and training objective are kept fixed.

**Limitations:**

yes

**Strengths And Weaknesses:**

## Strengths
- **S1: interesting theoretical perspective on positional encodings.** The paper examines the symmetry of attention with respect to different positional encoding schemes, in particular RoPE.
- **S2: Extensive experimental evaluation.** The authors perform a diverse set of experiments on different Transformer architectures, datasets, seeds, and re-initialization strategies.

## Weaknesses
- **W1: Limited novelty.** The core ideas of functional equivalence, weight permutation symmetries, and linear mode connectivity have been studied extensively in prior work. The main new contribution here appears to be extending these analyses to the case of RoPE (APE have been covered in [1,2,3], among others). The conceptual novelty seems incremental, and it is not fully clear how the results meaningfully change our understanding of optimization or model behavior in practice.
- **W2: The theoretical analysis only considers a subset of Transformer symmetries.** The proposed algorithm only addresses the symmetries in the attention mechanism. While this is fine when only re-initializing the attention weights, it becomes problematic when  analyzing LMC when the entire model is re-initialized, as symmetries exist for the residual stream, and the MLP blocks.
- **W3: Difficulty interpreting the results.** The appendix contains a very large number of figures (close to 100), many of which correspond to individual seeds or slightly different experimental configurations. While this demonstrates substantial experimental effort, the presentation makes it difficult to extract the main empirical findings. The paper would benefit significantly from summarizing these experiments using aggregated statistics (e.g., mean and standard deviation across seeds) and presenting summary tables in the main text.
- **W4: Lack of clear synthesis of experimental findings.** The paper presents many experimental results but provides limited synthesis or high-level conclusions. It is difficult for the reader to determine what the main empirical takeaways are regarding the effect of positional encodings on LMC or functional equivalence.
- **W5: Matching algorithm not entirely novel.** While the alignment of the weights for the individual heads is novel, the alignment with respect to the order of attention heads is the same as in [3] to the best of my knowledge. It should be cited accordingly.
- **W6: Unclear finding regarding causal attention.** The paper claims that causal attention models exhibit higher loss barriers and suggests this is inherent to causal attention. However, the experiments do not isolate causal attention as the relevant factor. The compared models differ in several other aspects, including training objective and dataset complexity (e.g., GPT-style vs BERT-style training). Moreover, similar loss barriers are observed in other settings such as ViTs on ImageNet. It is therefore unclear whether the observed effect is truly attributable to causal attention.

## References
[1] Verma, N., & Elbayad, M. (2024). *Merging Text Transformer Models from Different Initializations*

[2] Rinaldi, F., Capitani, G., Bonicelli, L., Crisostomi, D., Bolelli, F., Ficarra, E., Rodolà, E., Calderara, S., & Porrello, A. (2025). *Update Your Transformer to the Latest Release: Re-Basin of Task Vectors*.

[3] Theus, A., Cabodi, A., Anagnostidis, S., Orvieto, A., Singh, S. P., & Boeva, V. (2025). *Generalized Linear Mode Connectivity for Transformers*.

---

> ### Author Rebuttal · Authors · 2026-03-31
>
> **W1.** We respectfully disagree that our work is incremental relative to [1,2,3]. While related in topic, our contribution is fundamentally different: we provide a characterization of symmetry for multihead attention (MHA) (Theorems 4.1-4.2). In contrast, weight matching and LMC observations are applications of this result, not the main contribution.
>
> *No prior work* establishes a full characterization of MHA with RoPE. Existing works identify specific invariances or propose empirical alignment methods, but do not address the “only-if” direction—whether a given set of transformations exhausts all symmetries.
>
> This distinction is crucial. **While symmetries such as head permutations and within-head linear actions are easy to verify, proving they are the only symmetries is substantially more difficult and requires a complete characterization**. Such results are central in prior literature (e.g., FNNs, MoEs, vanilla MHA), but have remained open for MHA+RoPE. Our work fills this gap.
>
> **Q2+W2.** We focus exclusively on the symmetry of MHA with the two most widely used PE: sinusoidal PE and RoPE. Accordingly, we take a pretrained model $A$, freeze components unrelated to attention, and independently train the attention blocks in two runs to obtain $B$ and $C$. We then apply our matching method to these components. This protocol is standard, introduces no bias from $A$, and aligns with prior work on LMC, while isolating the role of attention.
>
> **Matching full models.** There is currently *no prior work* that characterizes the symmetry of architectures of stacked blocks. Existing results are limited to individual layers. Without such a result, there is no principled way to design matching methods at the full-model level. Moreover, our experiments already show cases where LMC fails to exist even within attention blocks, suggesting that LMC need not hold in general and may also fail for full models.
>
> **Why not analyze full-model symmetry.** As in **W1**, this problem is highly challenging, and no prior work has successfully characterized the symmetry of even a few stacked layers, let alone an entire model.
>
> **Q1+W3+W4.** We believe the reviewer’s concern stems from a misunderstanding of how to interpret our experiments, particularly the role of the loss barrier and the use of mean and std.
> - **Loss barrier.** Our goal is to determine whether LMC exists, not to optimize a metric. The loss barrier is used only as an indicator: a near-zero value suggests the presence of LMC. It is therefore not a benchmark for comparison, and reporting mean/std across methods is not meaningful. The key question is the existence of a zero-barrier path, not how small the barrier is.
> - **Mean and standard deviation.** Multiple runs are used solely to ensure robustness and avoid cherry-picking. For each setting, we perform three independent runs, obtain three model pairs, and check whether LMC occurs in each case. The conclusion is qualitative and consistent—whether LMC appears or not—rather than a statistical estimate of performance.
>
> If needed, we can provide the mean and std of loss barrier in the next discussion.
>
> **W5.** We thank the Reviewer and will cite accordingly.
>
> **Q4+W6.** We acknowledge this point in the limitations. Our reasoning is as follows. In some cases, LMC does not fully exist, as indicated by a noticeably higher loss barrier (e.g., Figure 55), where the loss curve deviates from the near-linear behavior. Empirically, these cases predominantly correspond to models with causal attention.
>
> We attribute this discrepancy to an incomplete understanding of causal attention symmetry. While our characterized symmetries for vanilla attention also apply, their exhaustiveness remains unknown. Establishing this “only-if” direction is significantly more challenging, as discussed in our responses to **W1** and **Q2+W2**.
>
> In particular, causal attention may admit additional symmetries beyond those we characterize. Since our algorithm depends on this symmetry structure, missing symmetries may prevent it from finding an optimal $g$, leading to an apparent absence of LMC.
>
> **Q3.** Our results on the symmetry of MHA are used to ensure that the proposed algorithm is complete, in the sense that it does not miss any symmetry. This completeness is crucial: without a full characterization of symmetry, any matching-based procedure may fail to align models, leading to incorrect conclusions about the presence or absence of LMC.
>
> We observe that cases where LMC does not fully exist occur mostly in RoPE settings. We treat this as empirical evidence supporting the claim that architectural differences in symmetry structure affect the existence of LMC.
>
> Kindly see our response to **W2** of Reviewer cJKU for details on the link between symmetry and our algorithm.
>
> ---
>
> We thank the Reviewer for the feedback. If our responses address the concerns, we kindly hope the evaluation can be updated accordingly. We remain happy to continue the discussion.

---

> > ### Author Rebuttal · Reviewer_6aks · 2026-04-04
> >
> > I thank the authors for their response.
> >
> > First, regarding full-model or broader Transformer conclusions, I still think the scope needs to be stated more carefully. While the rebuttal emphasizes that the theory is restricted to attention, the paper reports experiments beyond isolated attention heads, including first-Transformer-layer and full-model settings. In these broader experiments, the method already incorporates standard feed-forward matching, so it is reasonable to ask why other known relevant symmetries, such as residual-path symmetry, are not similarly accounted for. My point was not that the paper must solve the full open problem of stacked-block symmetry, but that the interpretation of these broader LMC results should be qualified accordingly.
> >
> > Second, I do not find the argument against aggregated summaries persuasive. Even if LMC is ultimately interpreted as a qualitative property, the paper evaluates it through a continuous proxy, the loss barrier, and in practice one must decide what counts as “negligible.” This makes aggregate summaries across seeds and settings informative. Moreover, the paper itself already reports averaged barrier-based quantities and variability in Table 1, so aggregation is clearly meaningful within the authors’ own methodology. My request was not to replace qualitative conclusions with a benchmark score, but to summarize the empirical evidence in a way that makes the claimed trends easier to interpret, as they are currently spread across a very large number of figures. Along the same lines, I recommend replacing many of the per-seed figures with aggregated plots, for example mean interpolation curves with variability bands across seeds. Given the very large number of current figures, this would make the main empirical trends much easier to extract.

---

> > > ### Author Response · Authors · 2026-04-07
> > >
> > > We thank the Reviewer for continuing the discussion and address the concerns as follows.
> > >
> > > ---
> > >
> > > > First, regarding full-model or broader Transformer conclusions, I still think the scope needs to be stated more carefully. While the rebuttal emphasizes that the theory is restricted to attention, the paper reports experiments beyond isolated attention heads, including first-Transformer-layer and full-model settings. In these broader experiments, the method already incorporates standard feed-forward matching, so it is reasonable to ask why other known relevant symmetries, such as residual-path symmetry, are not similarly accounted for. My point was not that the paper must solve the full open problem of stacked-block symmetry, but that the interpretation of these broader LMC results should be qualified accordingly.
> > >
> > > **Answer.** We thank the Reviewer for the suggestion. In this work, we primarily focus on aligning attention components, and to a limited extent, additional transformer modules. However, we acknowledge that certain parts of the paper may give the impression that our results extend to the existence or non-existence of LMC in transformer-based models more broadly, for instance, in the Limitations and Future Work section.
> > >
> > > We agree with the Reviewer that such broader interpretations should be carefully qualified. We will revise the manuscript accordingly to ensure that these claims are properly scoped and clearly stated. Should the paper be accepted, we will incorporate all necessary revisions to reflect this clarification.
> > >
> > > >Second, I do not find the argument against aggregated summaries persuasive. Even if LMC is ultimately interpreted as a qualitative property, the paper evaluates it through a continuous proxy, the loss barrier, and in practice one must decide what counts as “negligible.” This makes aggregate summaries across seeds and settings informative. Moreover, the paper itself already reports averaged barrier-based quantities and variability in Table 1, so aggregation is clearly meaningful within the authors’ own methodology. My request was not to replace qualitative conclusions with a benchmark score, but to summarize the empirical evidence in a way that makes the claimed trends easier to interpret, as they are currently spread across a very large number of figures. Along the same lines, I recommend replacing many of the per-seed figures with aggregated plots, for example mean interpolation curves with variability bands across seeds. Given the very large number of current figures, this would make the main empirical trends much easier to extract.
> > >
> > > **Answer.** We thank the Reviewer for the thoughtful suggestion. We agree that, although LMC is ultimately interpreted as a qualitative property, it is evaluated in practice through quantitative proxies such as the loss barrier, and therefore aggregated summaries across seeds and settings are both meaningful and informative.
> > >
> > > While prior works often rely on per-seed visualizations to demonstrate the existence of LMC, we appreciate the Reviewer’s point that aggregated statistics (e.g., mean interpolation curves with variability bands, or summary measures of loss barriers) provide a clearer and more rigorous view of the empirical trends. We also acknowledge that our current presentation, with a large number of per-seed figures, makes it difficult to extract the main conclusions efficiently.
> > >
> > > In the revised version, we will incorporate aggregated summaries of the loss barrier and, where appropriate, replace multiple per-seed plots with consolidated visualizations (e.g., mean curves with variability bands across seeds). This will improve clarity and better highlight the key empirical findings.
> > >
> > > We find this suggestion particularly valuable and will ensure that the corresponding revisions are included in the final version of the paper if accepted.
> > >
> > > ---
> > >
> > > We thank the Reviewer for their feedback and are pleased that our responses have addressed their concerns. If so, we kindly ask that the evaluation be updated accordingly to reflect this.
> > >
> > > In accordance with the policy, while we would be happy to continue the discussion if further questions remain, this marks our final response.
> > >
> > > Best regards,
> > >
> > > Authors

---

### Official Review · Reviewer_NN1P · 2026-03-13

**Soundness:** 3
**Presentation:** 2
**Significance:** 3
**Originality:** 3
**Overall Recommendation:** 5
**Confidence:** 3

**Summary:**

This paper studied functional equivalence in attention layers, particularly, it studied variants of positional encoding approaches on how it affect the symmetry property of attention. It discovered that RoPE could reduce the symmetry, and the linear mode connectivity appears across a wide range of applications. This study include a comprehensive theoretical studies and experiments on large models and real-world datasets.

**Compliance With Llm Reviewing Policy:**

Affirmed.

**Key Questions For Authors:**

1. How is the symmetry affected by other factors, such as learning hyperparameters, training procedures, architecture deign choices?

**Limitations:**

It already includes discussions how it might fail in a larger model, it will be better if more discussion on what are potential mismatch between the theoretical proofs and empirical studies.

**Strengths And Weaknesses:**

1. Originality: This paper introduce a novel aspect and deep analysis on the functional equivalence of attention mechanism, especially on different positional encoding approaches’ affect on symmetry properties. It includes a comprehensive overview of existing works, in symmetry analysis and simpler models, this work has extended the scope into transformer and more precisely on the roles of positional encoding approaches.
2. Soundness: This work had an extensive detail on theoretical proofs and comprehensive experiments on different scales of models, and varied kinds of tasks.
3. Significance: The symmetry mechanisms is an important mechanisms, and the work is applied into real-world applications and large-scale modes, and the results are overall generalized, and provide implications for future downstream applications.
4. Presentation: The paper is overall well-structured, while the theoretical parts and the appendix are relatively dense to follow and digest, providing more intuitions and explanations will be helpful. Some major results and analysis are not included in the main papers.

---

> ### Author Rebuttal · Authors · 2026-03-30
>
> We thank the Reviewer for the response and address the concerns below.
>
> ---
>
> **Q1.** The symmetry arises purely from the architecture and remains invariant under any training scheme. In particular, learning hyperparameters and training procedures neither affect nor relate to the symmetry of the model.
>
> Regarding architectural design choices, when stacking neural blocks—such as feedforward networks, attention layers, and mixture-of-experts—the behavior of symmetry can be summarized as follows:
>
> - **Persistence of block-level symmetry.** The symmetry of each individual block is preserved within the full model. Consequently, the overall model inherits these symmetries, and understanding the symmetry of each component provides partial insight into the symmetry of the entire architecture.
> - **Emergent symmetry from composition.** Conversely, additional symmetries may arise from the interaction between stacked blocks that are not present at the level of a single block. To the best of our knowledge, there is currently no prior work that systematically characterizes the symmetry of multi-layer compositions.
>
> ---
>
> **Limitations. It already includes discussions how it might fail in a larger model, it will be better if more discussion on what are potential mismatch between the theoretical proofs and empirical studies.**
>
> **Answer.** Our theoretical results characterize the symmetry of multi-head attention. These results stand independently and provide a foundation for designing weight-matching methods, which are subsequently used to study linear mode connectivity (LMC). The potential non-existence of LMC in larger models arises from properties of the loss landscape, rather than from the symmetry characterization. We emphasize this distinction to clarify that our theoretical results are correct.
>
> ---
>
> We thank the Reviewer for the constructive feedback and thoughtful suggestions. If our responses adequately address the concerns, we kindly hope that the evaluation may be adjusted to reflect this. We remain open to further discussion during the next stage of discussion.

---

> > ### Author Rebuttal · Reviewer_NN1P · 2026-04-04
> >
> > Thanks for the additional clarifications, my concerns are mostly addressed. I will keep my original score as accept.

---

> > > ### Author Response · Authors · 2026-04-07
> > >
> > > We thank the Reviewer for their feedback and are pleased that our responses have addressed their concerns.
> > >
> > > Best regards,
> > >
> > > Authors

---

### Official Review · Reviewer_WUeY · 2026-03-13

**Soundness:** 3
**Presentation:** 3
**Significance:** 3
**Originality:** 3
**Overall Recommendation:** 5
**Confidence:** 3

**Summary:**

The paper aims to expand our understanding of linear mode connectivity (LMC) in transformers. A key contribution of the paper is that it explicitly takes the positional encodings used by the models into account, and characterizes the symmetries of multihead attention layers in the case of rotary positional encodings. Additionally, the authors present a weight matching algorithm inspired by Ainsworth et al. Empirically, the paper shows across a range of datasets and architectures that LMC emerges consistently in encoder-only models, but may fail in decoder-only models for large-scale language modeling.

**Compliance With Llm Reviewing Policy:**

Affirmed.

**Final Justification:**

The extension of the symmetry analysis to RoPE is, in my view, an important and useful contribution, even if its main impact is likely to be within a relatively specialized research area. I also consider it important that the clarifications and discussions provided in the rebuttal be incorporated into the paper. With these clarifications included, I find the paper acceptable for publication and believe it provides value to the community. Therefore, I raise my score to Accept.

**Key Questions For Authors:**

As I understand, you chose not to incorporate the orthogonal symmetries used by Theus et. al, can you elaborate on why you chose not to?

Did you consider matching activations instead of weights?

Could you elaborate more on the fine-tuning and re-initialization used in your methods?

**Limitations:**

Yes.

**Strengths And Weaknesses:**

Strengths

- The paper advances our understanding of linear mode connectivity (LMC) in transformers.
- It offers a valuable characterization of the symmetries arising in the RoPE setting.
- The empirical study includes thorough ablations across multiple datasets and architectures.

Weaknesses

- Although the paper is generally well written and introduces LMC and prior work in a way that is accessible to readers who are not already immersed in the area, some later methodological details—especially in Section 6—become difficult to follow. In particular, I found the role of re-initialization and fine-tuning somewhat unclear. My understanding is that the paper does not study standard LMC between two independently trained models; rather, the authors begin from a pretrained backbone, re-initialize selected attention or Transformer blocks, freeze the remaining parameters, and optimize only the re-initialized blocks to obtain the endpoints. In addition, Stage 2 appears to further optimize the alignment variables, and this procedure is also referred to as “fine-tuning” in parts of the text. Using the same term for these conceptually distinct procedures makes the methodology harder to parse. Relatedly, this setup differs from standard vanilla LMC, which typically considers independently trained endpoints rather than models fine-tuned from a shared pretrained backbone. This is not inherently problematic, but it should be clearly stated and discussed, as the current presentation may otherwise be misleading to the reader. I would therefore encourage the authors to clarify these details more explicitly in the main text.
- The paper only compares LMC barriers among the methods it proposes, and does not include comparisons to prior work such as Theus et al.

---

> ### Author Rebuttal · Authors · 2026-03-30
>
> We thank the Reviewer for the response and address the concerns below. For clarity and coherence, we found it appropriate to merge certain related weaknesses and questions and provide unified answers.
>
> ---
>
> **W1+Q3.** The connection between the symmetry of an architecture and the existence of LMC (up to symmetry) is both tight and fundamental. Understanding the symmetry of an architecture ensures that a matching method derived from this structure is optimal, and consequently, that any observation of LMC is reliable.
>
> In this paper, we focus exclusively on the symmetry of the Attention Block, considering the two most widely used positional encoding variants: sinusoidal PE and RoPE. Accordingly, we take a pretrained model $A$, freeze all components unrelated to attention, and independently train the remaining attention blocks in two separate runs to obtain $B$ and $C$. We then apply our proposed matching method to these components. This procedure is valid and introduces no bias from the original model $A$. Our study is restricted to the attention component, and our experimental protocol is consistent with prior work on LMC.
>
> Regarding the matching of two entirely different models and the observation of LMC, we note that there is currently no prior work analyzing the symmetry of full models composed of stacked blocks. Existing studies on neural architecture symmetry are limited to individual components, such as feedforward networks, attention, and mixture-of-experts. Without an understanding of the symmetry induced by stacking these blocks, there is no principled basis for designing a matching method at the level of the full model. Furthermore, our experiments include cases where LMC does not exist—even when restricted to attention blocks—indicating that LMC need not exist in general, and therefore may also fail to exist at the level of the full model.
>
> We thank the reviewer for this question. While we briefly described the freezing and fine-tuning procedure in the main text, we agree that the explanation can be improved, and we will revise the experimental section accordingly.
>
> **W2.** In this line of work, LMC is a phenomenon of the loss landscape whose existence is to be determined. The loss barrier serves as an indicator of this phenomenon: when it is approximately zero, it suggests the existence of LMC. Thus, the loss barrier is used to assess whether LMC exists, rather than as a metric for comparison. Consequently, comparing methods based on the loss barrier is not necessary in this context. We do not aim to outperform existing approaches; rather, our goal is simply to determine whether LMC exists.
>
> **Q1.** Theus et al. incorporate orthogonal symmetries within the residual architecture to design their algorithm. In contrast, our work focuses on theory-driven results and algorithms. As we currently lack a characterization of the symmetry of residual schemes, it is difficult to assert that any proposed method for such architectures would be reliable. While verifying that certain orthogonal transformations preserve the model under a residual scheme is relatively straightforward, establishing that these transformations exhaust all possible symmetries is a substantially more challenging problem.
>
> Furthermore, in our experiments restricted to RoPE-based attention layers, we observe cases where the loss barrier is noticeably higher than usual (e.g., Figure 55, where the loss curve deviates from the near-linear behavior seen in other figures), indicating that LMC does not fully exist. Since these experiments involve only attention heads, no matching over residual schemes is required; nevertheless, LMC can still fail. We emphasize that this observation does not contradict the results in [1], where LMC is shown to exist in attention, as the datasets differ. However, our findings suggest that on large-scale datasets, LMC does not necessarily exist in general.
>
> **Q2**. If we understand correctly, the reviewer refers to activations as the outputs of each layer in the model. At present, we have not explored any activation-based methods, which are inherently data-dependent. Our work instead focuses on a matching algorithm derived from our theoretical analysis of the symmetry of attention. We consider activation-based approaches to be an interesting complementary direction for studying LMC, and a promising avenue for future work.
>
> ---
>
> We thank the Reviewer for the constructive feedback and thoughtful suggestions. If our responses adequately address the concerns, we kindly hope that the evaluation may be adjusted to reflect this. We remain open to further discussion during the next stage of discussion.

---

> > ### Author Rebuttal · Reviewer_WUeY · 2026-04-04
> >
> > I thank the authors for their responses. I consider it important that the clarifications about the procedure are clearly and explicitly incorporated into the manuscript.
> >
> > I understand the authors’ intention to focus on theory-driven results and algorithms, however, I still have some remaining questions regarding the justification for not considering full-model alignment. In particular, it seems that one could in principle attempt to match full models as well, even without a complete symmetry characterization, and compare the resulting loss barriers as a measure of success for the alignment. At the same time, I would like to emphasize that I am not suggesting that such experiments are required, but rather that the rationale for this design choice should be clearly articulated.
> >
> > Finally, I would appreciate further clarification on the statement from the authors' rebuttal: "This protocol is standard and introduces no bias from A." In particular, in what precise sense this "no bias" should be interpreted, as the shared frozen backbone appears to introduce a non-trivial bias.
> >
> > I would like to emphasize that my intention is not to overstate these limitations, but rather to encourage a clearer presentation of the paper’s contribution and limitations.

---

> > > ### Author Response · Authors · 2026-04-07
> > >
> > > We thank the Reviewer for continuing the discussion and address the concerns as follows.
> > >
> > > ---
> > >
> > > > I understand the authors’ intention to focus on theory-driven results and algorithms, however, I still have some remaining questions regarding the justification for not considering full-model alignment. In particular, it seems that one could in principle attempt to match full models as well, even without a complete symmetry characterization, and compare the resulting loss barriers as a measure of success for the alignment. At the same time, I would like to emphasize that I am not suggesting that such experiments are required, but rather that the rationale for this design choice should be clearly articulated.
> > >
> > > **Answer.** We thank the Reviewer for the suggestion. In this work, we primarily focus on aligning attention components, and to a limited extent, additional transformer modules. We therefore appreciate the Reviewer’s request for full-model alignment experiments, which would indeed strengthen the completeness of the study.
> > >
> > > While we agree that, in principle, full-model alignment could be attempted even without a complete symmetry characterization, conducting such experiments requires substantial computational resources and careful design. Due to the limited time in the rebuttal phase, we are unable to include these additional experiments at this stage.
> > >
> > > Nevertheless, we would like to emphasize that our current experiments on attention components already serve to validate the necessity and effectiveness of the proposed theoretical framework. Should the paper be accepted, we will incorporate full-model alignment experiments in the revised version to further strengthen the empirical evaluation.
> > >
> > > > Finally, I would appreciate further clarification on the statement from the authors' rebuttal: "This protocol is standard and introduces no bias from A." In particular, in what precise sense this "no bias" should be interpreted, as the shared frozen backbone appears to introduce a non-trivial bias.
> > >
> > > **Answer.** Our primary objective is to study alignment at the level of attention components. To this end, we adopt the following protocol:
> > >
> > > - We begin with a model $A = f_1 \circ MHA \circ f_2$, and train it to convergence.
> > > - We then freeze the parameters of $f_1$ and $f_2$. Keeping these components fixed, we reinitialize the parameters of the MHA module and retrain only this module, producing two independently trained models $B$ and $C$.
> > > - As a result, $A,B,C$ share identical parameters for $f_1$ and $f_2$, while the MHA components in $B$ and $C$ are independently trained.
> > >
> > > *Why this protocol is appropriate for studying LMC in attention.*
> > > We agree that our statement in previous rebuttal (“this protocol is standard and introduces no bias from $A$”) was too brief and could be misleading. Indeed, $B$ and $C$ inherit shared context through $f_1$ and $f_2$. However, from the perspective of LMC over the MHA component, this setup is well-justified:
> > >
> > > - With $f_1$ and $f_2$ fixed, the composite model $f_1 \circ MHA \circ f_2$ can be viewed as a parameterization solely in terms of the MHA module. Thus, retraining the MHA component from different initializations yields two independent solutions in the same parameter space, which is the standard setting in LMC studies.
> > > - Empirically, naive linear interpolation between the MHA parameters of $B$ and $C$ results in a significantly high loss barrier. This indicates that the two solutions are nontrivially separated, and therefore suitable for evaluating alignment methods that aim to uncover low-loss connecting paths.
> > >
> > > We believe this clarification better explains the rationale behind our experimental design.
> > >
> > > ---
> > >
> > > We thank the Reviewer for their feedback and are pleased that our responses have addressed their concerns. If so, we kindly ask that the evaluation be updated accordingly to reflect this.
> > >
> > > In accordance with the policy, while we would be happy to continue the discussion if further questions remain, this marks our final response.
> > >
> > > Best regards,
> > >
> > > Authors

---

### Decision · Program_Chairs · 2026-04-30

**Decision:**

Accept (regular)

**Comment:**

This paper presents a study of functional equivalence and parameter space symmetries for self-attention which considers sinusoidal and RoPE positional embeddings and the impact on mode connectivity.  Reviewers were generally positive praising in particular the valuable characterization of symmetries of self attention with RoPE and the wide breadth and quality of the experiments.   Several issues including the clarity of section 6 were partially addressed in the rebuttal. The biggest issues and lone dissenting vote came from 6aks.  I appreciate the strong engagement and discussion between 6aks and the authors.  The largest concerns are limited novelty relative to previous work and the scope of work both in that it only considers symmetries of self-attention and that the conclusions are overclaimed relative to what the theory and experiments show.  The issue is compounded by the volume of experimental data which could use more synthesis.  The authors would do well to take these critiques in to consideration in revision.